# ONE REFLECTION AT A TIME: PROVABLE COMPRESSION AND DENOISING OF ORTHOGONAL MATRICES

## ABSTRACT

Orthogonal matrices have become central to modern machine learning for several reasons, including gradient norm preservation, allowing invertible transformations, and supporting efficient parameterization. While any orthogonal matrix can be written as a product of Householder reflections, existing constructive methods do not address approximation, denoising, or reflector minimality under noise and computational constraints. We propose a simple greedy eigenspace-based algorithm that approximates or recovers an orthogonal matrix $\mathbf{V}$ using a product of Householder reflections. Our method (i) *preserves orthogonality* at every approximation level $K$, (ii) yields *provable per-iteration error bounds*, (iii) *terminates exactly* with a *minimality certificate*, and (iv) has *denoising guarantees* under additive random noise (Ginibre/GOE). Empirically, our approach outperforms previous methods across reconstruction error metrics. Experimental ablations on expRNN and ViT show controllable accuracy based on compression level when replacing learned weights with $K$-Householder products, with storage and matvec costs reduced from $\mathcal{O}(n^2)$ to $\mathcal{O}(Kn)$. Our results position greedy Householder products as a theoretically grounded tool for projection, compression, and denoising of orthogonal matrices in learning systems.

## 1 INTRODUCTION

Orthogonal transformations play a pivotal role across modern ML architectures, from stability in training to structured, invertible mappings. The ubiquity of orthogonal layers in today's ML systems stems from their ability to stabilize optimization, preserve norms, and support efficient structure. They are critical in dictionary learning (Rubinstein et al., 2010; Rusu et al., 2016), orthogonal RNNs (Arjovsky et al., 2016; Lezcano-Casado & Martınez-Rubio, 2019), normalizing flows (Tomczak & Welling, 2016), and orthogonal attention mechanisms in transformers (Büyükakyüz, 2024).

In many of these applications, large orthogonal matrices are either learned directly or used as priors or initialization layers. Classical decompositions (e.g., QR or SVD) lack structure, while fast transforms like DCT and Hadamard matrices are fixed and lack expressivity. In contrast, structured products of *Householder reflections* are both flexible and computationally efficient, making them a compelling tool for expressing orthogonal matrices. A Householder reflection is an orthogonal transformation defined by a single unit vector. It is defined as

$$\mathbf{H} = \mathbf{I} - 2\mathbf{u}\mathbf{u}^\mathsf{T}, \tag{1}$$

where $\mathbf{u} \in \mathbb{R}^n$ is a unit-norm vector, known as the *Householder vector*. Products of these reflections can express arbitrary orthogonal matrices, and their structured form enables significant gains in memory and compute.

Householder parameterizations are already used in RNNs (Mhammedi et al., 2017), where they reduce the cost of maintaining orthogonality to $\mathcal{O}(Tmn)$ for $m$ reflections. (Mathiasen et al., 2020) exploit their parallelizability in normalizing flows, while (Tomczak, 2024) highlight their use in latent density estimation. Importantly, these works demonstrate the scalability and ease of integration of Householder-based layers into deep architectures.

Recent work shows that even a small number of reflections can suffice in practice. (Yuan et al., 2024) demonstrates that only a few Householder vectors are needed to adapt pretrained LLMs.

(Huang et al., 2022) uses Householder-based attention for vision transformers to enforce orthogonality without expensive constraints. Similarly, (Geng et al., 2024) applies Householder orthogonalization to design efficient and sparse Capsule Networks.

Beyond deep learning, Householder methods are useful in sparse signal processing. Structured orthogonal dictionaries have been leveraged in speech (Ji et al., 2018), vision (Liu et al., 2021), denoising (Zhu & Ng, 2020), representation learning (Yüce et al., 2022), graphs (Vincent-Cuaz et al., 2021), and generative modeling (Zeng et al., 2023). Prior works (Rusu et al., 2016; Rusu & Thompson, 2017; Narayanan & Abhilash, 2022; Liu et al., 2025) show that Householder-based transforms are compact, interpretable, and powerful for these tasks. Motivated by this, we aim to address the following:

1. Expressiveness: How well can an orthogonal matrix be approximated by a small number (e.g., $\mathcal{O}(\log n)$) of Householder reflections?

2. Denoising: Can noisy orthogonal matrices be denoised using projections onto the space of such products?

In this paper, we propose a greedy, eigenspace-based algorithm to approximate or recover an orthogonal matrix using a minimal number of Householder reflections. Our approach has the following advantages:

1. **Provably accurate**, with formal approximation and projection guarantees;

2. **Robust to noise**, enabling denoising of structured orthogonal matrices;

3. **Widely applicable** to ML tasks, including dictionary learning, model initialization, and neural compression.

Our algorithm iteratively projects the input onto eigenspaces defined by Householder reflections. It yields provable iteration-wise error bounds and consistent, interpretable decompositions, unlike previous methods. This framework is amenable to applications such as fast orthogonal dictionary learning with theoretical guarantees, structured approximations of learned matrices in RNNs and transformers; efficient representation compression in neural and signal models, and low parameter orthogonal initializations in deep networks.

## 2 PRIOR WORK

**Constructive Decomposition of Orthogonal Matrices**    The work of (Uhlig, 2001) proves that any orthogonal matrix $\mathbf{V} \in \mathbb{R}^{n \times n}$ can be exactly expressed as a product of at most $n - m$ Householder reflections, where $m = \dim\left(\ker(\mathbf{V} - \mathbf{I}_n)\right)$. This result provides a guarantee for Householder-based decomposition, and its constructive proof offers a method to iteratively increase the dimension of the 1-eigenspace via reflections. The construction proceeds by identifying a direction $\boldsymbol{x}$ such that $\mathbf{V}\boldsymbol{x} \neq \boldsymbol{x}$, then defining a reflection $\mathbf{u} = \mathbf{V}\boldsymbol{x} - \boldsymbol{x}$, followed by normalization, increasing the 1-eigenspace dimension.

While elegant, this method is sensitive to perturbations: if $\mathbf{V}$ is only approximately factorizable by a few reflections (e.g., due to noise), the procedure cannot return a useful decomposition. The method also allows choosing $\boldsymbol{x}$ arbitrarily. However, different choices of $\boldsymbol{x}$ yield different factor sequences, affecting reproducibility and intermediate error. Moreover, these intermediate decompositions are not optimal projections in the span of $k$-reflection matrices (they are, in fact, sub-optimal with high probability). These limitations reduce its usefulness in learning settings where fast, approximate, and consistent decompositions are desirable.

**Comparison with other approximation methods.**    One of our goals is to approximate a given orthogonal matrix using a product of $K$ Householder reflectors, while ensuring that the result remains orthogonal for all $K \leq n$. This distinguishes our setting from common matrix approximation methods used in machine learning, such as truncated singular value decomposition (SVD), low-rank factorization, or pruning. These techniques approximate general matrices and do not, in general, preserve orthogonality, a key property in applications such as RNNs. Similarly, other techniques such as QR decompositions, Cayley/exponential (skew-symmetric) parameterizations, and compositions of Givens rotations also do not meet our requirements. See section A.1 for more details.

To the best of our knowledge, the only prior method that progressively approximates an orthogonal matrix using Householder reflections is the constructive decomposition algorithm described above, which we compare against directly. Nevertheless, we also add some experiments to show the utility of our method compared to some of these techniques in the appendix.

**Fast Orthonormal Sparsifying Transforms** The Hm-DLA algorithm proposed by (Rusu et al., 2016) tackles orthogonal dictionary learning via Householder reflections. The method alternates between two stages: sparse coding (solving for $\mathbf{X}$ given a current estimate of the orthogonal dictionary $\mathbf{V}$), and dictionary update (refining $\mathbf{V}$ as a product of $m$ reflections). The updates aim to minimize the reconstruction error $\|\mathbf{Y} - \mathbf{V}\mathbf{X}\|_F^2$. Its per-iteration complexity is $\mathcal{O}(n^2pm))$, given $\mathbf{X}, \mathbf{Y} \in \mathbb{R}^{n \times p}$ and the underlying assumption that $\mathbf{V}$ is a product of $m$ Householder reflections. Thus, the number of reflections $m$ is a user-defined hyperparameter and not minimized by the algorithm. The method does not provide theoretical guarantees on exact recovery or approximation quality. (Rusu & Thompson, 2017) also proposed a method for Dictionary Learning via Givens rotations, called Gm-DLA. It is noted in the paper that the performance of Gm-DLA is better than Hm-DLA only for very small $m$, and thus we do not add it here in our experiments for comparison.

**Our contribution:** We propose an algorithm that addresses the approximate regime by identifying the best approximation in a low-dimensional subspace defined by $k$ reflections. The method provides iteration-wise error control, is robust to noise, and guarantees a unique, interpretable decomposition aligned with the input's eigenspace structure. Our approach delivers a constructive, noise-aware decomposition with provable approximation bounds while minimizing the number of required reflections—crucial for memory- and compute-constrained settings. Please note that we apply our method only to square matrices, though it is possible to extend our method to rectangular matrices (see E.5). To the best of our knowledge, our method is the first to:

1. Approximate arbitrary orthogonal matrices via a greedy sequence of Householder projections;

2. Guarantee error reduction with explicit spectral error bounds;

3. Identify the minimal number of reflections $m$ needed to exactly represent a matrix in $\mathcal{H}_m$, or approximate the best projection otherwise;

4. Provide provable robustness to additive noise, including Frobenius-norm denoising bounds under standard random matrix models;

## 3 APPROXIMATING ORTHOGONAL MATRICES

### 3.1 PROBLEM STATEMENT AND DISCUSSION

Given a target real, orthogonal matrix $\mathbf{V} \in \mathbb{R}^{n \times n}$ with $\mathbf{V}^\mathsf{T}\mathbf{V} = \mathbf{I}$ and natural number $m$, we wish to determine real Householder matrices $\mathbf{H}_1, \mathbf{H}_2, \ldots, \mathbf{H}_m \in \mathbb{R}^{n \times n}$ that minimize $\|\mathbf{V} - \mathbf{H}_1\mathbf{H}_2 \ldots \mathbf{H}_m\|_F$. We denote the Householder matrices obtained from our algorithms as $\hat{\mathbf{H}}_k$ for some k and the product of these obtained matrices by $\hat{\mathbf{V}}_k$ for some k. Here $\|\mathbf{A}\|_F = \sqrt{\mathrm{tr}(\mathbf{A}^\mathsf{T}\mathbf{A})}$ denotes the Frobenius norm of a matrix $\mathbf{A}$. We let $\mathcal{H}_m = \{\mathbf{A} \in \mathbb{R}^{n \times n} : \mathbf{A} = \mathbf{H}_1\mathbf{H}_2 \ldots \mathbf{H}_k$ for some $k \leq m\}$ be the set of $n \times n$ real matrices that can be written as the product of some $k \leq m$ Householder matrices. Therefore, $\mathcal{H}_n$ includes all $n \times n$ orthogonal matrices. The computational and storage benefits of Householder matrices (see A.2, D.1.1) motivate the following problems: 1) approximate arbitrary matrices $\mathbf{A}$ with those in $\mathcal{H}_m$ for $m \ll n$, and 2) find the decomposition $\mathbf{V} \approx \hat{\mathbf{H}}_1\hat{\mathbf{H}}_2 \ldots \hat{\mathbf{H}}_k$.

We discuss an $\mathcal{O}(n^3m)$ algorithm ($\mathcal{O}(n^2m)$ on implementing Lanczos' algorithm in it) that, with input $\mathbf{V} \in \mathbb{R}^{n \times n}$ and $m$, finds an approximation to $\mathbf{V}$ in $\mathcal{H}_m$ and identifies the corresponding Householder matrices $\hat{\mathbf{H}}_i$ such that $\mathbf{V} \approx \prod \hat{\mathbf{H}}_i$. We show that the proposed algorithm correctly identifies if a given matrix $\mathbf{V} \in \mathcal{H}_m$, and finds corresponding Householder factors $\mathbf{H}_i$. For an arbitrary $\mathbf{V}$, our algorithm can be seen as a greedy approach to finding the projection of $\mathbf{V}$ on $\mathcal{H}_m$. We also obtain error bounds on the approximation. Note that we use the fact that the classical Householder decomposition is not necessarily unique (see A.1.1).

The proposed algorithm (Algorithm 1) operates in the eigenspace (instead of the column space like QR) and will correctly identify if $\mathbf{V} \in \mathcal{H}_m$. We hope that such approaches will be useful as a pre-computation step to decompose matrices $\mathbf{V}$ into Householder factors so that $\mathbf{Vx}$ can be approximated efficiently. We discuss some notations and ideas critical to our method in the following section.

### 3.2 NOTATION AND PRELIMINARIES

We denote by $\mathbf{u}_i$ the unit norm vectors corresponding to the Householder reflectors $\mathbf{H}_i$, i.e. $\mathbf{H}_i = \mathbf{I} - 2\mathbf{u}_i\mathbf{u}_i^\mathsf{T}$. We denote by $\mathbf{V}_{\mathrm{sym}} = (\mathbf{V} + \mathbf{V}^\mathsf{T})/2$ the symmetric part of an orthogonal matrix $\mathbf{V}$. We also denote by $\mathbf{I}_n$ (or simply $\mathbf{I}$ when $n$ is clear from the context) the $n \times n$ identity; for symmetric matrices $\mathbf{A}$, we denote by $\lambda_{\min}(\mathbf{A})$ the smallest eigenvalue of $\mathbf{A}$. We use $\mathrm{tr}(\mathbf{M})$ to represent the trace of the matrix $\mathbf{M}$, $\det(\mathbf{M})$ to denote its determinant and $\mathrm{mult}(\mathbf{M}, \lambda)$ or $m_\lambda$ to denote the (geometric) multiplicity of its eigenvalue $\lambda$.

Before we state our main results, we make the following simple observations. Let $\mathcal{E}_\mathbf{V}^1$ be the eigenspace of the matrix $\mathbf{V}$ corresponding to the eigenvalue 1, i.e. $\mathcal{E}_\mathbf{V}^1 = \{\mathbf{x} : \mathbf{Vx} = \mathbf{x}\}$. We see that for Householder matrices $\mathbf{H}_i = \mathbf{I} - 2\mathbf{u}_i\mathbf{u}_i^\mathsf{T}$, the space $\mathcal{E}_{\mathbf{H}_i}^1$ is the $n-1$ dimensional subspace orthogonal to $\mathbf{u}_i$. Note also that if $\mathbf{V} = \mathbf{H}_1\mathbf{H}_2\ldots\mathbf{H}_m$, then

$$\cap\mathcal{E}_{\mathbf{H}_i}^1 \subseteq \mathcal{E}_\mathbf{V}^1.$$

Thus, the eigenspace of $\mathbf{V}$ corresponding to eigenvalue 1 includes the subspace $\cap\mathcal{E}_{\mathbf{H}_i}^1$: the latter subspace is orthogonal to all $\mathbf{u}_1, \mathbf{u}_2, \ldots, \mathbf{u}_m$ and hence is of dimension at least $n - m$. Thus if $\mathbf{V} = \mathbf{H}_1\mathbf{H}_2\ldots\mathbf{H}_m \in \mathcal{H}_m$, then $\mathcal{E}_\mathbf{V}^1$ is at least $n - m$ dimensional; this forms a necessary condition for $\mathbf{V} \in \mathcal{H}_m$. This condition is known to be sufficient as well Uhlig (2001).

For symmetric orthogonal matrices, finding the Householder decomposition is equivalent to the eigenvalue decomposition (see section A.3). The result below generalizes this approach to arbitrary (non-symmetric) real orthogonal matrices with the following goals: Given $\mathbf{V}$ and $m$: 1) identify if $\mathbf{V} \in \mathcal{H}_m$, and 2) Find $\hat{\mathbf{H}}_1, \hat{\mathbf{H}}_2, \ldots, \hat{\mathbf{H}}_m$ such that $\mathbf{V} \approx \hat{\mathbf{H}}_1\hat{\mathbf{H}}_2\ldots\hat{\mathbf{H}}_m$, and 3) give a bound on the error in this approximation.

### 3.3 ALGORITHM

Consider the case when $m = 1$, i.e., approximating an input matrix $\mathbf{V}$ with a single Householder matrix $\mathbf{H}$. We have the target squared error objective $\|\mathbf{V} - \mathbf{H}\|_F^2$, with $\mathbf{H} = \mathbf{I} - 2\mathbf{uu}^T$:

$$
\begin{aligned}
\hat{\mathbf{H}} &= \underset{\mathbf{H} \in \mathcal{H}_1}{\arg\min}\|\mathbf{V} - \mathbf{H}\|_F^2 \\
&= \underset{\mathbf{H} \in \mathcal{H}_1}{\arg\min}\left(2n - 2\mathrm{tr}(\mathbf{H}^\mathsf{T}\mathbf{V})\right) \\
\hat{\mathbf{u}} &= \underset{\|\mathbf{u}\|_2=1}{\arg\min}\left(2n - 2\mathrm{tr}(\mathbf{V}) + 4\mathbf{u}^\mathsf{T}\mathbf{Vu}\right) \text{ (from 1)} \\
&= 2n - 2\mathrm{tr}(\mathbf{V}) + 4\lambda_{\min}(\mathbf{V}_{\mathrm{sym}}),
\end{aligned}
\tag{2}
$$

where in the last step we have used $\min(\mathbf{u}^\mathsf{T}\mathbf{Vu}) = \min(\mathbf{u}^\mathsf{T}\mathbf{V}_{\mathrm{sym}}\mathbf{u}) = \lambda_{\min}(\mathbf{V}_{\mathrm{sym}})$. Thus, the projection of $\mathbf{V}$ in $\mathcal{H}_1$ is obtained as $\mathbf{I} - 2\hat{\mathbf{u}}\hat{\mathbf{u}}^\mathsf{T}$, where $\hat{\mathbf{u}}$ is a unit eigenvector corresponding to the smallest eigenvalue of the symmetric part $\mathbf{V}_{\mathrm{sym}}$.

For approximation in $\mathcal{H}_m$, for $m > 1$, The error $\|\mathbf{V} - \hat{\mathbf{H}}\|_F^2$ may not reduce as favorably as in the $m = 1$ case. Hence we adopt the following strategy to find a optimal solution: first, approximate the input $\mathbf{V}$ in $\mathcal{H}_1$ (as in observation equation 2 above). Let $\mathbf{H}_1$ be the approximation obtained. We then construct $\mathbf{V}_1 = \hat{\mathbf{H}}_1^\mathsf{T}\mathbf{V} = \hat{\mathbf{H}}_1\mathbf{V}$ and approximate $\mathbf{V}_1$ in $\mathcal{H}_1$ to obtain $\hat{\mathbf{H}}_2$, and repeat the process for $\mathbf{V}_2 = \hat{\mathbf{H}}_2\mathbf{V}_1$. This algorithm is summarized in Algorithm 1. Theorem 1 below gives performance guarantees on this algorithm, which we later show to be optimal B.4.

**Theorem 1.** *Algorithm 1 terminates at step $m$ iff $\mathbf{V} \in \mathcal{H}_m$. Furthermore, this is the smallest $m$ such that $\mathbf{V} \in \mathcal{H}_m$.*

**Theorem 2.** *Algorithm 1, if truncated to $k$ steps gives the lowest possible reconstruction error among all $\hat{\mathbf{V}} \in \mathcal{H}_k$ for a given $\mathbf{V} \in \mathcal{H}_m$.*

**Theorem 3.** *For an arbitrary input $\mathbf{V} \in \mathbb{R}^{n \times n}$, the Householder decomposition $\hat{\mathbf{V}}_m$ obtained from Algorithm 1 (truncated to $m$ steps) satisfies the error bound*

$$\|\mathbf{V} - \hat{\mathbf{V}}_m\|_F \leq \sqrt{2\left(n - \operatorname{tr}(\mathbf{V}) - 2\lfloor m/2 \rfloor + \sum_{i=1}^{m} \lambda_i\right)}$$

*where the $\lambda_1 \leq \lambda_2 \leq \ldots$ are the eigenvalues of $\mathbf{V}_{sym}$.*

Note that the Householder reflectors obtained by the algorithm are not necessarily orthogonal (as in the case when the input $\mathbf{V}$ is symmetric). The construction in Uhlig (2001) does not use the eigenvectors of the matrix and thus does not come with approximation bounds. This bound is valuable in various deep learning settings where there's a trade-off between the accuracy of approximating a general rotation using reflectors and the benefits of reduced computational complexity.

---

**Algorithm 1** Approximating an orthogonal matrix in $\mathcal{H}_m$

---

**Input:** orthogonal matrix $\mathbf{V}$
**Output:** Householder matrices $\mathbf{H}_1, \ldots, \mathbf{H}_m$; total count $m$

1: Set $k = 0$, $\hat{\mathbf{V}} = \mathbf{I}$, $\mathbf{V}_k = \mathbf{V}$
2: **while** $\lambda_{\min}(\mathbf{V}_k) \neq 1$ and $k < n$ **do**
3:     Compute $\mathbf{V}_k = \hat{\mathbf{V}}^\top \mathbf{V}$
4:     Let $\mathbf{M}_k = (\mathbf{V}_k + \mathbf{V}_k^\top)/2$
5:     Let $\mathbf{u}_k$ be eigenvector of $\mathbf{M}_k$ corresponding to $\lambda_{\min}$
6:     $\mathbf{H}_k = \mathbf{I} - 2\mathbf{u}_k\mathbf{u}_k^\top$
7:     $\hat{\mathbf{V}} \leftarrow \hat{\mathbf{V}}\mathbf{H}_k$
8:     $k \leftarrow k + 1$
9: **end while**
10: **return** $\mathbf{H}_1, \ldots, \mathbf{H}_k, m = k$

---

Note the subtle difference between $\hat{\mathbf{V}}$ and $\mathbf{V}_k$. The proofs presented in B.3 rely on $\mathbf{V}_k$. Also note that the approximation does not require $m$ and outputs $m$ if the input matrix is exactly orthogonal. One can also choose to approximate the input orthogonal matrix in a smaller space of a product of fewer than $m$ Householder reflectors by simply stopping the algorithm at the required number of iterations. If a given input matrix isn't orthogonal, we can consider it to be an orthogonal matrix to which noise has been added, as elaborated in Section 4.

### 3.4 PROOF

**Lemma 1.** *We recall the following known facts about real orthogonal matrices. We use these observations for the proof of Theorem 1 and Theorem 3.*

1. *Any orthogonal matrix $\mathbf{V} \in \mathbb{R}^{n \times n}$ is normal, and hence has a decomposition of the from $\mathbf{V} = \mathbf{U}\mathbf{D}\mathbf{U}^\star$ where $\mathbf{U} \in \mathbb{C}^{n \times n}$ is a unitary matrix, and $\mathbf{D} \in \mathbb{C}^{n \times n}$ is a diagonal matrix with diagonal entries complex units. In particular, eigenvectors corresponding to different eigenvalues of $\mathbf{V}$ are orthogonal.*

2. *If $\lambda$ is an eigenvalue of $\mathbf{V}$, so is $\bar{\lambda}$. If $\mathbf{w}$ is an eigenvector corresponding to $\lambda$, then $\bar{\mathbf{w}}$ is an eigenvector corresponding to $\bar{\lambda}$. Also, since the complex eigenvalues appear in conjugate pairs, $\det(\mathbf{V}) = \prod \lambda_i = -1$ if and only if $-1$ is an eigenvalue of $\mathbf{V}$ with odd multiplicity.*

3. *The eigenvectors of $\mathbf{V}_{sym}$ are the diagonal entries of $Re(\lambda)$ (counted with multiplicities). If $\mathbf{V}\mathbf{w} = \lambda\mathbf{w}$ then $Re(\lambda)$ is an eigenvalue of $\mathbf{V}_{sym}$ with corresponding eigenvectors $\mathbf{w} + \bar{\mathbf{w}}$ and $(\mathbf{w} - \bar{\mathbf{w}})/i$. Likewise, every eigenvector of $\mathbf{V}_{sym}$ is associated with eigenvectors $\mathbf{z}$ and $\bar{\mathbf{z}}$ of $\mathbf{V}$.*

4. *If $\mathbf{w}$ is an eigenvector of $\mathbf{V}$ with eigenvalue $\lambda$, then $\mathbf{w}$ is an eigenvector of $\mathbf{V}^T$ with eigenvalue $\bar{\lambda}$. If $\lambda = \pm 1$, then any eigenvector of $\mathbf{V}_{sym}$ with eigenvalue $\lambda$ is also an eigenvector of $\mathbf{V}$ with eigenvalue $\lambda$.*

**Lemma 2.** *Consider the vector $\mathbf{u}$ in the algorithm at step $k$ (the eigenvector corresponding to eigenvalue $\lambda_{\min}$ of $(\mathbf{V}_k)_{sym}$). Let $\{\mathbf{z}, \bar{\mathbf{z}}\}$ be the eigenvectors of $\mathbf{V}_k$ associated to $\mathbf{u}$. Every eigenvector $\mathbf{w}$ of $\mathbf{V}_k$ other than $\{\mathbf{u}, \bar{\mathbf{u}}\}$ is an eigenvector of $\mathbf{V}_{k+1}$. Further, if $\lambda_{\min} = -1$, the vector $\mathbf{u}$ picked in step $k$ of the algorithm satisfies $\mathbf{V}_{k+1}\mathbf{u} = (\mathbf{V}_{k+1})_{sym}\mathbf{u} = \mathbf{u}$.*

To prove Theorem 1, we show that the eigenspace $\mathcal{E}^1_{\mathbf{V}_k}$ corresponding to the eigenvalue 1 increases in dimension by 1 for each iteration.

**Lemma 3.** *The trace and eigenspace for eigenvalue 1 increase at each iteration: $tr(\mathbf{V}_{k+1}) - tr(\mathbf{V}_k) = -2\lambda_{\min}(\mathbf{V}_k)_{sym}$, and $dim(\mathcal{E}^1_{\mathbf{V}_{k+1}}) - dim(\mathcal{E}^1_{\mathbf{V}_k}) = 1$.*

### 3.4.1 PROOF OF THEOREM 1, THEOREM 2, AND THEOREM 3

Algorithm 1 terminates when $\hat{\mathbf{V}} = \mathbf{V}$ or equivalently when $\mathbf{V}_{m+1} = \mathbf{I}$. In this case, by construction, $\hat{\mathbf{V}}$ is a product of $m$ Householder matrices. Now suppose by way of contradiction $\mathbf{V} \in \mathcal{H}_p$ for some $p < m$. Then as observed before, the eigenspace $\mathcal{E}^1_{\mathbf{V}}$ is at least $n - p$ dimensional. By Lemma 3 above, at iteration $p$, $\mathbf{V}_{p+1}$ would have an $n-$dimensional eigenspace; so the algorithm should have terminated before step $p$. Theorem 2 follows from the fact that the greedy solution is optimal for the first iteration and the reduction in error at every subsequent iteration is also optimal. The result swiftly follows from induction B.4.

For Theorem 3, we have $\min \|\mathbf{V} - \hat{\mathbf{V}}\|^2_F = \min 2[n - tr(\hat{\mathbf{V}}^\top \mathbf{V})] = 2[n - tr(\mathbf{V}_{k+1})]$. Note that Lemma 3 gives a recursion on $tr(\mathbf{V}_{k+1})$ in terms of the smallest eigenvalue of $(\mathbf{V}_k)_{\text{sym}}$. Similar to the proof of Lemma 3, we note that the smallest eigenvalue of $(\mathbf{V}_k)_{\text{sym}}$ is $-1$ for every other iteration (when $p + k$ as in Proof of Lemma 3 is odd). In any iteration, no new eigenvalues other than $\pm 1$ are introduced. Thus $\lambda_{\min}(\mathbf{V}_k)_{\text{sym}}$ is the $k/2^{\text{th}}$ smallest eigenvalue of $\mathbf{V}_{\text{sym}}$.

$$\|\mathbf{V} - \hat{\mathbf{V}}_{\mathbf{m}}\|^2_F = 2\left(n - tr(\mathbf{V}) - 2\sum_k \lambda_{\min}(\mathbf{V}_{\mathbf{k}})_{\text{sym}}\right), \tag{3}$$

so the statement of the theorem follows (note that the eigenvalues appear in pairs, allowing us to rewrite as in the statement of the theorem; refer to B.6 for more details). Finally, note that each step of the algorithm requires an eigen decomposition, and thus takes at most $\mathcal{O}(n^3)$ time. On using Lanczos' algorithm, this is roughly $\mathcal{O}(n^2)$ per iteration. See Table 2 in section D for more details.

## 4 DENOISING

In many practical scenarios, the orthogonal matrix of interest is corrupted by noise—either during estimation or as a result of data perturbations. This has several applications in computer vision, computer graphics Bhamre et al. (2015); Wang & Singer (2013), and even point cloud networks Daigavane et al. (2025); Lawrence et al. (2019). Other applications included standard signal processing techniques such as blind source separation where mixing matrices are often constrained to be orthogonal. However, due to noise in the measurements, this is not always the case, leading to noisy estimates of mixing matrices Huang et al. (2015). Gaussian noise in measurements impacts performance here and techniques have been developed to try to deal with the same Arora et al. (2012). Other forms of noise can arise in orthogonal weight matrices in RNNs and transformers- during quantization, communication during federated learning, pruning, or even dropout, making denoising a critical problem in modern ML systems. Note that the naïve application of the Orthogonal Procrustes solution doesn't guarantee recovery of the ground truth matrix (refer C.4.1). This section shows that a variant of our greedy Householder decomposition algorithm can be adapted for denoising such matrices. Specifically, given a noisy orthogonal matrix $\mathbf{V} = \mathbf{V}^* + \mathbf{N}$, where $\mathbf{V}^* \in \mathbb{R}^{n \times n}$ is orthogonal and $\mathbf{N}$ is an additive noise matrix, we aim to recover a clean estimate $\hat{\mathbf{V}} \approx \mathbf{V}^*$. Our denoising approach builds on Algorithm 1 by applying $m$ Householder projections—where $m$ is the number of Householder matrices composing the true orthogonal matrix $\mathbf{V}^*$.

### 4.1 DENOISING A SINGLE HOUSEHOLDER MATRIX

We begin with the following setting: the ground truth orthogonal matrix is a single Householder reflection of the form $\mathbf{H}^* = \mathbf{I} - 2\mathbf{u}^*\mathbf{u}^{*\top}$, where $\mathbf{u} \in \mathbb{R}^n$ is a unit vector. The observed matrix is a

noise-perturbed version:

$$\mathbf{V} = \mathbf{H}^* + \frac{1}{n^\alpha}\mathbf{N}, \tag{4}$$

where $\mathbf{N} \in \mathbb{R}^{n \times n}$ is a real Ginibre matrix with i.i.d. entries $\mathbf{N}_{ij} \sim \mathcal{N}(0,1)$, and $\alpha > 0$ controls the noise level. Let $\mathbf{N}_{\text{sym}} = (\mathbf{N} + \mathbf{N}^\top)/2$ denote the symmetric part of the noise matrix. This is a real Wigner-type matrix. Following the notation above, we have:

**Lemma 4.** *Consider a noise perturbed Householder matrix $\mathbf{V} = \mathbf{H}^* + (1/n^\alpha)\mathbf{N}$ where $\mathbf{N}$ is a real Ginibre matrix. The minimum eigenvalue of $\mathbf{V}_{sym}$ satisfies*

$$\lambda_{\min}(\mathbf{V}_{sym}) \leq -1 + Cn^{1/2-\alpha} \quad w.h.p^1$$

**Theorem 4.** *Following from Lemma 4, the error between the denoised Householder matrix $\hat{\mathbf{H}}$ (obtained by denoising $\mathbf{V}$) and ground truth Householder matrix $\mathbf{H}^*$ satisfies the following bound:*

$$\left\|\hat{\mathbf{H}} - \mathbf{H}^*\right\|_F^2 \leq Cn^{1-2\alpha} \quad w.h.p$$

This shows that the denoising error decays polynomially in $n$, and improves with stronger decay of noise (i.e., larger $\alpha$). This theoretical behavior is reflected empirically in Figure 4.

## 4.2 DENOISING ARBITRARY ORTHOGONAL MATRICES

We now extend the above to the general case, i.e., we try to denoise a noise perturbed orthogonal matrix where the ground truth matrix is a product of $m$ Householder reflectors:

$$\mathbf{V} = \mathbf{H}_1^*\mathbf{H}_2^* \cdots \mathbf{H}_m^* + \frac{1}{n^\alpha}\mathbf{N}, \tag{5}$$

It is difficult to use the ideas in section 4.1 directly for a product of $m$ Householder matrices to which noise has been added. Thus, we first use a trick to get it into a form where all of the aforementioned ideas are directly applicable. From Theorem 1, we know that there exists a minimal decomposition of any given product of Householder reflectors. We represent these reflectors by $\tilde{\mathbf{H}}_1, \tilde{\mathbf{H}}_2 \cdots \tilde{\mathbf{H}}_m$ .Thus, instead of attempting to denoise the above, we attempt to denoise the corresponding optimal projection in $\mathcal{H}_m$. Thus, we denoise

$$\mathbf{V} = \tilde{\mathbf{H}}_1\tilde{\mathbf{H}}_2 \cdots \tilde{\mathbf{H}}_m + \frac{1}{n^\alpha}\mathbf{N}, \tag{6}$$

Though equations 5 and 6 have an identical LHS, denoising $\mathbf{V}$ from equation 6 aligns perfectly with our algorithm and helps simplify the derivation of the theoretical bounds. The reflector vectors our algorithm chooses are close to the reflectors of the Householders in equation 6, and not those of equation 5, despite the fact that both expressions are the same noisy version of the ground truth orthogonal matrix. Let the recovered matrices be $\hat{\mathbf{H}}_1 \cdots \hat{\mathbf{H}}_m$ and $\mathbf{V}^* = \mathbf{H}_1^*\mathbf{H}_2^* \cdots \mathbf{H}_m^*$.

**Theorem 5.** *Following the notation above, for an arbitrary scaled-GOE perturbed orthogonal matrix $\mathbf{V} \in \mathbb{R}^{n \times n}$ and the recovered matrix $\hat{\mathbf{V}} = \hat{\mathbf{H}}_1\hat{\mathbf{H}}_2 \cdots \hat{\mathbf{H}}_m$, the error between between $\mathbf{V}^*$ and $\hat{\mathbf{V}}$ can be bounded above w.h.p., for some universal constant $\theta$ as follows:*

$$\|\hat{\mathbf{V}} - \mathbf{V}^*\|_F^2 \leq \mathcal{O}\Big(\frac{m \log^\theta n}{n^{2\alpha}} \sum_{k=1}^m \|\mathbf{S}^{(k)}\|_F^2\Big) + \mathcal{O}\Big(m \sum_{k=1}^m \|\Delta\mathbf{A}^{(k)}\|_2^3\Big)$$

*where $\mathbf{A}^{(k)} = \mathbf{V}_{sym}^{(k)}$, the iterate at the $k^{th}$ step and $\Delta\mathbf{A} = (\Delta\mathbf{V})_{sym}^{(k)}$ is the corresponding scaled noise term at that step. $\mathbf{S}^{(k)}$ is the reduced resolvent corresponding to the projector defined for that iterate ($\sum_{j=1}^{r_k} \mathbf{v}_j\mathbf{v}_j^\top$; $r_k$ is the rank of the projector- either 1 or 2 depending on the iteration). Furthermore, if $\mathbf{V}^*$ is Haar-distributed on $O(n)$, the bound reduces to*

$$\|\hat{\mathbf{V}} - \mathbf{V}^*\|_F^2 \leq \mathcal{O}\Big(\frac{mn \log^{\theta+1} n}{n^{2\alpha}}\Big) + \mathcal{O}\Big(m \sum_{k=1}^m \|\Delta\mathbf{A}^{(k)}\|_2^3\Big)$$

Figure 4 shows the denoising results. $\alpha > 1/2$ is required for the error to decay with $n$. More details on the previous couple of sections are available in Appendix section C.

---

[1] with high probability (refer to 58 for more details)

---

**Algorithm 2** Denoising an orthogonal matrix

---

**Input:** $\mathbf{V} = \mathbf{V}^* + \mathbf{N}$, a noisy orthogonal matrix, stopping parameter $\epsilon$;

**Output:** $\hat{\mathbf{V}}$, the denoised estimate of $\mathbf{V}^*$

1: Set $k = 0$, $\hat{\mathbf{V}} = \mathbf{I}$, $\mathbf{V}_0 = \mathbf{V}$
2: **while** $\lambda_{\min}(\mathbf{V}_k)_{\text{sym}} < \epsilon$ **do**
3:     Compute the symmetric part: $(\mathbf{V}_k)_{\text{sym}} = (\mathbf{V}_k + \mathbf{V}_k^\mathsf{T})/2$
4:     Let $\hat{\mathbf{u}}$ be the eigenvector of $(\mathbf{V}_k)_{\text{sym}}$ corresponding to $\lambda_{\min}$
5:     Define $\hat{\mathbf{H}}_k = \mathbf{I} - 2\hat{\mathbf{u}}\hat{\mathbf{u}}^\mathsf{T}$
6:     Update: $\mathbf{V}_{k+1} \leftarrow \hat{\mathbf{H}}_k\mathbf{V}_k$, $\hat{\mathbf{V}} \leftarrow \hat{\mathbf{V}}\hat{\mathbf{H}}_k$
7:     $k \leftarrow k + 1$
8: **end while**
9: **Return** $\hat{\mathbf{V}}$

---

**An elaboration of the stopping condition in the denoising algorithm**   If $m$ is already known, one can simply run Algorithm 2 while $k < m$. However, when it isn't known, as is the case in most practical applications, it suffices to find the point at which the minimum eigenvalue crosses $\epsilon = 0.9$. This is reasonably close to optimality. See section E.3 for a detailed discussion.

**Discussion**   This analysis shows that our algorithm can identify the correct eigendirection for a Householder reflector under high-dimensional noise, using only the symmetry of the input. To the best of our knowledge, this is the first time such a recovery guarantee has been derived for a greedy eigenspace-based orthogonal factorization, particularly one that leverages spectral structure in a way that is both theoretically grounded and algorithmically efficient. In Section C.5, we provide an extension of the main theoretical guarantees to arbitrary noise distributions- though these can be viewed as slightly weaker bounds on the error.

## 5   SIMULATIONS

We empirically evaluate the reconstruction capabilities of our method across different matrix constructions and baselines. We also conduct an ablation study, applying our techniques to orthogonal RNNs and ViTs as a compression technique on the trained recurrent and attention matrices, respectively. Moreover, we show that our method works as a compression technique for initialization, where we compress all of the mimetic initialization weights (Trockman & Kolter, 2023) onto the space of a product of Householder matrices and then train the ViT on CIFAR10. We show that even with a few Householder reflectors, we can retain the accuracy values obtained by using the original initialization. We include some of the experiments here and place the rest of them, along with their details, in section D.

All experiments use an orthogonal matrix $\mathbf{V} \in \mathbb{R}^{n \times n}$ (with $n = 500$ unless stated otherwise), constructed as a product of $m$ Householder reflections. Each Householder reflection $\mathbf{H}_i = \mathbf{I} - 2\mathbf{u}_i\mathbf{u}_i^\mathsf{T}$ is generated from a unit-norm vector $\mathbf{u}_i \in \mathbb{R}^n$ ($n = 500$ unless stated otherwise). Unless specified, we consider the $\mathbf{u}_i$'s to be drawn as $\mathbf{u}_i \sim \mathcal{N}(0, \mathbf{I})$, normalized to unit norm, though our simulation results hold for any choice of the $\mathbf{u}_i$'s. The error metric used is defined as follows: **Reconstruction error:** $\|\mathbf{V} - \hat{\mathbf{V}}_k\|_F$, where $\hat{\mathbf{V}}_k$ is the approximation after $k$ Householder reflections. $m$ refers to the ground truth number of reflections, while index $k$ refers to the current iterate. For the illustration of noisy recovery, the noisy matrix is generated as

$$\mathbf{V} = \mathbf{V}^* + \frac{1}{n^\alpha}\mathbf{N}, \tag{7}$$

where $\mathbf{N}$ is a Ginibre matrix, and $\alpha$ is varied to generate various SNR values (see D.3 for more details, D.3.4 for other noise types). All other experiments have been included in Appendix D.

## 6   DISCUSSION AND FUTURE WORK

While our method provides a provably efficient and robust framework for approximating orthogonal matrices using Householder products, certain limitations arise due to the inherent structural properties of these reflections. In particular, approximating matrices such as $-\mathbf{I}$ requires the maximal number of reflections, $m = n$, since each Householder matrix $\mathbf{H} = \mathbf{I} - 2\mathbf{u}\mathbf{u}^\top$ preserves a

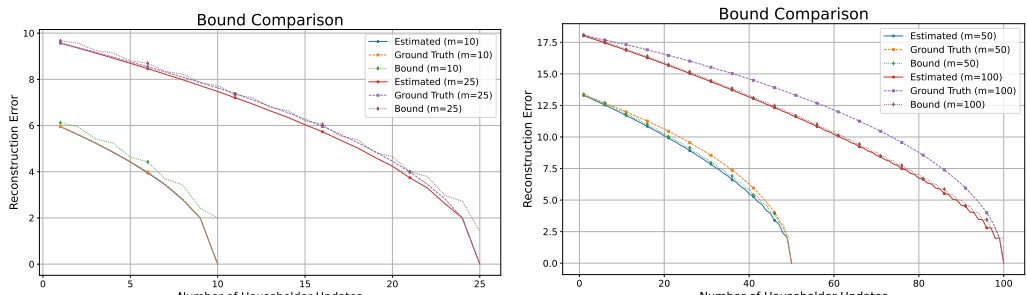

Figure 1: Reconstruction error vs. number of Householder updates for our method vs. the bound presented in Theorem 3 vs. the ground truth products. $m = 10, 25$ (left), $m = 50, 100$ (right); $n = 500$. Our bound is very close to the actual value of error derived from the products. Since the bound relies solely on the initial orthogonal matrix provided and the number of iterations, we can provide this value with only one eigenvalue decomposition. Also, observe that our Householder products give lower error than the ground truth Householder products while comparing distance from the input orthogonal matrix, not only indicating non-uniqueness, but also an improvement.

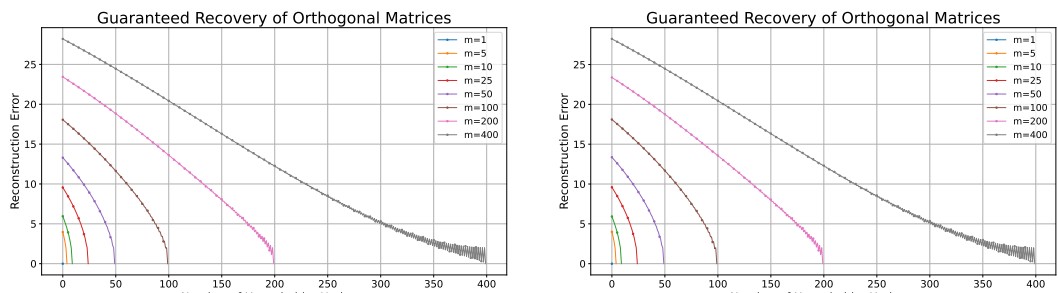

Figure 2: Reconstruction error vs. number of Householder updates (which is equal to the number of iterations of Algorithm 1) plots for two different distributions of the Householder vector: **Gaussian:** $\mathbf{u}_i \sim \mathcal{N}(0, \mathbf{I})$, normalized to unit norm; **Bernoulli:** Entries of $\mathbf{u}_i$ are i.i.d. from $\{-1, +1\}$, normalized. Since $\mathbf{V}$ is a product of $m$ Householder matrices, our method achieves zero error at update $m$, with only finite precision residual remaining. More experiments for other distributions have been added in the appendix in section D.2.1.

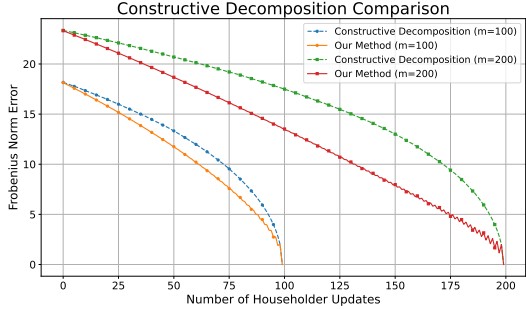

Figure 3: Reconstruction error vs. number of Householder updates for our method vs. the constructive decomposition of (Uhlig, 2001); $m = 100$ and $m = 200$, $n = 500$. Unlike their randomized eigenvalue-one-eigenspace augmentation approach, our algorithm yields a stable decrease in reconstruction error, since each update carefully projects $\mathbf{V}$ onto the span of the first $k$ Householder products.

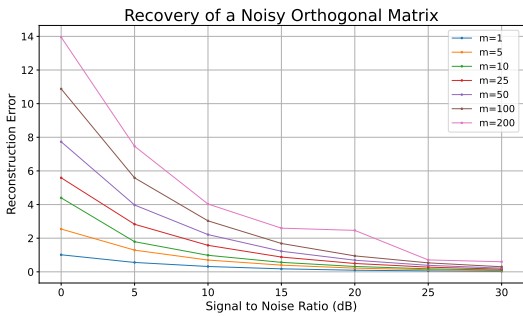

Figure 4: Reconstruction error vs. signal-to-noise ratio (SNR) in dB. Our method improves significantly with increasing SNR. We use $n = 500$ and scaled Ginibre noise here.

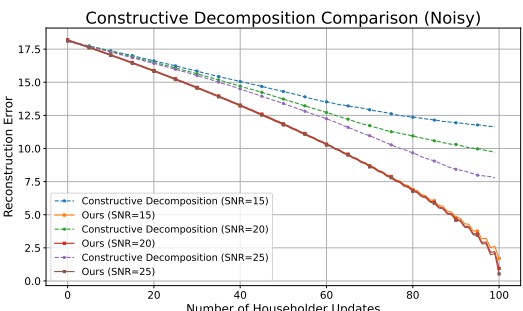

Figure 5: Comparison with (Uhlig, 2001) under scaled Ginibre noise. Our method achieves lower reconstruction error throughout, outperforming the constructive decomposition by more than a factor of 10 for certain SNR values. Our method is far more robust to noise. We use $n = 500$.

large subspace point-wise (i.e., the 1-eigenspace has co-dimension 1). As shown in equation 11, $-\mathbf{I}$ admits a decomposition as a product of $n$ Householder reflections, but no fewer, as implied by Theorem 1. This highlights a fundamental issue: while Householder matrices are computationally attractive (requiring $\mathcal{O}(n)$ time and storage), they may be inefficient for approximating certain orthogonal matrices. It is natural to consider augmenting or generalizing the basic building block.

One promising direction is to explore generalized reflectors of the form: $\mathbf{G} = z_1\mathbf{I} - z_2\mathbf{uu}^\top$, where $z_1, z_2 \in \mathbb{C}$ are subject to constraints (e.g., unitary or norm-preserving conditions) to ensure $\mathbf{G}$ remains orthogonal or near-orthogonal. Even if we limit these to the constraints we used for this work, we would be able to represent $-\mathbf{I}$ with just a couple of generalized reflectors instead of $n$. Under slightly weaker constraints, these structures could provide a richer class of building blocks, capable of capturing more diverse orthogonal behavior with fewer terms. Another direction is to investigate low-rank corrections or sparse perturbations to Householder products, which could improve representational power without sacrificing the structure needed for fast computation. From a learning perspective, investigating implicit regularization effects of such constrained parameterizations in deep networks (e.g., in ViTs or RNNs) remains an open and exciting area.

While our current focus has been on real-valued orthogonal matrices, many applications in signal processing and quantum computing require complex unitary matrices. Extending our method to the complex domain—and analyzing its theoretical and empirical behavior—would substantially broaden its applicability.

Finally, we believe that our method can not only be used after training procedures (especially when parameters are constrained to the orthogonal manifold) to identify if learned parameter matrices lie in $\mathcal{H}_m$ with $m < n$, which will help with both storage and computation, but also to analyze and control regularization. We also hope that some of the ideas introduced in this work can be used across a broad range of applications in the future- in machine learning, signal processing, optimization, and beyond.

# 7 REPRODUCIBILITY DETAILS

All simulations were implemented in Python using NumPy and Matplotlib. Random orthogonal matrices $\mathbf{V}$ were constructed as products of $m$ Householder matrices, with reflector vectors sampled from standard Gaussian or Bernoulli distributions and normalized to unit norm. Sparse signal matrices $\mathbf{X}$ used for reconstruction experiments had sparsity $1/n$ with nonzero entries drawn from $\mathcal{N}(0, 1)$.

Noise matrices $\mathbf{N}$ in the denoising experiments were drawn from a real Ginibre ensemble (entries i.i.d. $\mathcal{N}(0, 1)$) and scaled by $1/n^\alpha$. All eigenvalue computations used NumPy's 'eig' or 'eigh' methods. The Lanczos' algorithm is directly imported from scipy.sparse.linalg and we use 'eigsh' for the same. Timing is done using the time module.

Reconstruction error was measured using both absolute and relative Frobenius norms.

Experiments, except those in the ablation study, were run on a single CPU core (11th Gen Intel i5-11357; 16GB RAM) and did not require GPU acceleration. The experiments in the ablation study were run on one NVIDIA T4 GPU. Please refer to sections 5, D.3, and D.4 for more details.

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

## A  PRELIMINARIES

### A.1  OTHER FACTORIZATIONS

1. *QR decomposition:* The QR decomposition returns a full orthogonal matrix but does not yield structured approximations with a tunable number of reflectors. Though the matrix constructed in the intermediate steps $\mathbf{Q} = \mathbf{H}_1 \mathbf{H}_2 \cdots \mathbf{H}_k$ is orthogonal, it is not optimal.

2. *Cayley/exponential (skew-symmetric) parameterizations:* Such parameterizations ensure that the parameter matrices in networks stay in the space of orthogonal matrices, but they do not have any inherent compression capabilities or representational benefits—in terms of storage or time complexity of matvec.

3. *Givens rotations:* Though Givens rotations are used for fast multiplications, sequences of such rotations do not have a canonical per-step "best next rotation" toward a fixed target $\mathbf{V}$. Unless one solves a separate optimization, there is no built-in guarantee that adding one more rotation is the locally optimal projection, nor that the truncation error decreases with the number of Givens updates.

Alternative parameterizations typically update all parameters globally and do not provide these per-step projection guarantees or certificates by default. Furthermore, none of the above methods involves any denoising capability. Our method, using Householder matrices and their products, offers expressivity similar to these methods but with greedy guarantees and denoising theory.

Table 1: Comparison of orthogonal matrix approximation methods. Only our method and the constructive decomposition (Uhlig, 2001) support $K$-Householder decomposition with orthogonal outputs at each $K$.

| Method | Orthogonal for all $K$ | Structured | Solves Our Task? |
|---|---|---|---|
| Truncated SVD | No | No | No |
| QR decomposition | Yes | Yes | No |
| Cayley/exponential (skew-symmetric) | Yes | No | No |
| Givens rotations | Yes | Yes | No |
| **Constructive decomposition** | Yes | Yes | Yes |
| **Ours** | Yes | Yes | Yes |

### A.1.1  EXAMPLE OF QR DECOMPOSITION

Note also that the Householder decomposition as above is not necessarily unique (example below). The classic Householder QR decomposition Golub & Van Loan (2013) obtains the decomposition

$$\mathbf{A} = \mathbf{H}_1 \mathbf{H}_2 \cdots \mathbf{H}_k \mathbf{R},$$

for an arbitrary matrix $\mathbf{A}$, where $\mathbf{R}$ is an upper triangular matrix.

With input as an orthogonal matrix $\mathbf{V}$, the Householder QR decomposition can be used to decompose $\mathbf{V}$ into a product of Householder matrices. This means that any orthogonal matrix $\mathbf{V}$ can be expressed as

$$\mathbf{V} = \mathbf{H}_1 \mathbf{H}_2 \cdots \mathbf{H}_n \mathbf{R},$$

for an orthogonal diagonal matrix $\mathbf{R}$ (i.e., $\mathbf{R}$ has diagonal entries $\pm 1$).

However, since the Householder QR algorithm operates column-wise, it typically returns $n$ Householder factors, even when the input matrix $\mathbf{V} \in \mathcal{H}_m$ has much fewer ($m < n$) Householder factors.

Consider an example with the input matrix $\mathbf{V} \in \mathbb{R}^{3 \times 3}$. In particular, $\mathbf{V} \in \mathcal{H}_1$ is a Householder reflector. The Householder QR decomposition decomposes $\mathbf{V}$ into a product of 3 Householder matrices and an additional upper triangular matrix as follows:

$$\mathbf{V} = \mathbf{I} - 2\mathbf{u}\mathbf{u}^\top, \quad \text{with } \mathbf{u} = [2/3, 1/3, 2/3]^\top.$$

The QR decomposition gives

$$\mathbf{V} = \mathbf{H}_1\mathbf{H}_2\mathbf{H}_3\mathbf{R},$$

where $\mathbf{H}_1, \mathbf{H}_2, \mathbf{H}_3$ are Householder matrices and $\mathbf{R}$ is a diagonal matrix with $-1, -1, 1$ on the diagonal (from top to bottom).

$$\mathbf{V} = \begin{bmatrix} 1/9 & -4/9 & -8/9 \\ -4/9 & 7/9 & -4/9 \\ -8/9 & -4/9 & 1/9 \end{bmatrix} = \mathbf{H}_1\mathbf{H}_2\mathbf{H}_3\mathbf{R}, \quad \text{where}$$

$$\mathbf{H}_1 = \begin{bmatrix} -1/9 & 4/9 & 8/9 \\ 4/9 & 37/45 & -16/45 \\ 8/9 & -16/45 & 13/45 \end{bmatrix},$$

$$\mathbf{H}_2 = \begin{bmatrix} 1 & 0 & 0 \\ 0 & -3/5 & 4/5 \\ 0 & 4/5 & 3/5 \end{bmatrix},$$

$$\mathbf{H}_3 = \begin{bmatrix} 1 & 0 & 0 \\ 0 & 1 & 0 \\ 0 & 0 & -1 \end{bmatrix},$$

$$\mathbf{R} = \begin{bmatrix} -1 & 0 & 0 \\ 0 & -1 & 0 \\ 0 & 0 & 1 \end{bmatrix}.$$

This decomposition is not optimal, but it is obtained using the QR decomposition algorithm. As $\mathbf{V} \in \mathcal{H}_1$, the optimal decomposition is just

$$\mathbf{V} = \mathbf{HI},$$

where matrix $\mathbf{R}$ in the QR decomposition is identity, and $\mathbf{V}$ is already a Householder matrix corresponding to Householder vector

$$\mathbf{u} = [2/3, 1/3, 2/3]^\top.$$

### A.2 Representations of products of Householder matrices

To illustrate why using Householder matrices is useful in neural network parametrization as well as in signal processing, we present some standard ways to store and represent products of Householder matrices below.

**Direct expansion (in terms of Householder vectors)**   Note that the expression for an arbitrary $\mathbf{V}$ in $\mathcal{H}_m$ is of the form:

$$\mathbf{V} = \mathbf{I} - 2\sum_{i=1}\mathbf{u}_i\mathbf{u}_i^\mathsf{T} + 4\sum_{\substack{i_1,i_2 \\ i_1<i_2}} k_{i_1 i_2}\mathbf{u}_{i_1}\mathbf{u}_{i_2}^\mathsf{T} - \cdots$$

$$+ (-2)^m \sum_{i_1<i_2<\cdots<i_m} \left(\prod_{r=1}^{m-1} k_{i_r i_{r+1}}\right)\mathbf{u}_{i_1}\mathbf{u}_{i_m}^\mathsf{T} \tag{8}$$

where $\mathbf{u}_1, \mathbf{u}_2, \ldots$ are unit norm vectors and $k_{ij} = \mathbf{u}_i^\mathsf{T}\mathbf{u}_j$ are the pairwise inner products.

**Basis kernel representation**   Sun & Bischof (1995): Let $\mathbf{V} \in \mathbb{R}^{n \times n}$ be an orthogonal matrix. The *basis-kernel representation* expresses $\mathbf{V}$ in the form

$$\mathbf{V} = \mathbf{I} - \mathbf{Y}\mathbf{S}\mathbf{Y}^\mathsf{T},$$

where:

- $\mathbf{Y} \in \mathbb{R}^{n \times r}$ contains basis vectors for a subspace of $\mathbb{R}^n$,
- $\mathbf{S} \in \mathbb{R}^{r \times r}$ is a symmetric matrix, and
- $\mathbf{I}$ is the $n \times n$ identity matrix.

This representation is compact and efficient when $r \ll n$, and it generalizes Householder transformations. It is related to the WY representation.

**WY representation**   Bischof & Van Loan (1987): Given a product of $m$ Householder reflectors,

$$\mathbf{V} = \mathbf{H}_1 \mathbf{H}_2 \cdots \mathbf{H}_m, \tag{9}$$

where each $\mathbf{H}_i = \mathbf{I} - 2\mathbf{u}_i \mathbf{u}_i^\mathsf{T}$ is a Householder matrix, the WY representation expresses $\mathbf{V}$ in a compact form as:

$$\mathbf{V} = \mathbf{I} - \mathbf{W}\mathbf{Y}^\mathsf{T}, \tag{10}$$

where:

- $\mathbf{Y} = [\mathbf{u}_1, \ldots, \mathbf{u}_m] \in \mathbb{R}^{n \times m}$ collects the Householder vectors,
- $\mathbf{W} \in \mathbb{R}^{n \times m}$ is computed such that the product $\mathbf{W}\mathbf{Y}^\mathsf{T}$ reproduces the original sequence of Householder transformations.

Note that there also exists a more storage-efficient version of the same Schreiber & Van Loan (1989).

### A.3   RECOVERING SYMMETRIC ORTHOGONAL MATRICES

If $\mathbf{V}$ is known to be symmetric, we can find the decomposition easily: to see this, note that a symmetric orthogonal $\mathbf{V}$ has a full set of orthogonal eigenvectors with eigenvalues $\pm 1$, and so has a decomposition of the form:

$$\mathbf{V} = \underbrace{\mathbf{v}_1 \mathbf{v}_1^\mathsf{T} + \mathbf{v}_2 \mathbf{v}_2^\mathsf{T} + \ldots + \mathbf{v}_k \mathbf{v}_k^\mathsf{T}}_{\geq n-m \text{ terms}} - \mathbf{v}_{k+1} \mathbf{v}_{k+1}^\mathsf{T} - \ldots - \mathbf{v}_n \mathbf{v}_n^\mathsf{T}.$$

Using $\sum \mathbf{v}_i \mathbf{v}_i^\mathsf{T} = \mathbf{I}$, we may rewrite the above as

$$\mathbf{V} = \mathbf{I} - 2 \sum_{i=k+1}^{n} \mathbf{v}_i \mathbf{v}_i^\mathsf{T} = \prod_{i=k+1}^{n} \left( \mathbf{I} - 2\mathbf{v}_i \mathbf{v}_i^\mathsf{T} \right), \tag{11}$$

where the last equality from sum to product follows from the orthogonality of the eigenvectors $\mathbf{v}_i$.

Thus, the above gives a straightforward way to find the Householder decomposition for symmetric $\mathbf{V} \in \mathcal{H}_m$: we simply find the eigenvectors $\mathbf{v}_{k+1}, \mathbf{v}_{k+2}, \ldots, \mathbf{v}_n$ corresponding to eigenvalue $-1$, and generate the Householder factors $\mathbf{H}_i$ as

$$\mathbf{H}_i = \mathbf{I} - 2\mathbf{v}_i \mathbf{v}_i^\mathsf{T}.$$

## B PROOFS

### B.1 LEMMA 1

*Proof.* Most of these results follow from standard linear algebra techniques, but we include them here for completeness.

1. A normal matrix is a complex square matrix that commutes with its conjugate transpose (i.e., $\mathbf{A}\mathbf{A}^* = \mathbf{A}^*\mathbf{A}$) and has the spectral decomposition specified in Lemma 1 Horn & Johnson (2012). Since all real orthogonal matrices $\mathbf{V}$ satisfy $\mathbf{V}\mathbf{V}^* = \mathbf{V}^*\mathbf{V}$, the first point in Lemma 1 follows.

2. Let $\mathbf{V}\mathbf{w} = \lambda\mathbf{w}$. Applying the conjugate operation, and using $\mathbf{V} = \overline{\mathbf{V}}$, we get
$$\mathbf{V}\overline{\mathbf{w}} = \overline{\lambda}\,\overline{\mathbf{w}}.$$

3. Let $\mathbf{V}\mathbf{w} = \lambda\mathbf{w}$. Then,
$$\mathbf{V}\overline{\mathbf{w}} = \overline{\lambda}\,\overline{\mathbf{w}},$$
$$\mathbf{V}^\mathsf{T}\mathbf{w} = \overline{\lambda}\mathbf{w}$$
$$\left(\mathbf{V}_{\mathrm{sym}}\right)\overline{\mathbf{w}} = \left(\frac{\lambda + \overline{\lambda}}{2}\right)\overline{\mathbf{w}}$$

Thus,
$$\mathbf{V}_{\mathrm{sym}}\overline{\mathbf{w}} = \mathrm{Re}(\lambda)\overline{\mathbf{w}}$$
$$\mathbf{V}_{\mathrm{sym}}\mathbf{w} = \mathrm{Re}(\lambda)\mathbf{w}$$
$$\mathbf{V}_{\mathrm{sym}}(\mathbf{w} + \overline{\mathbf{w}}) = \mathrm{Re}(\lambda)(\mathbf{w} + \overline{\mathbf{w}})$$
Instead of adding the above, had we subtracted and divided by $i$, we would have the other eigenvector mentioned in the Lemma, corresponding to the same eigenvalue. The final part follows arguments similar to those above.

4. Let $\mathbf{V}\mathbf{w} = \lambda\mathbf{w}$. Then,
$$\mathbf{V}^\mathsf{T}\mathbf{V}\mathbf{w} = \mathbf{V}^\mathsf{T}\lambda\mathbf{w}$$
$$\mathbf{w} = \mathbf{V}^\mathsf{T}\lambda\mathbf{w},$$
$$\mathbf{w}\frac{\overline{\lambda}}{|\lambda|^2} = \mathbf{V}^\mathsf{T}\mathbf{w}$$

Since $|\lambda|^2 = 1$ for eigenvalues of orthogonal matrices, $\overline{\lambda}$ is an eigenvalue of $\mathbf{V}^\mathsf{T}$ corresponding to the eigenvector $\mathbf{w}$.

If $\lambda = \pm 1$, then $\mathbf{V}\mathbf{w} = \pm\mathbf{w}$ and
$$\mathbf{V}^\mathsf{T}\mathbf{V}\mathbf{w} = \pm\mathbf{V}^\mathsf{T}\mathbf{w},$$
$$\pm\mathbf{V}^\mathsf{T}\mathbf{w} = \mathbf{w},$$
$$\left(\mathbf{V}_{\mathrm{sym}}\right)\mathbf{w} = \pm\mathbf{w}$$

$\square$

### B.2 LEMMA 2

*Proof.* Note that
$$\mathbf{V}_{k+1}\mathbf{w} = \hat{\mathbf{H}}_k\mathbf{V}_k\mathbf{w}$$
$$= \mathbf{V}\mathbf{w} - 2\hat{\mathbf{u}}\hat{\mathbf{u}}^\mathsf{T}\mathbf{V}\mathbf{w}.$$

Since $\hat{\mathbf{u}}$ is in the span of $\mathbf{z}, \overline{\mathbf{z}}$, it follows that $\hat{\mathbf{u}}^\mathsf{T}\mathbf{w} = 0$ (since $\mathbf{z}, \overline{\mathbf{z}}$ are both orthogonal to $\mathbf{w}$ on $\mathbb{C}^n$). So we get
$$\mathbf{V}_{k+1}\mathbf{w} = \mathbf{V}_k\mathbf{w},$$
completing the first part. The second part follows from a similar calculation. $\square$

### B.3   LEMMA 3

*Proof.* For the trace, we see

$$\text{tr}(\mathbf{V}_{k+1}) = \text{tr}(\hat{\mathbf{H}}_k \mathbf{V}_k)$$
$$= \text{tr}(\mathbf{V}_k) - 2\text{tr}(\hat{\mathbf{u}}^\mathsf{T} \mathbf{V}_k \hat{\mathbf{u}}),$$

giving the required expression.

For the eigenspace, suppose that the input $\mathbf{V} \in \mathcal{H}_p$ is a product of $p$ Householder matrices for $p \leq n$. Recall that every $n \times n$ orthogonal matrix is a product of at most $n$ Householder matrices Uhlig (2001). Here, we distinguish the following two cases:

1. *Case 1: $p + k$ is odd:* In this case,

$$\mathbf{V}_k = \hat{\mathbf{H}}_k \hat{\mathbf{H}}_{k-1} \ldots \hat{\mathbf{H}}_1 \mathbf{V}$$

   is a product of $p + k$ Householder matrices, and so

$$\det(\mathbf{V}_k) = (-1)^{p+k} = -1.$$

   It follows that $-1$ is an eigenvalue of $\mathbf{V}_k$ and $(\mathbf{V}_k)_{\text{sym}}$. Thus, by the construction of the algorithm, the vector $\mathbf{u}$ picked at step $k$ satisfies

$$(\mathbf{V}_k)_{\text{sym}}\mathbf{u} = \mathbf{V}_k\mathbf{u} = -\mathbf{u}.$$

   From Lemma 2, it follows that

$$\mathcal{E}^1_{\mathbf{V}_{k+1}} = \mathcal{E}^1_{\mathbf{V}_k} + \cup_\alpha \{\alpha u\}, \tag{12}$$

   and the Lemma follows.

2. *Case 2: $p + k$ is even:* If

$$\lambda_{\min} = \lambda_{\min}((\mathbf{V}_k)_{\text{sym}}) = -1,$$

   we make a similar argument to the above case. So suppose $\lambda_{\min} > -1$. Note that $\mathbf{V}_{k+1}$ is a product of $p + k + 1$ Householder matrices and hence has an eigenvalue of $-1$ with odd multiplicity. For complex unit $\lambda$, let

$$k_\lambda = \text{mult}(\mathbf{V}_{k+1}, \lambda) - \text{mult}(\mathbf{V}_k, \lambda).$$

   Note that by definition

$$\sum k_\lambda = 0, \tag{13}$$

   where the sum is over eigenvalues of either $\mathbf{V}_k$ or $\mathbf{V}_{k+1}$. From Lemma 2, since all eigenvectors other than $\mathbf{u}$ and $\bar{\mathbf{u}}$ are retained,

$$|k_\lambda| \leq 2. \tag{14}$$

   Also, for any $\lambda$ other than those associated with $\mathbf{u}, \bar{\mathbf{u}}$, the multiplicity cannot decrease, so these $k_\lambda$ are non-negative; in particular

$$k_1 \geq 0 \quad \text{since } \lambda_{\min} \neq 1.$$

   As observed previously, $k_{-1}$ is odd (positive). Furthermore, because $\mathbf{V}_{k+1}$ and $\mathbf{V}_k$ are real orthogonal matrices, the complex eigenvalues occur in conjugate pairs, and so

$$k_\lambda = k_{\bar{\lambda}}.$$

   Hence, the only way these conditions hold is with

$$k_{-1} = 1 \quad \text{and} \quad k_1 = 1, \tag{15}$$

   which means the eigenspace for eigenvalue 1 increases in dimension by 1.

$\square$

### B.3.1 ON THE COUNTING ARGUMENT IN LEMMA 3

We present a slightly more detailed version of the proof of Lemma 3 below.

*Multiplicity change constraint:* Let $\mathbf{V}_k \in \mathbb{R}^{n \times n}$ be orthogonal, with eigenvalues on the unit circle, and let $\mathbf{V}_{k+1} = \mathbf{H}_k \mathbf{V}_k$ where $\mathbf{H}_k$ is a Householder reflector constructed from an eigenvector corresponding to $\lambda_{\min}((\mathbf{V}_k)_{\text{sym}})$. Let $m_\lambda^{(k)}$ denote the multiplicity of eigenvalue $\lambda$ in $\mathbf{V}_k$, and define the multiplicity change

$$k_\lambda := m_\lambda^{(k+1)} - m_\lambda^{(k)}.$$

Then, if $\lambda_{\min}((\mathbf{V}_k)_{\text{sym}}) > -1$, we must have

$$k_{-1} = 1, \quad k_1 = 1, \quad k_{\lambda_{\min}} = -2,$$

and $k_\lambda = 0$ for all other $\lambda$.

*Proof. Conservation of multiplicities.* The total multiplicity across all eigenvalues is preserved:

$$\sum_\lambda m_\lambda^{(k)} = \sum_\lambda m_\lambda^{(k+1)} = n, \tag{16}$$

which gives

$$\sum_\lambda k_\lambda = 0.$$

*Support of multiplicity change.* By Lemma 2, the Householder update $\mathbf{V}_{k+1} = \mathbf{H}_k \mathbf{V}_k$ changes the spectrum only in the two-dimensional invariant subspace spanned by the chosen eigenvector $\mathbf{u}$ and its complex conjugate $\bar{\mathbf{u}}$. Therefore, $k_\lambda = 0$ for all $\lambda$ except possibly $\{-1, 1, \lambda_{\min}, \bar{\lambda}_{\min}\}$. The magnitude of change satisfies

$$|k_\lambda| \leq 2 \quad \text{for all } \lambda,$$

since at most two eigenvalues are altered in a single iteration.

*Case distinction on $\lambda_{\min}$.* If $\lambda_{\min} = -1$, we fall into case 1 of the main proof, and the lemma is trivial. We thus assume $\lambda_{\min} > -1$.

*Parity of $k_{-1}$.* By Lemma 1, for any orthogonal (or unitary) matrix, the multiplicity of eigenvalue $-1$ changes by an *odd* integer when the update vector is not in the $-1$ eigenspace. Since $\lambda_{\min} > -1$, the chosen eigenvector is not in the $-1$ eigenspace; hence $k_{-1}$ is odd and positive. From above, the only odd positive value allowed is

$$k_{-1} = 1.$$

*Structure of $\lambda_{\min}$.* Because $\lambda_{\min} > -1$ and $\lambda_{\min} \neq 1$, it must have nonzero imaginary part: $\text{Im}(\lambda_{\min}) \neq 0$. Hence $\lambda_{\min}$ and $\bar{\lambda}_{\min}$ occur as a conjugate pair in the spectrum of $\mathbf{V}_k$, each with the same multiplicity. These two eigenvalues together contribute multiplicity change

$$k_{\lambda_{\min}} = k_{\bar{\lambda}_{\min}}. \tag{17}$$

*Elimination of possibilities for $k_{\lambda_{\min}}$.* Since the chosen eigenvector corresponds to $\lambda_{\min}$, after applying the Householder, this pair is *removed* from the spectrum of $\mathbf{V}_{k+1}$; hence, the multiplicity of each decreases by 1. Thus,

$$k_{\lambda_{\min}} = k_{\bar{\lambda}_{\min}} = -1,$$

so the total multiplicity change for the pair is

$$k_{\lambda_{\min}} + k_{\bar{\lambda}_{\min}} = -2.$$

*Solving for $k_1$.* The sum of all multiplicity changes gives

$$k_{-1} + k_1 + (k_{\lambda_{\min}} + k_{\bar{\lambda}_{\min}}) = 0$$
$$1 + k_1 - 2 = 0 \quad \Rightarrow \quad k_1 = 1.$$

Thus, the only possible multiplicity change pattern under the stated assumptions is

$$k_{-1} = 1, \quad k_1 = 1, \quad k_{\lambda_{\min}} = k_{\bar{\lambda}_{\min}} = -1,$$

with all others zero. This matches the statement of the lemma. $\square$

### B.4 THEOREM 2

Consider an arbitrary approximation $\hat{\mathbf{V}} \in \mathcal{H}_k$ to some ground truth orthogonal matrix $\mathbf{V} \in \mathcal{H}_m$. Let the best choice of $\mathbf{H}$ to update $\hat{\mathbf{V}}$ be $\hat{\mathbf{H}}$, i.e., $\hat{\mathbf{V}} \leftarrow \hat{\mathbf{V}}\hat{\mathbf{H}}$. We analyze the improvement in reconstruction error owing to this choice of $\mathbf{H}$. Consider

$$\hat{\mathbf{H}} = \arg\min_{\mathbf{H}} \delta \quad \text{where,}$$

$$
\begin{aligned}
\delta &= \|\mathbf{V} - \hat{\mathbf{V}}\mathbf{H}\|_F^2 - \|\mathbf{V} - \hat{\mathbf{V}}\|_F^2 \\
&= \|\mathbf{V}\|_F^2 + \|\hat{\mathbf{V}}\mathbf{H}\|_F^2 - 2\mathrm{tr}((\hat{\mathbf{V}}\mathbf{H})^\top \mathbf{V}) - (\|\mathbf{V}\|_F^2 + \|\hat{\mathbf{V}}\|_F^2 - 2\mathrm{tr}(\hat{\mathbf{V}}^\top \mathbf{V})) \\
&= 2\mathrm{tr}(\hat{\mathbf{V}}^\top \mathbf{V} - (\hat{\mathbf{V}}\mathbf{H})^\top \mathbf{V}) \\
&= 2\mathrm{tr}(\hat{\mathbf{V}}^\top \mathbf{V} - (\mathbf{I} - 2\mathbf{u}\mathbf{u}^\top)\hat{\mathbf{V}}^\top \mathbf{V}) \\
&= 4\mathrm{tr}(\mathbf{u}\mathbf{u}^\top \hat{\mathbf{V}}^\top \mathbf{V}) \\
&= 4\mathbf{u}^\top (\hat{\mathbf{V}}^\top \mathbf{V})\mathbf{u}
\end{aligned}
\tag{18}
$$

Note that we required $\arg\min$ since we want the decrease in error to be as negative as possible. The solution to the problem above is choosing $\mathbf{u} = \hat{\mathbf{u}}$ as the eigenvector corresponding to the most negative eigenvalue of the symmetric part of the matrix $(\hat{\mathbf{V}}^\top \mathbf{V})$. **This is precisely our update rule**. Therefore, given some fixed approximation $\hat{\mathbf{V}} \in \mathcal{H}_k$, our update rule leads to best projection in $\mathcal{H}_{k+1}$.

We now use induction to prove the optimality of Algorithm 1. Base case: this follows from equation 2. Had $\hat{\mathbf{V}} \in \mathcal{H}_k$ been optimal, Algorithm 1 would recover the optimal $\hat{\mathbf{V}} \in \mathcal{H}_{k+1}$- this follows from 18. Therefore, the recovery obtained from Algorithm 1 is optimal.

### B.5 ADDITIONAL JUSTIFICATION FOR THE EIGENVALUE INDEXING IN THEOREM 3

Let $\lambda_1 \le \lambda_2 \le \cdots \le \lambda_n$ denote the eigenvalues of $\mathbf{V}_{\text{sym}}$ in ascending order, listed with multiplicity. By construction of Algorithm 1 and Lemma 3, at each *even* iteration index $k$, the only eigenvalues of $(\mathbf{V}_k)_{\text{sym}}$ that differ from $(\mathbf{V}_{k-2})_{\text{sym}}$ are those in the two-dimensional invariant subspace spanned by the eigenvector chosen at iteration $k-2$ (corresponding to $\lambda_{\min}((\mathbf{V}_{k-2})_{\text{sym}})$) and its conjugate, together with possible changes in the $\pm 1$ eigenspaces.

We prove the following claim by induction on $m$: for every integer $m \ge 0$,

$$\lambda_{\min}\big((\mathbf{V}_{2m})_{\text{sym}}\big) = \lambda_{m+1}.$$

Equivalently, for even $k = 2m$:

$$\lambda_{\min}((\mathbf{V}_k)_{\text{sym}}) = \lambda_{k/2+1}.$$

*Base case ($m = 0$).* For $k = 0$, we have $(\mathbf{V}_0)_{\text{sym}} = \mathbf{V}_{\text{sym}}$, hence

$$\lambda_{\min}((\mathbf{V}_0)_{\text{sym}}) = \lambda_1,$$

so the claim holds for $m = 0$.

*Inductive step.* Fix $m \ge 0$ and suppose the claim holds for $m$. Put $k = 2m$, so by the inductive hypothesis:

$$\lambda_{\min}\big((\mathbf{V}_k)_{\text{sym}}\big) = \lambda_{m+1}.$$

By Lemma 2 and the two-dimensional support of the update, the single Householder update $\mathbf{V}_k \mapsto \mathbf{V}_{k+1}$ can only:

1. introduce eigenvalue $-1$ with odd multiplicity (making it the new minimum at odd indices), and

2. act nontrivially only on the two-dimensional invariant subspace spanned by the chosen eigenvector (and its conjugate).

Hence at $k + 1$:

$$\lambda_{\min}((\mathbf{V}_{k+1})_{\text{sym}}) = -1.$$

Applying the next update $k + 1 \mapsto k + 2$ removes all introduced $-1$ entries and, by the two-dimensional support, removes the eigenvalue(s) corresponding to the chosen $\lambda_{m+1}$ (and its conjugate if complex) from the spectrum while leaving all eigenvalues strictly below the next original eigenvalue $\lambda_{m+2}$ unchanged. Therefore:

$$\lambda_{\min}\big((\mathbf{V}_{k+2})_{\text{sym}}\big) = \lambda_{m+2},$$

which is precisely the claim for $m + 1$. This completes the induction.

*Remark on multiplicities:* If some eigenvalues of $\mathbf{V}_{\text{sym}}$ have multiplicity $> 1$, the indexing $\lambda_1 \leq \cdots \leq \lambda_n$ is understood in the non-decreasing sense, counting each repeated eigenvalue separately. In this case, the removal step at each even $k$ deletes one instance of $\lambda_{k/2}$ (and its conjugate if complex) from the multiset of eigenvalues, so the induction and indexing statements remain valid.

### B.6    ON THE TRUE BOUND IN THEOREM 3

The proof of Theorem 3 actually results in two separate bounds for separate cases, which we have collapsed into a single one applicable for both. Hence, there is a slight gap in the bound and the final error in Figure 1. To be more precise, the true bounds (which would give 0 error at $m$ steps) are as follows:

Consider two cases: $\mathbf{V}$ being a product of an even number of Householder matrices, and $\mathbf{V}$ being a product of an odd number of Householder matrices. The error after Algorithm 1 truncated to $m$ steps is given by:

$$\|\mathbf{V} - \hat{\mathbf{V}}\|_F = \sqrt{2\left(n - \text{tr}(\mathbf{V}) - 2\lfloor m/2 \rfloor - 2 \sum_{i=1}^{\lceil m/2 \rceil} |\lambda_i|\right)}, \qquad (19)$$

and for the second (odd) case:

$$\|\mathbf{V} - \hat{\mathbf{V}}\|_F = \sqrt{2\left(n - \text{tr}(\mathbf{V}) - 2\lceil m/2 \rceil - 2 \sum_{i=2}^{\lfloor m/2 \rfloor} |\lambda_i|\right)}, \qquad (20)$$

where $\lambda_i$'s are the set of the $\lceil m/2 \rceil$ or $\lfloor m/2 \rfloor$ unique most negative eigenvalues.

#### B.6.1    ILLUSTRATION OF RECOVERY FOR $\mathbf{V} \in \mathcal{H}_2$

We illustrate the **recovery of an arbitrary orthogonal matrix $\mathbf{V} \in \mathcal{H}_2$**; that is, we seek matrices $\mathbf{H}_3, \mathbf{H}_4$ such that

$$\mathbf{V} = \mathbf{H}_1\mathbf{H}_2 = \mathbf{H}_3\mathbf{H}_4.$$

Let

$$\mathbf{V} = (\mathbf{I} - 2\mathbf{u}_1\mathbf{u}_1^\top)(\mathbf{I} - 2\mathbf{u}_2\mathbf{u}_2^\top)$$
$$= \mathbf{I} - 2\mathbf{u}_1\mathbf{u}_1^\top - 2\mathbf{u}_2\mathbf{u}_2^\top + 4(\mathbf{u}_1^\top\mathbf{u}_2)\mathbf{u}_1\mathbf{u}_2^\top. \qquad (21)$$

For notational simplicity, define $k = \mathbf{u}_1^\top\mathbf{u}_2$. Then the symmetric part is

$$\mathbf{V}_{\text{sym}} = \mathbf{I} - 2\mathbf{u}_1\mathbf{u}_1^\top - 2\mathbf{u}_2\mathbf{u}_2^\top + 2k(\mathbf{u}_1\mathbf{u}_2^\top + \mathbf{u}_2\mathbf{u}_1^\top) \qquad (22)$$
$$\mathbf{V}_{\text{sym}}\mathbf{u}_1 = \big(\mathbf{I} - 2\mathbf{u}_1\mathbf{u}_1^\top - 2\mathbf{u}_2\mathbf{u}_2^\top + 2k(\mathbf{u}_1\mathbf{u}_2^\top + \mathbf{u}_2\mathbf{u}_1^\top)\big)\mathbf{u}_1$$
$$= -\mathbf{u}_1 + 2k^2\mathbf{u}_1,$$
$$\mathbf{V}_{\text{sym}}\mathbf{u}_2 = -\mathbf{u}_2 + 2k^2\mathbf{u}_2.$$

Thus, $\mathbf{u}_1$ and $\mathbf{u}_2$ are eigenvectors with the same eigenvalue $-1 + 2k^2$. Any vector in the span of $\mathbf{u}_1$ and $\mathbf{u}_2$, i.e., $\alpha \mathbf{u}_1 + \beta \mathbf{u}_2$, is also an eigenvector. Imposing the unit norm constraint gives

$$\|\alpha \mathbf{u}_1 + \beta \mathbf{u}_2\|^2 = \alpha^2 + \beta^2 + 2\alpha\beta k = 1. \tag{23}$$

Consider any vector $\mathbf{w}$ orthogonal to both $\mathbf{u}_1$ and $\mathbf{u}_2$. Then

$$\mathbf{V}_{\text{sym}} \mathbf{w} = \mathbf{w},$$

so all such vectors have eigenvalue 1. Hence, the minimum eigenvalue of $\mathbf{V}_{\text{sym}}$ is

$$\lambda_{\min} = -1 + 2k^2, \tag{24}$$

with $-1 \le k \le 1$.

Set

$$\begin{aligned}
\mathbf{H}_3 &= \mathbf{I} - 2(\alpha \mathbf{u}_1 + \beta \mathbf{u}_2)(\alpha \mathbf{u}_1 + \beta \mathbf{u}_2)^{\mathsf{T}} \\
&= \mathbf{I} - 2\left( \alpha^2 \mathbf{u}_1 \mathbf{u}_1^{\mathsf{T}} + \alpha\beta(\mathbf{u}_1 \mathbf{u}_2^{\mathsf{T}} + \mathbf{u}_2 \mathbf{u}_1^{\mathsf{T}}) + \beta^2 \mathbf{u}_2 \mathbf{u}_2^{\mathsf{T}} \right)
\end{aligned} \tag{25}$$

Now,

$$\begin{aligned}
\mathbf{H}_3^{-1} \mathbf{V} &= \mathbf{H}_3 \mathbf{H}_1 \mathbf{H}_2 \\
&= \mathbf{I} - 2\mathbf{u}_1 \mathbf{u}_1^{\mathsf{T}}(1 - \alpha^2 - 2\alpha\beta k) \\
&\quad - 2\mathbf{u}_2 \mathbf{u}_2^{\mathsf{T}}(1 - \beta^2 + 2\alpha\beta k + 4\beta^2 k^2) \\
&\quad - 2\mathbf{u}_1 \mathbf{u}_2^{\mathsf{T}}(-2k + 2k\alpha^2 - \alpha\beta + 4\alpha\beta k^2) \\
&\quad - 2\mathbf{u}_2 \mathbf{u}_1^{\mathsf{T}}(-\alpha\beta - 2\beta^2 k)
\end{aligned} \tag{26}$$

Note that

$$-2k + 2k\alpha^2 - \alpha\beta + 4\alpha\beta k^2 = -\alpha\beta - 2\beta^2 k, \tag{27}$$

by using the unit norm condition. Thus, the coefficients of both $2\mathbf{u}_1 \mathbf{u}_2^{\mathsf{T}}$ and $2\mathbf{u}_2 \mathbf{u}_1^{\mathsf{T}}$ are equal to $-\alpha\beta - 2\beta^2 k$.

Finally, consider the vector

$$\mathbf{v} = -\beta \mathbf{u}_1 + (\alpha + 2\beta k)\mathbf{u}_2.$$

Define

$$\begin{aligned}
\mathbf{H}_4 &= \mathbf{I} - 2\mathbf{v}\mathbf{v}^{\mathsf{T}} \\
&= \mathbf{I} - 2\left( -\beta \mathbf{u}_1 + (\alpha + 2\beta k)\mathbf{u}_2 \right)\left( -\beta \mathbf{u}_1 + (\alpha + 2\beta k)\mathbf{u}_2 \right)^{\mathsf{T}}.
\end{aligned}$$

It can be verified that

$$\mathbf{H}_3 \mathbf{H}_1 \mathbf{H}_2 = \mathbf{H}_4.$$

All that is left is to analyze whether the vector $-\beta \mathbf{u}_1 + (\alpha + 2\beta k)\mathbf{u}_2$ is an eigenvector of

$$\left( \mathbf{H}_3 \mathbf{H}_1 \mathbf{H}_2 \right)_{\text{sym}} = \mathbf{I} - 2\mathbf{u}_1 \mathbf{u}_1^{\mathsf{T}} - 2\mathbf{u}_2 \mathbf{u}_2^{\mathsf{T}} - 2(\alpha\beta + 2\beta^2 k)\left( \mathbf{u}_1 \mathbf{u}_2^{\mathsf{T}} + \mathbf{u}_2 \mathbf{u}_1^{\mathsf{T}} \right). \tag{28}$$

Computing:

$$\begin{aligned}
&\left( \mathbf{H}_3 \mathbf{H}_1 \mathbf{H}_2 \right)_{\text{sym}} \left( -\beta \mathbf{u}_1 + (\alpha + 2\beta k)\mathbf{u}_2 \right) \\
&= -\beta \mathbf{u}_1 + 2\beta \mathbf{u}_1(1 - \alpha^2 - 2\alpha\beta k) + 2\beta k \mathbf{u}_2(1 - \beta^2 + 2\alpha\beta k + 4\beta^2 k^2) \\
&\quad - 2\beta \mathbf{u}_2(\alpha\beta + 2\beta^2 k) - 2\beta k \mathbf{u}_1(\alpha\beta + 2\beta^2 k) \\
&\quad + \mathbf{u}_2(\alpha + 2\beta k) - 2k \mathbf{u}_1(\alpha + 2\beta k)(1 - \alpha^2 - 2\alpha\beta k) \\
&\quad - 2\mathbf{u}_2(\alpha + 2\beta k)(1 - \beta^2 + 2\alpha\beta k + 4\beta^2 k^2) \\
&\quad + 2k \mathbf{u}_2(\alpha + 2\beta k)(\alpha\beta + 2\beta^2 k) + 2\mathbf{u}_1(\alpha + 2\beta k)(\alpha\beta + 2\beta^2 k)
\end{aligned} \tag{29}$$

Clubbing the coefficients of $\mathbf{u}_1$ and $\mathbf{u}_2$, we get:

$$
\begin{aligned}
\mathbf{u}_1\big( &-\beta + 2\beta(1 - \alpha^2 - 2\alpha\beta k) - 2\beta k(\alpha\beta + 2\beta^2 k) \\
&- 2k(\alpha + 2\beta k)(1 - \alpha^2 - 2\alpha\beta k) + 2(\alpha + 2\beta k)(\alpha\beta + 2\beta^2 k)\big) \\
+\mathbf{u}_2\big( &2\beta k(1 - \beta^2 + 2\alpha\beta k + 4\beta^2 k^2) - 2\beta(\alpha\beta + 2\beta^2 k) + (\alpha + 2\beta k) \\
&- 2(\alpha + 2\beta k)(1 - \beta^2 + 2\alpha\beta k + 4\beta^2 k^2) + 2k(\alpha + 2\beta k)(\alpha\beta + 2\beta^2 k)\big)
\end{aligned}
\tag{30}
$$

This simplifies to:

$$
\begin{aligned}
= \mathbf{u}_1\Big( &-\beta + 2\beta^3 + 4\alpha\beta^2 k + 2\alpha^2\beta\Big) \\
&+ \mathbf{u}_2(\alpha + 2\beta k)\Big(4\beta k(\alpha + 2\beta k) - 2\beta^2 + 1 - 2(\alpha + 2\beta k)^2\Big)
\end{aligned}
$$

Combining terms using the unit norm constraint, the result is:

$$
\begin{aligned}
&= -\beta\mathbf{u}_1\big(1 - 2\beta^2 - 2\alpha^2 - 4\alpha\beta k\big) + (\alpha + 2\beta k)\mathbf{u}_2\big(1 - 2\beta^2 - 2\alpha^2 - 4\alpha\beta k\big) \\
&= -\big(-\beta\mathbf{u}_1 + (\alpha + 2\beta k)\mathbf{u}_2\big)
\end{aligned}
\tag{31}
$$

(as $1 - 2\beta^2 - 2\alpha^2 - 4\alpha\beta k = -1$ by the unit norm condition).

Therefore, the vector $\mathbf{v} = -\beta\mathbf{u}_1 + (\alpha + 2\beta k)\mathbf{u}_2$ is an eigenvector of $(\mathbf{H}_3\mathbf{H}_1\mathbf{H}_2)_{\text{sym}}$ with eigenvalue $-1$, which is the minimum eigenvalue. Moreover, the eigenvector corresponding to this eigenvalue, which turns out to be the Householder vector for $\mathbf{H}_4$, is unique up to sign.

Note the following: $\mathbf{V}_{\text{sym}}$ had the eigenvalue $(-1 + 2k^2)$ with multiplicity 2 and the eigenvalue 1 with multiplicity $n - 2$. If we consider the symmetric part of $\mathbf{H}_3\mathbf{H}_1\mathbf{H}_2$, it turns out to be equal to $\mathbf{H}_3\mathbf{H}_1\mathbf{H}_2$. Once again, any vector orthogonal to both $\mathbf{u}_1$ and $\mathbf{u}_2$ is still an eigenvector with eigenvalue 1.

However, unlike the symmetric part of $\mathbf{H}_1\mathbf{H}_2$, whose minimum eigenvalue had multiplicity 2, the minimum eigenvalue in this case $(-1)$, which corresponds to the specific eigenvector $-\beta\mathbf{u}_1 + (\alpha + 2\beta k)\mathbf{u}_2$, has multiplicity 1. Since the matrix under consideration is symmetric, the space spanned by the eigenvectors has dimension $n$.

Thus, there exists another eigenvector in the plane spanned by $\mathbf{u}_1$ and $\mathbf{u}_2$, orthogonal to the above Householder vector. It can be further verified that this eigenvector has eigenvalue 1. Therefore, there are infinitely many solutions to the equation $\mathbf{H}_1\mathbf{H}_2 = \mathbf{H}_3\mathbf{H}_4$, since $\alpha$ and $\beta$ can be chosen arbitrarily, as long as the unit norm condition is satisfied.

### B.6.2 Illustration of the Optimality of the First Greedy Step

In the first step, the eigenvector chosen is $\alpha\mathbf{u}_1 + \beta\mathbf{u}_2$, corresponding to an eigenvalue of $-1 + 2k^2$. We show that such a choice is optimal in the space $\mathcal{H}_1$ to approximate an orthogonal matrix $\mathbf{V} = \mathbf{H}_1\mathbf{H}_2 \in \mathcal{H}_2$ (despite the fact that the minimum eigenvalue has multiplicity greater than 1).

Let the Householder matrix in $\mathcal{H}_1$ be $\mathbf{H}_3 = \mathbf{I} - 2\mathbf{v}\mathbf{v}^\top$. The error is thus:

$$
\begin{aligned}
&\|\mathbf{H}_1\mathbf{H}_2 - \mathbf{H}_3\|_F^2 \\
&= \|\mathbf{H}_3\mathbf{H}_1\mathbf{H}_2 - \mathbf{H}_3\mathbf{H}_3\|_F^2 \\
&= \|\mathbf{H}_3\mathbf{H}_1\mathbf{H}_2 - \mathbf{I}\|_F^2 \\
&= \operatorname{tr}\big((\mathbf{H}_3\mathbf{H}_1\mathbf{H}_2 - \mathbf{I})^\top(\mathbf{H}_3\mathbf{H}_1\mathbf{H}_2 - \mathbf{I})\big).
\end{aligned}
\tag{32}
$$

Our objective is to minimize this, so we have:

$$
\begin{aligned}
&\min \operatorname{tr}\big((\mathbf{H}_3\mathbf{H}_1\mathbf{H}_2 - \mathbf{I})^\top(\mathbf{H}_3\mathbf{H}_1\mathbf{H}_2 - \mathbf{I})\big) \\
&= \min\big(2\operatorname{tr}(\mathbf{I}) - 2\operatorname{tr}(\mathbf{H}_3\mathbf{H}_1\mathbf{H}_2)\big) \\
&= \max \operatorname{tr}(\mathbf{H}_3\mathbf{H}_1\mathbf{H}_2).
\end{aligned}
\tag{33}
$$

Substituting $\mathbf{H}_3 = \mathbf{I} - 2\mathbf{v}\mathbf{v}^\top$ and the expansion of $\mathbf{H}_1\mathbf{H}_2$, we get:

$$
\begin{aligned}
&\max \operatorname{tr}\left((\mathbf{I} - 2\mathbf{v}\mathbf{v}^\top)(\mathbf{I} - 2\mathbf{u}_1\mathbf{u}_1^\top - 2\mathbf{u}_2\mathbf{u}_2^\top + 4(\mathbf{u}_1^\top\mathbf{u}_2)\mathbf{u}_1\mathbf{u}_2^\top)\right) \\
&= \min \operatorname{tr}\left(\mathbf{v}\mathbf{v}^\top(\mathbf{I} - 2\mathbf{u}_1\mathbf{u}_1^\top - 2\mathbf{u}_2\mathbf{u}_2^\top + 4(\mathbf{u}_1^\top\mathbf{u}_2)\mathbf{u}_1\mathbf{u}_2^\top)\right) \\
&= \min \left(1 - 2(\mathbf{v}^\top\mathbf{u}_1)^2 - 2(\mathbf{v}^\top\mathbf{u}_2)^2 + 4k(\mathbf{v}^\top\mathbf{u}_1)(\mathbf{v}^\top\mathbf{u}_2)\right)
\end{aligned}
\tag{34}
$$

Clearly, if $\mathbf{v} \notin \operatorname{span}(\mathbf{u}_1, \mathbf{u}_2)$, then the value above is 1, which is suboptimal. To see this, consider the following- we aim to prove that the expression

$$
a^2 + b^2 - 2kab \geq 0
$$

is always non-negative for real scalars $a, b$, and for $k = \mathbf{u}_1^\top\mathbf{u}_2 \in [-1, 1]$, where $\mathbf{u}_1, \mathbf{u}_2$ are unit vectors.

Define the vector $\mathbf{x} = [a\ b]^\top$. Then we can write:

$$
a^2 + b^2 - 2kab = \mathbf{x}^\top \begin{bmatrix} 1 & -k \\ -k & 1 \end{bmatrix} \mathbf{x}
$$
$$
= \mathbf{x}^\top \mathbf{M} \mathbf{x}
$$

To determine whether $\mathbf{M}$ is positive semi-definite, we compute its eigenvalues:

$$
\begin{aligned}
\det(\mathbf{M} - \lambda\mathbf{I}) &= (1 - \lambda)^2 - k^2 = 0 \\
&\Rightarrow (1 - \lambda)^2 = k^2 \\
&\Rightarrow \lambda = 1 \pm k \\
&\Rightarrow 1 - |k| \leq \lambda \leq 1 + |k|.
\end{aligned}
$$

Since $k = \mathbf{u}_1^\top\mathbf{u}_2 \in [-1, 1]$, the eigenvalues are both non-negative. Therefore, $\mathbf{M}$ is positive semi-definite, and strictly positive definite when $|k| < 1$. Hence, for all real $a, b$, we conclude:

$$
a^2 + b^2 - 2kab = \mathbf{x}^\top \mathbf{M} \mathbf{x} \geq 0.
$$

The expression is equal to zero if and only if $\mathbf{x}$ lies in the null space of $\mathbf{M}$, which only happens when $\det(\mathbf{M}) = 0$, i.e., $|k| = 1$. In this case:

- If $k = 1$, then $\mathbf{u}_1 = \mathbf{u}_2$, and the equality holds if $a = b$.
- If $k = -1$, then $\mathbf{u}_1 = -\mathbf{u}_2$, and the equality holds if $a = -b$.

This concludes the proof. Thus, $\mathbf{v}$ must be of the form $\alpha\mathbf{u}_1 + \beta\mathbf{u}_2$, which is precisely our choice, illustrating optimality in the recovery of orthogonal matrices belonging to $\mathcal{H}_2$. We have already shown above that $\mathbf{H}_4$ is the optimal choice; thus, this process gives optimal recovery.

## C   PROOFS-DENOISING

### C.1   PRELIMINARIES FOR DENOISING

**Gaussian orthogonal Ensemble (GOE)**   A random matrix $\mathbf{X} \in \mathrm{GOE}_n$ satisfies: for $1 \leq i \leq j \leq n$, the entries $\mathbf{X_{ij}}$ are independently distributed as $\mathbf{X}_{ii} \sim \mathcal{N}(0, 2)$ for diagonal elements, and $\mathbf{X}_{ij} \sim \mathcal{N}(0, 1)$ for off-diagonal elements (with $\mathbf{X}_{ji} = \mathbf{X}_{ij}$ ensuring symmetry).

**Real Ginibre Ensemble**   Ginibre (1965) A random matrix $\mathbf{X} \in \mathbb{R}^{n \times n}$ is said to belong to the *real Ginibre ensemble (GinOE)* if its entries are independently chosen with probability density functions

$$\frac{1}{\sqrt{2\pi}} e^{-\frac{1}{2}x_{jk}^2}, \quad 1 \leq j, k \leq n. \tag{35}$$

The joint probability density function of all the independent entries is

$$f(\mathbf{X}) = \prod_{1 \leq j,k \leq n} \frac{1}{\sqrt{2\pi}} e^{-\frac{1}{2}x_{jk}^2}$$

$$= (2\pi)^{-\frac{1}{2}n^2} \exp\left(-\frac{1}{2}\sum_{j,k=1}^{n} x_{jk}^2\right)$$

**Semicircle Distribution**   Jiang (2021) The (standard) semicircle distribution is the probability measure on $[-2, 2]$ with density

$$d\mu_{\mathrm{SC}}(x) = \frac{1}{2\pi}\sqrt{4 - x^2}\, dx.$$

**Tracy–Widom Law**   Let $\lambda_{\max}$ be the largest eigenvalue of a matrix $\mathbf{G} \in \mathrm{GOE}_n$, i.e., a real symmetric $n \times n$ random matrix from the Gaussian orthogonal Ensemble. Then, as $n \to \infty$, the properly centered and scaled largest eigenvalue converges in distribution to the *Tracy–Widom distribution of type 1*, denoted $F_1(t)$. Hence, as $n \to \infty$,

$$\lambda_{\max} \Rightarrow \sqrt{2n} + \frac{1}{\sqrt{2}n^{1/6}}F_1, \tag{36}$$

**Weyl's Inequality**   Let $\mathbf{A}, \mathbf{B} \in \mathbb{C}^{n \times n}$ be Hermitian, and let $\lambda_1(\cdot) \leq \lambda_2(\cdot) \leq \cdots \leq \lambda_n(\cdot)$ denote eigenvalues in non-increasing order. Then, for any $1 \leq k \leq n$,

$$\lambda_k(\mathbf{A}) + \lambda_1(\mathbf{B}) \leq \lambda_k(\mathbf{A} + \mathbf{B}) \leq \lambda_k(\mathbf{A}) + \lambda_n(\mathbf{B}).$$

**Davis-Kahan sinΘ Theorem**   Let $\mathbf{A}, \mathbf{E} \in \mathbb{C}^{n \times n}$ be Hermitian, set $\widetilde{\mathbf{A}} = \mathbf{A} + \mathbf{E}$. Let $\mathbf{u}$ be a unit eigenvector of $\mathbf{A}$ with eigenvalue $\lambda$, and let $\widetilde{\mathbf{u}}$ be a unit eigenvector of $\widetilde{\mathbf{A}}$ with eigenvalue $\widetilde{\lambda}$. Let $\mathcal{S} = \mathrm{span}\{\mathbf{u}\}$ and $\widetilde{\mathcal{S}} = \mathrm{span}\{\widetilde{\mathbf{u}}\}$. Assume the spectral gap

$$\mathrm{dist}\big(\lambda,\ \mathrm{spec}(\mathbf{A}) \setminus \{\lambda\}\big) = \delta > 0. \quad \text{then,}$$

$$\sin\angle(\mathcal{S}, \widetilde{\mathcal{S}}) \leq \frac{\|\mathbf{E}\|_2}{\delta}.$$

Our denoising algorithm (Algorithm 2) works for noise matrices belonging to GOE as well as for those belonging to GinOE. The denoising guarantees provided use the preliminaries above.

### C.2   LEMMA 4

From the Tracy-Widom Law, we know:

$$\lambda_{\max}(\mathbf{N}_{\mathrm{sym}}) \leq C n^{1/2-\alpha} \quad \text{w.h.p.,}$$

for some constant $C > 0$. We analyze the symmetric part of the noisy observation, $\mathbf{V}_{\text{sym}}$, and apply Weyl's inequality, which gives:

$$\lambda_i(\mathbf{H}^*) + \lambda_{\min}(\mathbf{N}_{\text{sym}}) \leq \lambda_i(\mathbf{V}_{\text{sym}}) \leq \lambda_i(\mathbf{H}^*) + \lambda_{\max}(\mathbf{N}_{\text{sym}}).$$

Since the eigenvalues of a Householder matrix are $\{-1, +1, \ldots, +1\}$, the smallest eigenvalue of $\mathbf{V}_{\text{sym}}$ satisfies:

$$\lambda_{\min}(\mathbf{V}_{\text{sym}}) \leq -1 + Cn^{1/2-\alpha} \quad \text{w.h.p.}$$

and the remaining eigenvalues are all at least

$$1 - Cn^{1/2-\alpha} \quad \text{w.h.p.}$$

## C.3 THEOREM 4

From Lemma 4, it follows that the eigenvector corresponding to the smallest eigenvalue of $\mathbf{V}_{\text{sym}}$ remains close to the true vector $\mathbf{u}$, provided $\alpha > 1/2$. To quantify this closeness, we use the Davis–Kahan $\sin\Theta$ theorem, which yields:

$$\sin\left(\angle(\hat{\mathbf{u}}, \mathbf{u})\right) \leq \frac{1}{n^\alpha} \left\| \frac{\mathbf{N} + \mathbf{N}^\top}{2} \right\|_2 = Cn^{1/2-\alpha} \quad \text{w.h.p.} \tag{37}$$

Since the eigengap of a Householder matrix is 2. We now use this to bound the error between the recovered ($\hat{\mathbf{H}} = \mathbf{I} - 2\hat{\mathbf{u}}\hat{\mathbf{u}}^\mathsf{T}$) and true Householder matrices in Frobenius norm:

$$\left\| (\mathbf{I} - 2\mathbf{u}^*\mathbf{u}^{*\top}) - (\mathbf{I} - 2\hat{\mathbf{u}}\hat{\mathbf{u}}^\top) \right\|_F^2$$
$$= \left\| 2\hat{\mathbf{u}}\hat{\mathbf{u}}^\top - 2\mathbf{u}^*\mathbf{u}^{*\top} \right\|_F^2$$
$$= 4 \left\| \hat{\mathbf{u}}\hat{\mathbf{u}}^\top - \mathbf{u}^*\mathbf{u}^{*\top} \right\|_F^2$$
$$= 4 \cdot 2 \left( 1 - \cos^2\left(\angle(\hat{\mathbf{u}}, \mathbf{u}^*)\right)\right)$$
$$= 8\sin^2\left(\angle(\hat{\mathbf{u}}, \mathbf{u}^*)\right). \tag{38}$$

Substituting the earlier bound, we conclude:

$$\left\| \hat{\mathbf{H}} - \mathbf{H}^* \right\|_F^2 \leq Cn^{1-2\alpha} \quad \text{w.h.p.}$$

## C.4 THEOREM 5

Consider

$$\mathbf{V} = \mathbf{H}_1^*\mathbf{H}_2^* \cdots \mathbf{H}_m^* + \frac{1}{n^\alpha}\mathbf{N}, \text{ let}$$
$$\mathbf{V}^* = \mathbf{H}_1^*\mathbf{H}_2^* \cdots \mathbf{H}_m^* \text{ and} \tag{39}$$
$$\mathbf{\Delta V} = \frac{1}{n^\alpha}\mathbf{N}$$

For the sake of notational simplicity, let

$$\mathbf{A} = \mathbf{V}_{\text{sym}}, \ \mathbf{\Delta A} = (\mathbf{\Delta V})_{\text{sym}} \tag{40}$$

Let the terms involved in the noise-free recovery of the above be as follows: at the $k^{th}$ step, the current iterate is $\mathbf{V}^{(k)*}$, its symmetric part is $\mathbf{A}^{(k)*}$, the corresponding eigenvector chosen is $\mathbf{u}^{(k)*}$, and the Householder reflector formed from this vector is $\mathbf{H}_k^*$. Consider a similar notation for the corresponding noisy versions: $\mathbf{V}^{(k)}, \ \mathbf{A}^{(k)}, \ \mathbf{u}^{(k)}, \ \mathbf{H}^{(k)}$.

Consider symmetric matrix $\mathbf{V}^{(k)*}$. Using spectral decomposition,

$$\mathbf{V}^{(k)*} = \mathbf{U}\mathbf{\Lambda}\mathbf{U}^\top, \ \Lambda = \text{diag}(\lambda_1^{(k)*}, ..\lambda_n^{(k)*}) \tag{41}$$

Define the rank 1 projector on $\mathbf{u}_j$ as $\Pi_j$, i.e.,

$$\Pi_j = \mathbf{u}_j\mathbf{u}_j^\top, \text{ then,}$$
$$\mathbf{A}^{(k)*}\Pi_j = \lambda_j^{(k)*}\Pi_j, \tag{42}$$
$$\sum_j \Pi_j = 1$$

We obtain the above by using the fact that the eigen basis above consists of the eigenvectors of $\mathbf{U}$, which are orthogonal to each other, as well as equation 11. Note that if the eigenvalues are not simple, as is the case for $\mathbf{V}^{(k)*}$, the projector then consists of a sum of rank 1 terms, i.e., the sum of projectors corresponding to that eigenvalue. We will show our results for any eigenvector corresponding to an eigenvalue since we have already established the relationships between the eigenvectors and eigenvalues in Lemma 1. From here forth, we note that $\Pi_i$ is the projector for the most negative eigenvalue, and drop the subscript $i$ for notational simplicity. Let subscript $i$ denote the minimum eigenvalue.

Let

$$\tilde{\mathbf{A}}^{(k)} = \mathbf{A}^{(k)} + \Delta\mathbf{A}^{(k)} \text{ and}$$
$$\tilde{\mathbf{A}}^{(k)}\tilde{\Pi} = \tilde{\lambda}_i^{(k)}\tilde{\Pi}, \text{ where}$$
$$\tilde{\Pi}^{(k)} = \Pi^{(k)} + \Delta\Pi^{(k)}, \text{ and} \tag{43}$$
$$\tilde{\lambda}_i^{(k)} = \lambda_i^{(k)} + \Delta\lambda_i^{(k)}$$

Also consider the two orthogonal subspaces (col denotes column space) $(\mathcal{P}; \mathcal{Q} = \mathcal{P}^\perp)$

$$\mathcal{P} = \text{col}(\Pi^{(k)}),$$
$$\mathcal{Q} = \text{col}(\mathbf{I} - \Pi^{(k)})$$

Where the orthogonality of the subspaces stems from the fact that $(\mathbf{I} - \mathbf{u}\mathbf{u}^\top)\mathbf{v} = (\mathbf{v} - (\mathbf{u}^\top\mathbf{v})\mathbf{u}) \perp (\mathbf{u}\mathbf{u}^\top)\mathbf{v} = (\mathbf{u}^\top\mathbf{v})\mathbf{u}$. Thus,

$$(\mathbf{A}^{(k)} + \Delta\mathbf{A}^{(k)})(\Pi^{(k)} + \Delta\Pi^{(k)}) = (\lambda_i^{(k)} + \Delta\lambda_i^{(k)})(\Pi^{(k)} + \Delta\Pi^{(k)})$$
$$\Rightarrow \mathbf{A}^{(k)}\Delta\Pi^{(k)} + \Pi^{(k)}\Delta\mathbf{A}^{(k)} = \lambda_i^{(k)}\Delta\Pi^{(k)} + \Delta\lambda_i^{(k)}\Pi^{(k)} \tag{44}$$
$$\Rightarrow (\mathbf{A}^{(k)} - \lambda_i^{(k)}\mathbf{I})\Delta\Pi^{(k)} + \Pi^{(k)}\Delta\mathbf{A}^{(k)} = \Delta\lambda_i^{(k)}\Pi^{(k)}$$

on dropping second-order terms. Let $Q$ be the projection operator corresponding to the subspace $\mathcal{Q}$. Then,

$$Q(\mathbf{A}^{(k)} - \lambda_i^{(k)}\mathbf{I})\Delta\Pi^{(k)} + Q\Pi^{(k)}\Delta\mathbf{A}^{(k)} = 0$$
$$\Rightarrow \Delta\Pi_Q^{(k)} = (\mathbf{A}^{(k)} - \lambda_i^{(k)}\mathbf{I})^{-1}\Delta\mathbf{A}_Q^{(k)} \tag{45}$$

We add the subscript to indicate that the inverse is computed in the complement space. Following standard perturbation theory, we define the reduced resolvent $S$ on the complement space $\mathcal{Q}$ for the $k^{th}$ iterate as

$$S_k := Q(\mathbf{A}^{(k)} - \lambda_i^{(k)}\mathbf{I})^{-1}Q = \sum_{j:\lambda_j \neq \lambda_i} \frac{\Pi_j^{(k)}}{\lambda_j - \lambda_i} \tag{46}$$

Now, we use Theorem 3 from (Greenbaum et al., 2020) (which can be extended to subspaces as per (Kato, 2013)) to obtain

$$\Delta\Pi^{(k)} = -S_k\Delta\mathbf{A}^{(k)}\Pi^{(k)} - \Pi^{(k)}\Delta\mathbf{A}^{(k)}S_k + \mathcal{O}(\|\mathbf{A}^{(k)}\|^2) \tag{47}$$

where the projection operator simply projects onto a subspace instead of a single vector. At step $k$, the algorithm constructs a Householder reflector from a chosen unit vector $\widehat{\mathbf{u}}^{(k)}$. We have,

$$\|\widehat{\mathbf{H}}_k - \mathbf{H}_k^*\|_F^2 \le C\|\widetilde{\Pi}^{(k)} - \Pi^{(k)}\|_F^2.$$

$$\le \|\Delta\Pi^{(k)} - S_k\Delta\mathbf{A}^{(k)}\Pi^{(k)} - \Pi^{(k)}\Delta\mathbf{A}^{(k)}S_k + \mathcal{O}(\|\mathbf{A}^{(k)}\|^2)\|_F^2 \quad (48)$$

$$\le \|\Delta\Pi^{(k)} - S_k\Delta\mathbf{A}^{(k)}\Pi^{(k)} - \Pi^{(k)}\Delta\mathbf{A}^{(k)}S_k\|_F^2 + \mathcal{O}(\|\mathbf{A}^{(k)}\|^3)$$

Let $\Delta\mathbf{H}_k = \mathbf{H}_k - \mathbf{H}_k^*$. Thus,

$$\mathbf{V} - \mathbf{V}^* = \mathbf{H}_m\mathbf{H}_{m-1}\cdots\mathbf{H}_1 - \mathbf{H}_m^*\mathbf{H}_{m-1}^*\cdots\mathbf{H}_1^*$$

$$= \sum_{j=1}^{m}\left(\prod_{k=1}^{j-1}\mathbf{H}_k^*\right)(\Delta\mathbf{H}_j)\left(\prod_{k=j+1}^{m}\mathbf{H}_k\right) + \mathbf{R} \quad (49)$$

Where $\mathbf{R}$ constitutes all remaining terms with products of two or more $\Delta\mathbf{H}$'s. Now we use the distributional properties of our perturbation. Let $\mathbf{U}^{(k)} \in \mathbb{R}^{n \times r_k}$, where $r_k \in \{1, 2\}$, with the precise value of $r_k$ depending on the parity of the iteration.

Using Hanson-Wright concentration bounds (Rudelson & Vershynin, 2013), we have w.h.p.

$$\|\Delta\mathbf{H}_k\|_F^2 \le C\frac{r_k\|\mathbf{S}^{(k)}\|_F^2}{n^{2\alpha}}\log^\theta n + \mathcal{O}(\|\Delta\mathbf{A}^{(k)}\|_2^3),$$

$$\Rightarrow \sum_{k=1}^{m}\|\Delta\mathbf{H}_k\|_F^2 \le C\frac{\log^\theta n}{n^{2\alpha}}\sum_{k=1}^{m}r_k\|\mathbf{S}^{(k)}\|_F^2 + \mathcal{O}\left(\sum_{k=1}^{m}\|\Delta\mathbf{A}^{(k)}\|_2^3\right). \quad (50)$$

for a constant $C$ and $\theta^2$ chosen according to required concentration. Using $\|A + B\|_F^2 \le 2\|A\|_F^2 + 2\|B\|_F^2$,

$$\|\widehat{\mathbf{V}} - \mathbf{V}^*\|_F^2 \le 2\left\|\sum_{k=1}^{m}\mathbf{P}_{k+1}^*\Delta\mathbf{H}_k\mathbf{Q}_{k-1}^*\right\|_F^2 + 2\|\mathbf{R}\|_F^2$$

$$\le 2m\sum_{k=1}^{m}\|\Delta\mathbf{H}_k\|_F^2 + 2\|\mathbf{R}\|_F^2 \quad \text{(Cauchy-Schwarz Inequality)} \quad (51)$$

We bound $\|\mathbf{R}\|_F$ to obtain our final expression. We use the Cauchy-Schwarz Inequality once again in the derivation below.

$$\|\mathbf{R}\|_F \le \sum_{t\ge 2}\sum_{1\le i_1 < \cdots < i_t \le m}\left\|\mathbf{P}^*\,\Delta\mathbf{H}_{i_1}\,\Delta\mathbf{H}_{i_2}\cdots\Delta\mathbf{H}_{i_t}\,\mathbf{Q}^*\right\|_F$$

$$\le \sum_{t\ge 2}\sum_{1\le i_1 < \cdots < i_t \le m}\prod_{s=1}^{t}\|\Delta\mathbf{H}_{i_s}\|_F$$

$$\Rightarrow \|\mathbf{R}\|_F^2 \le C'\left(\sum_{k=1}^{m}r_k\right)\sum_{k=1}^{m}\|\Delta\mathbf{H}_k\|_F^2 \le C'm\sum_{k=1}^{m}\|\Delta\mathbf{H}_k\|_F^2, \quad (52)$$

$$\text{since } \sum_{k=1}^{m}r_k \le 2m.$$

Combining the equations above, the final bound is

$$\|\widehat{\mathbf{V}} - \mathbf{V}^*\|_F^2 \le \mathcal{O}\left(\frac{m\log^\theta n}{n^{2\alpha}}\sum_{k=1}^{m}r_k\|\mathbf{S}^{(k)}\|_F^2\right) + \mathcal{O}\left(m\sum_{k=1}^{m}\|\Delta\mathbf{A}^{(k)}\|_2^3\right). \quad (53)$$

---

[2]where $\theta$ is chosen appropriately to obtain the desired tail bound from the Hanson-Wright inequality

A note on the typical case: let $\mathbf{V}^*$ be Haar-distributed on $O(n)$. Then with probability at least $1 - n^{-c}$ for some $c > 0$,

$$\sum_{k=1}^{m} \|S^{(k)}\|_F^2 = \mathcal{O}(n \log n).$$

Therefore,

$$\|\widehat{\mathbf{V}} - \mathbf{V}^*\|_F^2 \leq \mathcal{O}\Big(\frac{mn \log^{\theta+1} n}{n^{2\alpha}}\Big) + \mathcal{O}\Big(m \sum_{k=1}^{m} \|\Delta \mathbf{A}^{(k)}\|_2^3\Big). \tag{54}$$

The second term in the above expression is at most $\mathcal{O}(m^2 n^{3(1/2-\alpha)})$. If $m = \Omega(\sqrt{n})$ in such scenarios, it is better to directly apply Orthogonal Procrustes for best denoising, though all computational benefits of the approximated matrices are lost. Refer (Tao & Vu, 2010), (Mehta, 2004) for the counting and interlacing arguments regarding eigenvalues of matrices distributed on $O(n)$.

### C.4.1 COMPARISON WITH ORTHOGONAL PROCRUSTES

A natural question arises from the analysis above- why not simply project the noisy matrix to the closest orthogonal matrix, and then find the optimal Householder decomposition using Algorithm 1? The answer lies in the possible distance between the orthogonal projection and the ground truth product of Householder reflectors. The error above can be bounded as (for $\mathbf{V} = \mathbf{V}^* + (1/n^\alpha)\mathbf{N}$,, where the product $\mathbf{H}_1 \mathbf{H}_2 \cdots \mathbf{H}_m = \mathbf{V}^*$ ):

$$\begin{aligned}
\|\mathbf{V}^* - \hat{\mathbf{V}}\|_F^2 &\leq \|\mathbf{V}^* - \mathbf{V}\|_F^2 + \|\mathbf{V} - \hat{\mathbf{V}}\|_F^2 \\
&\leq \sum_{i=1}^{n} (Cn^{1/2-\alpha})^2 + \|\mathbf{U}(\mathbf{\Sigma} - \mathbf{I})\mathbf{V}^\mathsf{T}\|_F^2 \\
&= \mathcal{O}(n^{2(1-\alpha)}) + \sum_{i=1}^{n} (\sigma_i - 1)^2 \\
&= \mathcal{O}(n^{2(1-\alpha)})
\end{aligned} \tag{55}$$

where $\mathbf{V} = \mathbf{U}\mathbf{\Sigma}\mathbf{V}^\mathsf{T}$. Thus, Orthogonal Procrustes is a poorer choice of solution for our case.

### C.5 EXTENSION TO GENERAL NOISE DISTRIBUTIONS

In this section, we provide slightly looser bounds on the recovery of ground truth orthogonal matrices that have been perturbed by arbitrary noise distributions. The recovery error between the ground truth $\mathbf{V}^* = \prod_{i=1}^{m} \mathbf{H}_i^*$ and the estimated $\hat{\mathbf{V}} = \prod_{i=1}^{m} \hat{\mathbf{H}}_i$ is

$$\begin{aligned}
\big\|\mathbf{H}_1^* \cdots \mathbf{H}_m^* - \hat{\mathbf{H}}_1 \cdots \hat{\mathbf{H}}_m\big\|_F &= \big\|\mathbf{H}_1^* \cdots \mathbf{H}_m^* - \hat{\mathbf{H}}_1 \cdots \hat{\mathbf{H}}_{m-1}\mathbf{H}_m^* \\
&\quad + \hat{\mathbf{H}}_1 \cdots \hat{\mathbf{H}}_{m-1}\mathbf{H}_m^* - \hat{\mathbf{H}}_1 \cdots \hat{\mathbf{H}}_m\big\|_F \\
&= \big\|(\mathbf{H}_1^* \cdots \mathbf{H}_{m-1}^* - \hat{\mathbf{H}}_1 \cdots \hat{\mathbf{H}}_{m-1})\mathbf{H}_m^* \\
&\quad + \hat{\mathbf{H}}_1 \cdots \hat{\mathbf{H}}_{m-1}(\mathbf{H}_m^* - \hat{\mathbf{H}}_m)\big\|_F \\
&\leq \big\|\mathbf{H}_1^* \cdots \mathbf{H}_{m-1}^* - \hat{\mathbf{H}}_1 \cdots \hat{\mathbf{H}}_{m-1}\big\|_F + \big\|\mathbf{H}_m^* - \hat{\mathbf{H}}_m\big\|_F.
\end{aligned} \tag{56}$$

On recursively repeating this procedure for the first term, we get:

$$\begin{aligned}
\big\|\mathbf{H}_1^* \cdots \mathbf{H}_m^* - \hat{\mathbf{H}}_1 \cdots \hat{\mathbf{H}}_m\big\|_F &\leq \sum_{i=1}^{m} \|\mathbf{H}_i^* - \hat{\mathbf{H}}_i\|_F \\
&\leq Cmn^{-\alpha}\|\mathbf{N}_{\text{sym}}\|_2 \\
&= \mathcal{O}(mn^{-\alpha}\|\mathbf{N}_{\text{sym}}\|_2)
\end{aligned} \tag{57}$$

This procedure also illustrates why it was essential to convert equation 5 in the format of equation 6. This bound is weaker than the one in Theorem 5 when we had i.i.d Gaussian noise. The $\alpha$ required for error to reduce with iterations is 1 if we use the bound in 57. If we know the spectral norm of the symmetric part of the noise matrix $\mathbf{N}_{\text{sym}}$, the result can be made more concrete. Consider the following examples:

For any symmetric random matrix $\mathbf{A} \in \mathbb{R}^{n \times n}$ with independent, zero mean, sub-gaussian entries on and above the diagonal, the operator norm is bounded as follows Vershynin (2018):

$$\|\mathbf{A}\| \leq C(\sqrt{n} + t)$$
$$\text{with probability at least } 1 - 4\exp(-t^2).$$
(58)

For $\mathbf{A} \in \mathbb{R}^{n \times m}$, we have

$$\|\mathbf{A}\| \leq C(\sqrt{n} + \sqrt{m} + t)$$
$$\text{with probability at least } 1 - 2\exp(-t^2).$$
(59)

The equations above end up covering Gaussian, Uniform, and non-normalized low rank noise. We provide a simple derivation for low rank noise $\mathbf{N} = \mathbf{U}\mathbf{V}^\top$, with $\mathbf{U}, \mathbf{V} \in \mathbb{R}^{n \times r}$ below; if the entries of $\mathbf{U}, \mathbf{V}$, are generated using $\mathcal{N}(0,1)$, then $\|\mathbf{N}_{\text{sym}}\|_2 = \Theta(n+r)$, giving us the loose bound (noting $r \leq n$),

$$\|\mathbf{V}^* - \hat{\mathbf{V}}\|_F = \mathcal{O}\left(mn^{1-\alpha}\right)$$
(60)

Had the columns of $\mathbf{U}, \mathbf{V}$ been normalized, we get

$$\|\mathbf{V}^* - \hat{\mathbf{V}}\|_F = \mathcal{O}\left(mn^{-\alpha}\right)$$
(61)

If we model one sided anisotropic noise (where $\boldsymbol{\Sigma}$ is an appropriate covariance matrix),

$$\mathbf{N} = \boldsymbol{\Sigma}^{1/2}\mathbf{X}$$
(62)

then, our bound yields

$$\|\mathbf{V}^* - \hat{\mathbf{V}}\|_F = \mathcal{O}\left(mn^{1/2-\alpha}\sqrt{\lambda_{\max}(\boldsymbol{\Sigma})}\right)$$
(63)

**Remark** We illustrate the recovery of an orthogonal matrix in Figure 6 and the recovery for noisy orthogonal matrices in Figure 7. Later on, in section E.2, we also show how the eigenvalues behave during the recovery process.

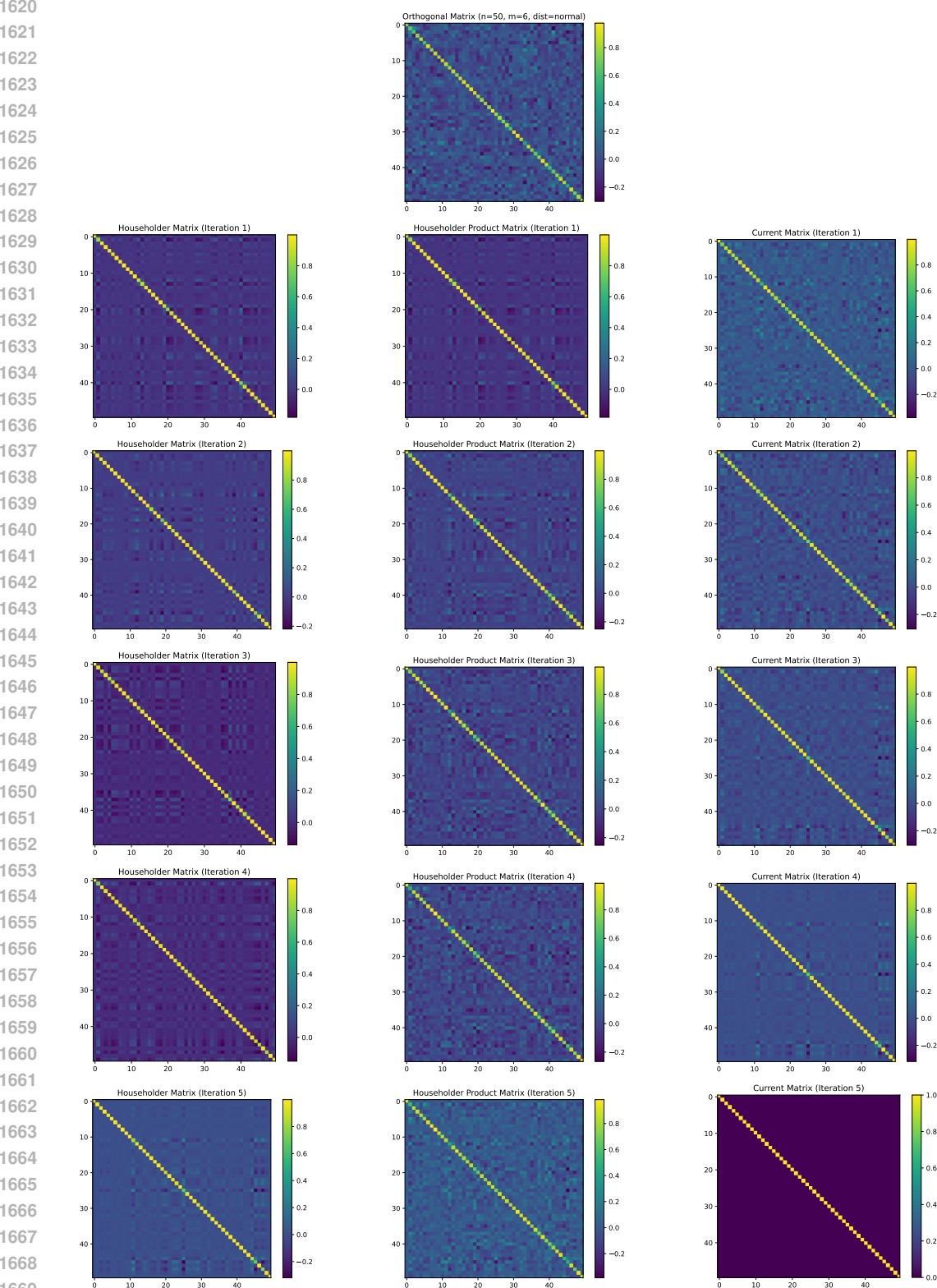

Figure 6: Illustration of orthogonal matrix recovery for a $n = 50$ sized orthogonal matrix composed as a product of $m = 5$. From left to right, the Householder matrix recovered, our current approximation (product of Householders our algorithm has recovered till now), and the current orthogonal iterate.

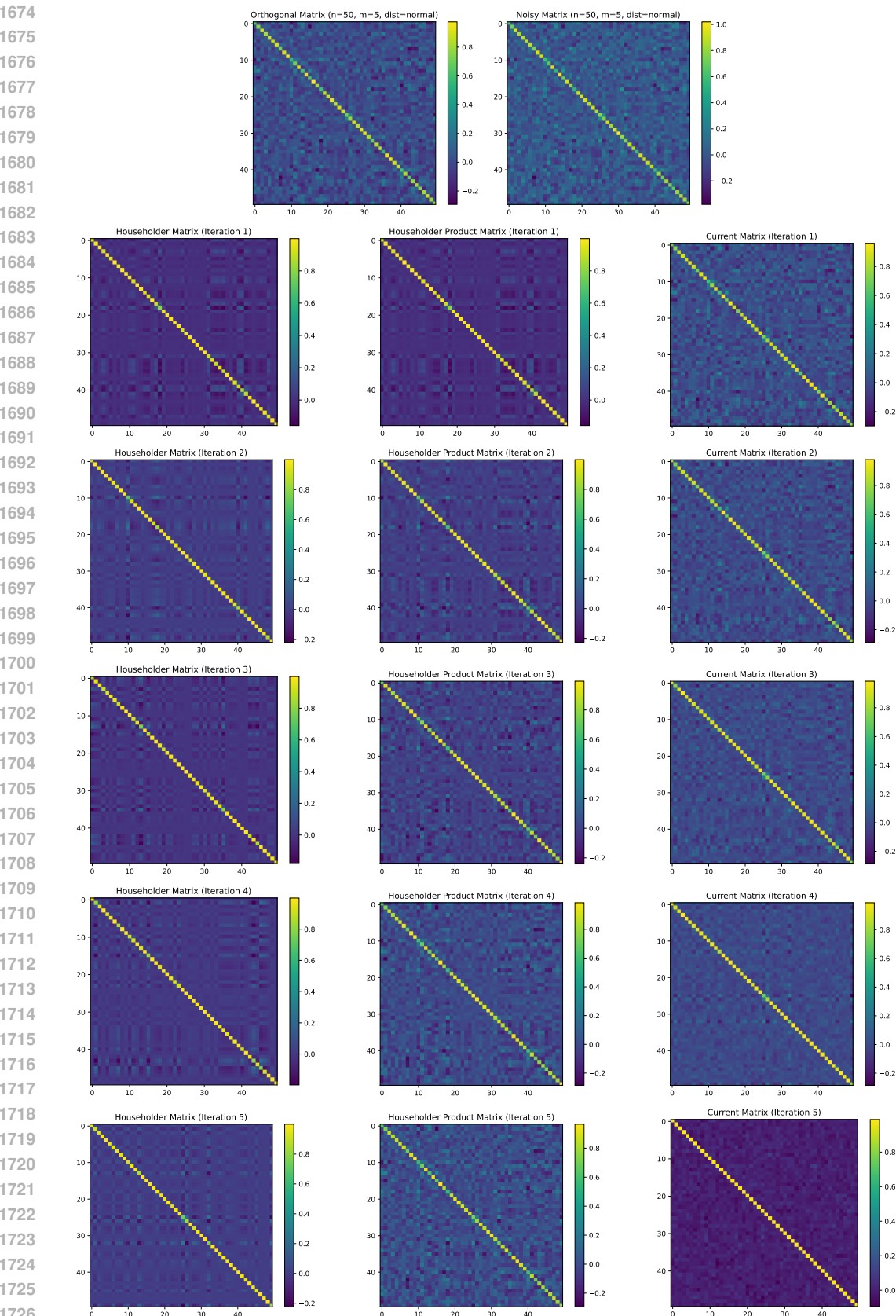

Figure 7: A repeat of Figure 6 but for a noisy version of an input orthogonal matrix with SNR=10.

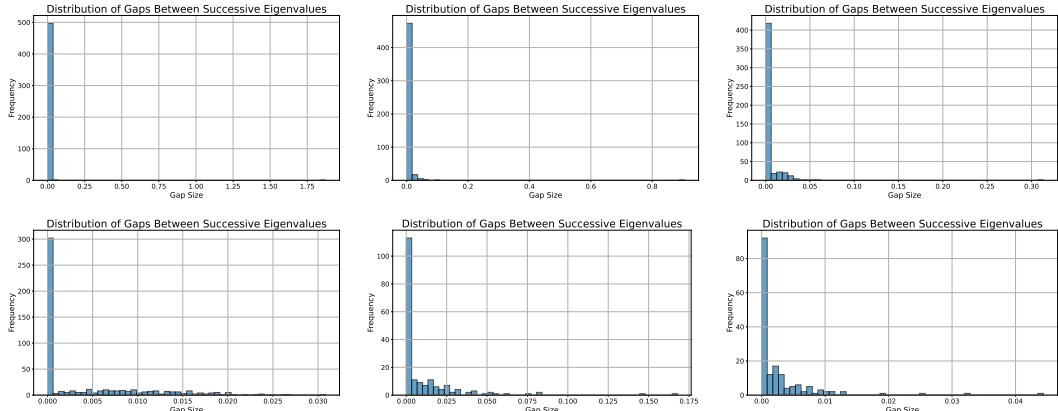

Figure 8: Plots of gap distributions for various matrices. The first four plots are for $n = 500$ and $m = 10, 100, 200, 400$ for an orthogonal matrix generated via the sparse Gaussian distribution. The last two plots are from the spectra of the trained orthogonal recurrent matrices for the copy task and sequential MNIST task respectively. Lanczos' algorithm performs equally well in all cases.

# D ADDITIONAL EXPERIMENTS AND ABLATION STUDY

## D.1 COMPUTATIONAL TIME AND MEMORY USAGE

### D.1.1 COMPUTATIONAL COMPLEXITY

In terms of computational complexity, $\mathbf{H}_i\mathbf{x}$ can be computed in $\mathcal{O}(n)$ arithmetic operations for every $\mathbf{x} \in \mathbb{R}^n$, and in general for any $\mathbf{V} \in \mathcal{H}_m$, the matrix action $\mathbf{V}\mathbf{x}$ can be computed with $\mathcal{O}(mn)$ arithmetic operations if the decomposition is known:

$$\mathbf{V} = \mathbf{H}_1\mathbf{H}_2\ldots\mathbf{H}_m.$$

In terms of FLOPs, applying one Householder reflection

$$\mathbf{H} = \mathbf{I} - 2\mathbf{u}\mathbf{u}^\mathsf{T}$$

to $\mathbf{x} \in \mathbb{R}^n$ requires a dot product and an axpy, $\approx 4n$ FLOPs. A product of $m$ reflections, therefore, costs $\approx 4mn$ FLOPs per vector. By contrast, a dense $n \times n$ matrix–vector multiply costs $\approx 2n^2$ FLOPs. For batch size $B$, the per-step costs are $4Bmn$ vs. $2Bn^2$, giving an idealized FLOP speedup of $n/(2m)$.

These savings are realized by storing the underlying reflectors corresponding to the Householder $\mathbf{H}_i$ instead of storing the matrix $\mathbf{V}$ (the storage requirement thus comes down from $n^2$ to $mn$).

### D.1.2 USING LANCZOS' ALGORITHM TO IMPROVE SPEED

Note that we can use Lanczos' algorithm for faster eigenvalue computation for the symmetric matrix $\mathbf{V}_{sym}$. Table 2 gives a comparison of times taken by the algorithm with and without Lanczos' algorithm (using the default 20 iterations) implemented for varying $m$. In terms of convergence, we observe that generally, 10 Lanczos iterations using SciPy's eigsh solver are suffice, with the residual pretty being floating point precision error when compared to the results from the dense eigh solver. We see this across $m$ values, meaning the theoretical $\mathcal{O}(n^2)$ holds, and we see significant improvements in speed. In Figure 8, we show the gap distributions in various matrices of interest to us- Lanczos' algorithm performs equally well on all of them.

### D.1.3 SCALING OF TIME WITH MATRIX SIZE

As expected, for a fixed $m$, when the size of the orthogonal matrix ($n$) increases, the time taken by the algorithm increases. Here, we present timing details for $m$ fixed at 25 and $n$ increasing from 25 to 5000, while using the version of the algorithm that uses Lanczos' algorithm to determine the smallest eigenvalue-eigenvector pair. Note that as $m$ increases, this time will also increase as illustrated in Table 2. Our results are summarized in Table 3.

Table 2: Time taken by the algorithm (in seconds) to run for an orthogonal matrix $\mathbf{V} \in \mathcal{H}_m$ for various $m$, with and without Lanczos' algorithm; $n = 500$.

| $m$ | Without Lanczos (in s) | With Lanczos (in s) |
|---|---|---|
| 1 | 0.1137 | 0.0399 |
| 5 | 0.5527 | 0.2556 |
| 10 | 1.0791 | 0.5051 |
| 25 | 2.6290 | 1.2075 |
| 50 | 5.4369 | 2.5639 |
| 100 | 11.7821 | 5.3986 |
| 200 | 25.3444 | 10.6191 |
| 400 | 56.6847 | 19.5269 |

Table 3: Time taken by the algorithm (in seconds) to run for an orthogonal matrix $\mathbf{V} \in \mathcal{H}_{25}$ for various $n$, with Lanczos' algorithm.

| $n$ | time (in s) |
|---|---|
| 25 | 0.0083 |
| 50 | 0.0154 |
| 100 | 0.0369 |
| 250 | 0.2377 |
| 500 | 1.1939 |
| 1000 | 5.3420 |
| 2000 | 64.1059 |

Table 4: Time speedup offered by our approach as compared to dense matrix multiplication for various (n,m) pairs. The times are averaged over 7 runs. For example, for $n = 1000, m = 100$, explicit multiplication takes $0.380$ ms while our method takes $0.314$ ms. For $n = 10000, m = 50$, explicit multiplication takes $70.20$ ms while our method takes $1.53$ ms. Note that some entries are missing since $m > n$ for those cells.

| n \ m | 1 | 10 | 20 | 50 | 100 | 200 | 300 | 400 |
|---|---|---|---|---|---|---|---|---|
| 100 | 0.88 | 0.66 | 0.53 | 0.63 | 0.36 | | | |
| 200 | 0.91 | 0.53 | 0.93 | 0.46 | 0.29 | 0.13 | | |
| 300 | 2.07 | 2.18 | 1.07 | 0.57 | 0.34 | 0.17 | 0.11 | |
| 400 | 4.37 | 1.99 | 1.93 | 1.10 | 0.49 | 0.27 | 0.13 | 0.10 |
| 500 | 6.96 | 3.53 | 1.97 | 1.56 | 0.69 | 0.32 | 0.22 | 0.17 |
| 750 | 13.41 | 5.44 | 7.74 | 1.89 | 1.08 | 0.52 | 0.32 | 0.59 |
| 1000 | 26.42 | 10.52 | 6.14 | 3.08 | 1.21 | 1.78 | 0.47 | 0.35 |
| 2000 | 65.14 | 52.09 | 22.48 | 13.99 | 4.04 | 1.92 | 2.21 | 0.65 |
| 5000 | 596.92 | 89.69 | 39.01 | 22.28 | 10.65 | 5.07 | 3.36 | 2.19 |
| 7500 | 760.27 | 151.76 | 124.55 | 26.78 | 15.59 | 6.27 | 3.49 | 4.06 |
| 10000 | 1381.88 | 221.48 | 99.44 | 45.80 | 16.44 | 10.28 | 6.98 | 6.36 |

### D.1.4 PUTTING IT ALL TOGETHER

In tables 4 and 5, we provide the speed ups offered by our method for various $(n, m)$ pairs in terms of time, static memory, and peak memory as compared to explicit matrix vector multiplication. Each entry in Table 4 is of the form $t_{\exp}/t_{house}$, where $t_{\exp}$ is the time taken for dense matrix multiplication, and $t_{house}$ is the time taken to apply a sequence of Householder reflectors to a vector. Note that in terms of implementation, NumPy calls BLAS libraries under the hood, making operations like matmul extremely fast. We implement our sequential Householder product calculation using Numba, which, although much faster than a naïve for loop implementation in Python, can't quite match the the performance of tuned BLAS kernels. Therefore, although we should get a speed up of $(n/2m)$ in theory, this is off by a constant factor in our implementation. This constant ranges somewhere in the range $(2, 4)$. In Table 5, for each $m$, we measure two memory speed up values- static and peak, denoted by s and p in the table. The static memory ratio is $s_{\exp}/s_{house}$ is the ratio of space required to store the parameters, while the peak memory ratio $p_{\exp}/p_{house}$ is the peak memory returned by the tracemalloc() function during function call (for each approach).

Table 5: Memory improvement offered (both static and peak) by our approach as compared to dense matrix multiplication for various (n,m) pairs. For example, for $n = 10000, m = 20$, explicit matrix multiplication requires $800$MB of static storage, while our method requires $1.6$MB. The peak memory for the same case is $0.081$MB for explicit and $0.08$MB for our method. Consider another example- $n = 500, m = 50$- the values are $2$MB and $0.2$MB for storage and $0.005$MB, $0.004$MB for peak memory for explicit multiplication and our method respectively. For rank-m SVD, the ratio is precisely 2, if we also use an approximation of $\mathbf{V}$ in $\mathcal{H}_m$.

| n \ m | 1 | | 10 | | 20 | | 50 | | 100 | | 200 | | 300 | | 400 | |
|---|---|---|---|---|---|---|---|---|---|---|---|---|---|---|---|---|
| | s | p | s | p | s | p | s | p | s | p | s | p | s | p | s | p |
| 100 | 100 | 1.23 | 10 | 1.23 | 5 | 1.23 | 2 | 1.23 | 1 | 1.23 | | | | | | |
| 200 | 200 | 1.13 | 20 | 1.13 | 10 | 1.13 | 4 | 1.13 | 2 | 1.13 | 1 | 1.13 | | | | |
| 300 | 300 | 1.09 | 30 | 1.09 | 15 | 1.09 | 6 | 1.09 | 3 | 1.09 | 1.5 | 1.09 | 1 | 1.09 | | |
| 400 | 400 | 1.07 | 40 | 1.07 | 20 | 1.07 | 8 | 1.07 | 4 | 1.07 | 2 | 1.07 | 1.33 | 1.07 | 1 | 1.07 |
| 500 | 500 | 1.06 | 50 | 1.06 | 25 | 1.06 | 10 | 1.06 | 5 | 1.06 | 2.5 | 1.06 | 1.67 | 1.06 | 1.25 | 1.06 |
| 750 | 750 | 1.04 | 75 | 1.04 | 37.5 | 1.04 | 15 | 1.04 | 7.5 | 1.04 | 3.75 | 1.04 | 2.5 | 1.04 | 1.875 | 1.04 |
| 1000 | 1000 | 1.03 | 100 | 1.03 | 50 | 1.03 | 20 | 1.03 | 10 | 1.03 | 5 | 1.03 | 3.33 | 1.03 | 2.5 | 1.03 |
| 2000 | 2000 | 1.02 | 200 | 1.02 | 100 | 1.02 | 40 | 1.02 | 20 | 1.02 | 10 | 1.02 | 6.67 | 1.02 | 5 | 1.02 |
| 5000 | 5000 | 1 | 500 | 1 | 250 | 1 | 100 | 1 | 50 | 1 | 25 | 1 | 16.67 | 1 | 12.5 | 1 |
| 7500 | 7500 | 1 | 750 | 1 | 375 | 1 | 150 | 1 | 75 | 1 | 37.5 | 1 | 25 | 1 | 18.75 | 1 |
| 10000 | 10000 | 1 | 1000 | 1 | 500 | 1 | 200 | 1 | 100 | 1 | 50 | 1 | 33.33 | 1 | 25 | 1 |

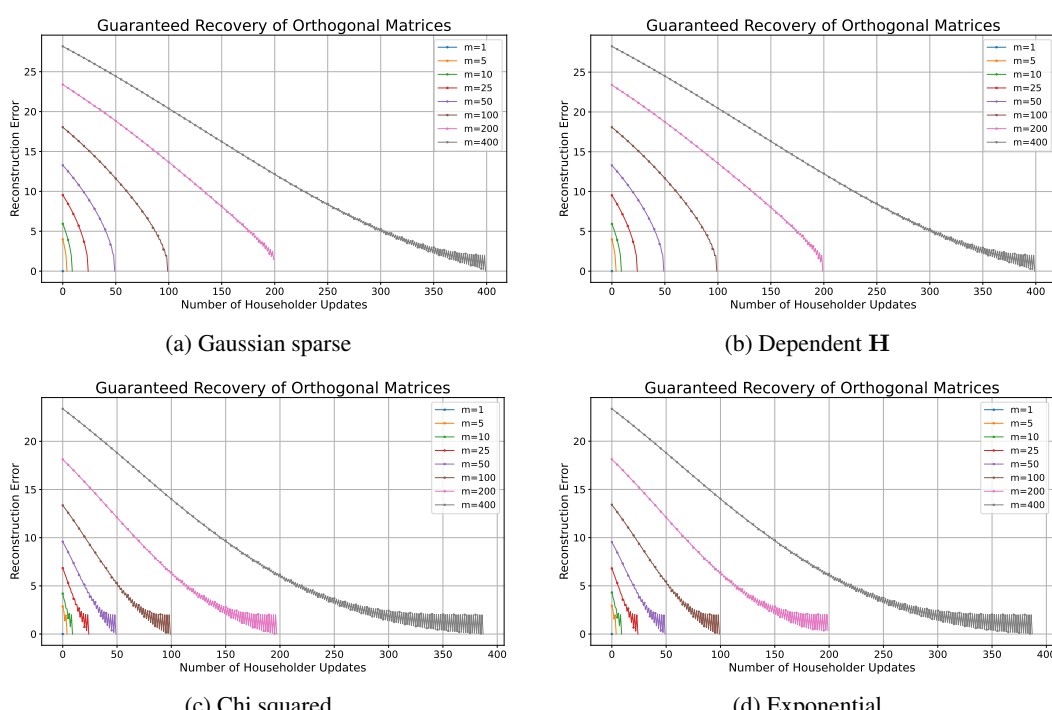

(a) Gaussian sparse        (b) Dependent $\mathbf{H}$

(c) Chi squared        (d) Exponential

Figure 9: Reconstruction error vs. number of Householder updates plots for various distributions of the Householder vector.

## D.2 EXPERIMENTS IN THE NOISELESS SETTING

### D.2.1 MORE EXPERIMENTS ON GUARANTEED RECOVERY

This is a continuation of section 5, where we illustrate recovery for other distributions of the Householder vector. The distributions we use are as follows: **Sparse Gaussian:** Only a fraction $\alpha = 0.02$ of entries are nonzero and drawn from $\mathcal{N}(0, 1)$, with the vector then normalized; **Dependent Gaussian:** $\mathbf{u}_i \sim \mathcal{N}(0, \mathbf{I})$, normalized to unit norm. $\mathbf{u}_{i+1}$ is chosen such that half of the entries of $\mathbf{u}_i$ are retained, the rest are modified, and then normalized; **Chi-squared:** $\mathbf{u}_i$ has i.i.d. entries $\sim \chi^2(2)$, normalized to unit norm. (The slightly earlier convergence here is due to the choice of $\epsilon$; reducing it

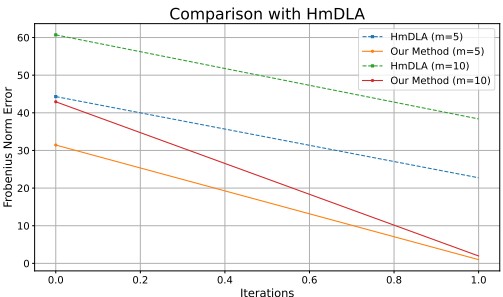 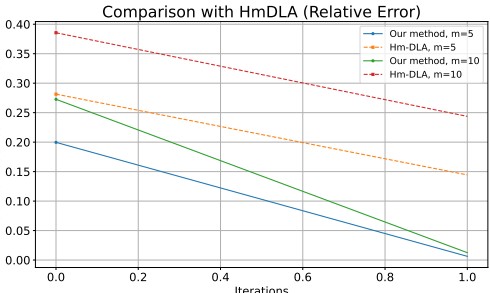

Figure 10: Comparison of absolute (left) and relative (right) error between our method and Hm-DLA Rusu et al. (2016). Our method consistently yields lower reconstruction and relative error, especially in the early iterations. Unlike Hm-DLA, which performs alternating minimization without direct projection guarantees, our method leverages eigenspace structure and performs greedy projection, minimizing error at each step.

Table 6: Comparison of time taken by Hm-DLA and our method. As stated earlier, $n = 500$ for all values of m. Our method outperforms Hm-DLA by factors ranging from 6 to 9. All times reported are in seconds.

| m | 5 | 10 | 20 |
|---|---|---|---|
| Hm-DLA | 0.10 | 0.23 | 0.53 |
| Our method | 0.01 | 0.03 | 0.06 |

slightly further would yield convergence at exactly $m$ iterations); **Exponential:** $\mathbf{u}_i$ has i.i.d. entries $\sim \text{Exp}(1)$, normalized to unit norm.

### D.2.2   COMPARISON WITH HM-DLA

We compare our method with the Hm-DLA algorithm of (Rusu et al., 2016). Firstly, we define relative error as follows: **Relative error:**

$$\text{Relative error} = \frac{\|\hat{\mathbf{Y}}_k - \mathbf{Y}\|_F}{\|\mathbf{Y}\|_F},$$

$$\mathbf{Y} = \mathbf{VX}, \quad \hat{\mathbf{Y}}_k = \hat{\mathbf{V}}_k \mathbf{X},$$

for a sparse coefficient matrix $\mathbf{X}$.

As shown in Figure 10, our method comprehensively outperforms Hm-DLA, with errors reducing by over 80% as $m$ increases. The experiments are conducted for a sparsity level $s = 0.1$, as is used in (Rusu et al., 2016), though our method outperforms Hm-DLA across sparsity levels. The data matrix is constructed as $\mathbf{Y} = \mathbf{VX}$, where $\mathbf{X} \in \mathbb{R}^{500 \times 500}$. Note that the number of Householder updates is $m \times$ (number of iterations). Even on reducing the number of columns of $\mathbf{X}$, we continue to observe similar trends. Moreover, we provide a comparison of the time taken to run both algorithms for various settings in Table 6. Our method outperforms this baseline across all settings in terms of computational efficiency as well.

We emphasize that commonly used orthogonal parameterizations such as Cayley transforms and exponential maps of skew-symmetric matrices target a different setting: they are designed for efficient parameterization during learning, but do not provide an anytime greedy projection of a fixed matrix with per-step error guarantees. For this reason, they are not directly comparable baselines for our problem, where the goal is approximation and denoising of a given orthogonal matrix. Our choice of baselines (Uhlig's constructive method and Hm-DLA) reflects the only prior approaches that address the same approximation setting.

### D.2.3   COMPARISON WITH GRADIENT BASED OPTIMIZATION

We also compare our technique with gradient based optimization of the $mn$ parameters of the m, n-dimensional Householder vectors. We optimize the objective function $\|\mathbf{V} - \hat{\mathbf{H}}_1\hat{\mathbf{H}}_2 \cdots \hat{\mathbf{H}}_m\|_F^2$.

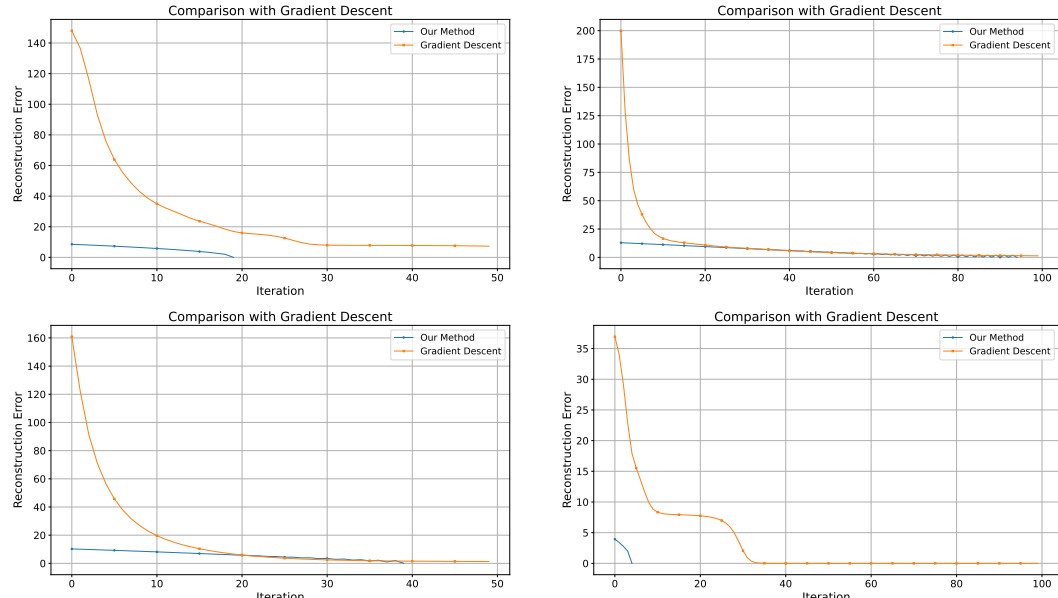

Figure 11: Comparison with gradient descent. Iteration refers to a gradient step for gradient descent and a Householder update for our method. We present 4 cases: $(n, m) = (500, 20), (100, 95), (100, 40), (100, 5)$, illustrating various scenarios. In each approach, our method reaches $0$ error in precisely $m$ iterations. However, owing to the non-convex loss landscape, such guarantees cannot be offered for gradient descent. On some occasions (plots 2 and 3), it gets close to $0$-error. Convergence is reached in plot 4, but much later than with our algorithm, and it does not occur at all in plot 1. The loss pattern in plot 4 is also somewhat unpredictable.

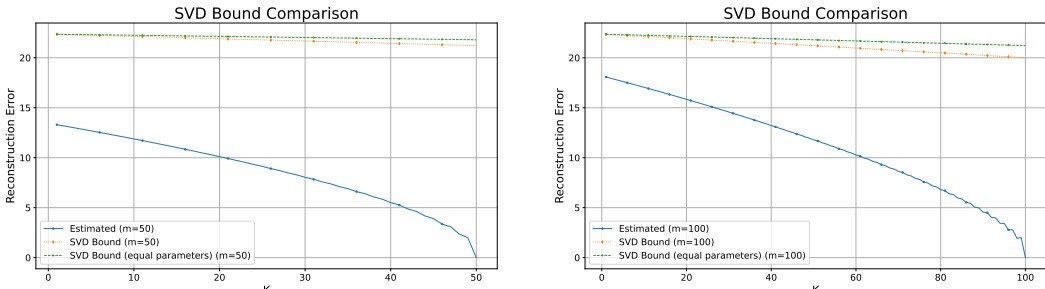

Figure 12: Comparison of our bound with SVD for two cases- same rank as our number of Householders used and for the same number of parameters. $K$ is the number of Householders used in our approximation while $m$ is the number of Householders used in the ground truth product. Our method comprehensively outperforms the SVD based approximations. We observe this across $m$ values.

Note that this loss function is non-convex in its parameters- $\{\mathbf{v}_1, \mathbf{v}_2, \cdots \mathbf{v}_m\}$, the set of Householder vectors defining the Householder matrices. We found that a learning rate of $0.03$ works best for most $(n, m)$ pairs- and is thus our choice for the experiments. We present our results in Figure 11.

### D.2.4 COMPARISON WITH SVD

For a given rank $K$, a rank-$K$ SVD yields an error of $\sqrt{n - K}$, since for an orthogonal matrix, all singular values are 1. Thus, the Frobenius norm error is only dependent on the singular values that have been changed to $0$ for the approximation. We present our results for bound comparison with SVD in Figure 12. Note that we use our exact bound (from equations 18 and 19) here.

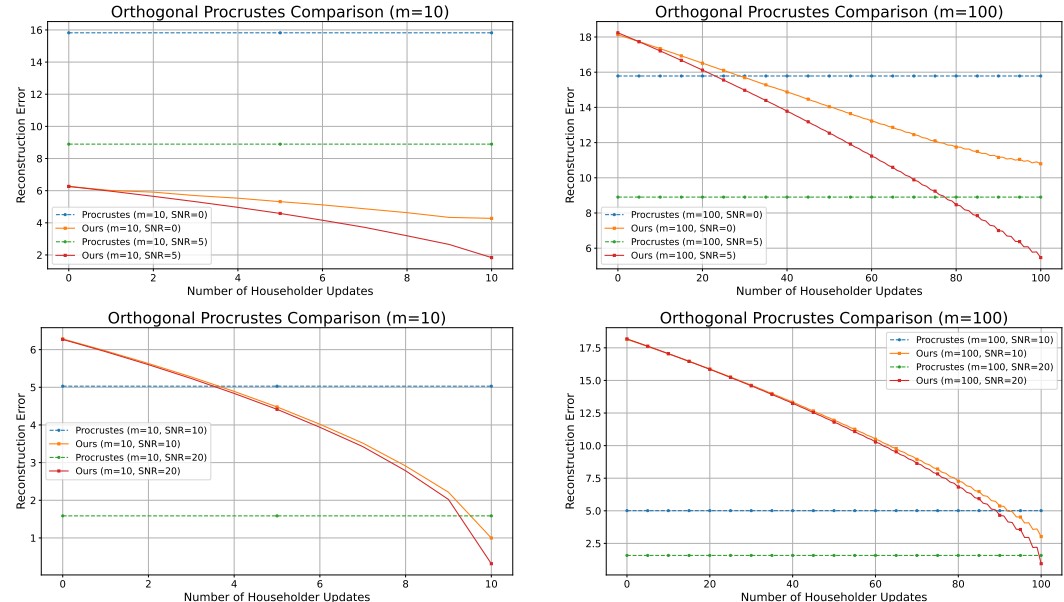

Figure 13: Comparison with the Orthogonal Procrustes solution under noise. Our proposed method provides a smoother tradeoff between error and the number of reflectors used (for $m = 10$ and $m = 100$), at SNR=$(0, 5), (10, 20)$ dB and dimension $n = 500$. Our method ends up being close to the ground truth orthogonal matrix than the Orthogonal Procrustes solution. In fact, the larger the noise, the better our method performs.

## D.3 DENOISING EXPERIMENTS

We simulate the denoising scenario by constructing a ground-truth orthogonal matrix $\mathbf{V}^* \in \mathbb{R}^{n \times n}$ (with $n = 500$) as a product of $m$ Householder matrices, each generated from a normalized Gaussian vector. A scaled Ginibre matrix is then added to create the noisy matrix:

$$\mathbf{V} = \mathbf{V}^* + \frac{1}{n^\alpha}\mathbf{N}.$$

We run Algorithm 2 for exactly $m$ iterations and evaluate the reconstruction error

$$\text{Reconstruction Error} = \|\hat{\mathbf{V}} - \mathbf{V}^*\|_F,$$

across different $m$ and SNR values. Here, Signal to Noise ratio, or SNR, is defined as the ratio of signal power to noise power:

$$\text{SNR} = \frac{\|\mathbf{V}^*\|^2}{\|(1/n^\alpha)\mathbf{N}\|^2}. \tag{64}$$

SNR(dB) $= 10 \log_{10}(\text{SNR})$. Figure 4 shows that the error decreases consistently with increasing SNR. For a given $\alpha$, the SNR(dB) can be expressed as

$$\text{SNR(dB)} = 10(2\alpha - 1) \log_{10}(n).$$

## D.3.1 COMPARISON WITH ORTHOGONAL PROCRUSTES

We also compare our method with the Orthogonal Procrustes projection for non-orthogonal matrices (see Figure 13). This is the best orthogonal projection in terms of proximity to the input for any non-orthogonal matrix. However, this is generally a dense matrix and requires $\mathcal{O}(n^2)$ storage. Furthermore, naïvely applying this method doesn't necessarily give the optimal solution (see Figure 14). Moreover, one also does not benefit from the computational gain of using Householder reflectors for matrix vector multiplication.

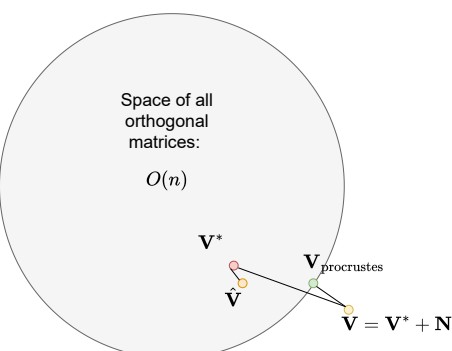

Figure 14: Illustrating why our method outperforms the Orthogonal Procrustes solution. Our goal is to recover the ground truth matrix $\mathbf{V}^*$, while the Orthogonal Procrustes solution finds the matrix closest to the noisy input matrix in the space of all orthogonal matrices. Since we haven't proved that our method is optimal for all choices of Householder vectors given a noisy input matrix, we depict $\hat{\mathbf{V}}$ as not necessarily being on the shortest path from $\mathbf{V}^*$ to $\mathbf{V}$. We conclude section C.4 with specific mathematical conditions under which the Orthogonal Procrustes solution should be adopted for Ginibre noise.

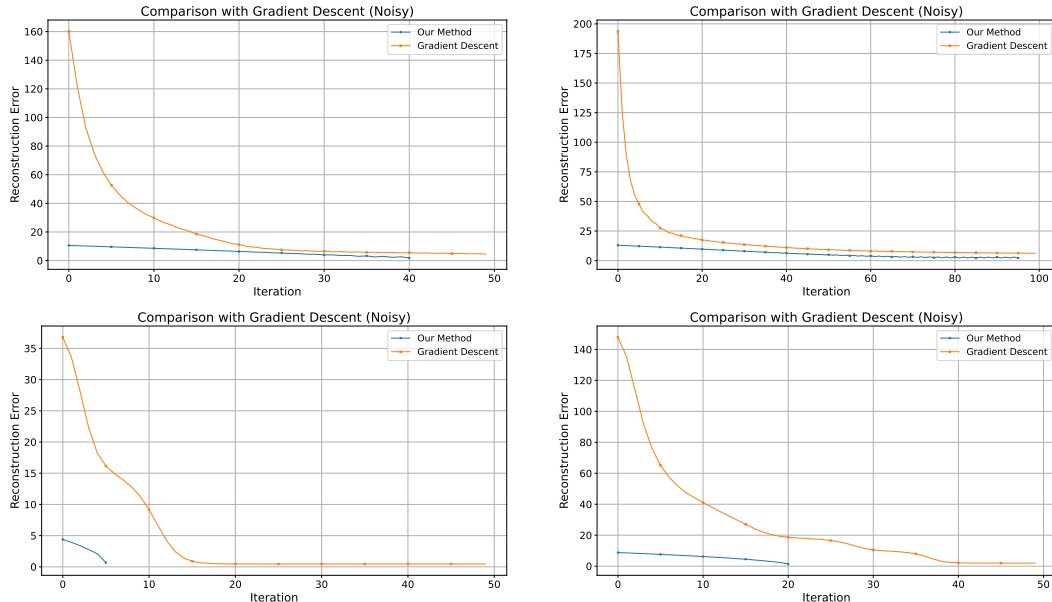

Figure 15: Comparison with gradient descent for denoising. Iteration refers to a gradient step for gradient descent and a Householder update for our method. We present 4 cases: $(n, m) = (500, 20), (100, 95), (100, 40), (100, 5)$, illustrating various scenarios. We observe trends similar to those in 11, but on this occasion, gradient descent doesn't converge in any case since the loss its optimizing is with respect to the noisy input matrix, and it cannot reach as close to the ground truth solution as our method.

### D.3.2 COMPARISON WITH GRADIENT BASED OPTIMIZATION

We repeat the same procedure introduced in Section D.2.3 and present our results in Figure 15.

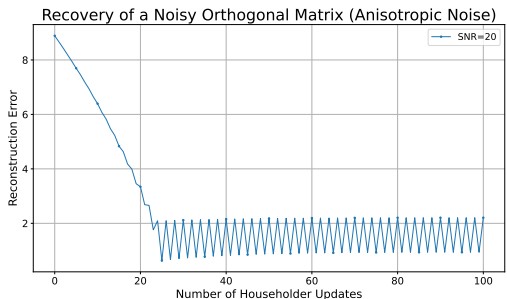 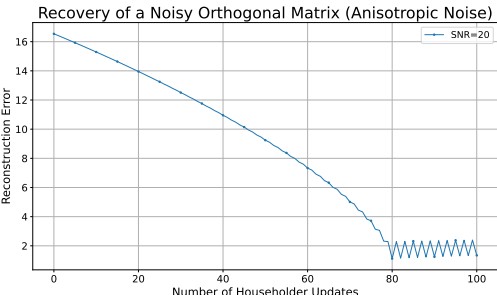

Figure 16: Reconstruction error for anisotropic noise $m = 25, n = 100$ and $m = 80, n = 500$. The baseline SNR used is 20dB. Notice the similarity in the error pattern from Ginibre noise. The columns' standard deviations are scaled by factors ranging from $0.5$ to $2$. Increasing these factors by a larger value results in slower convergence, as expected.

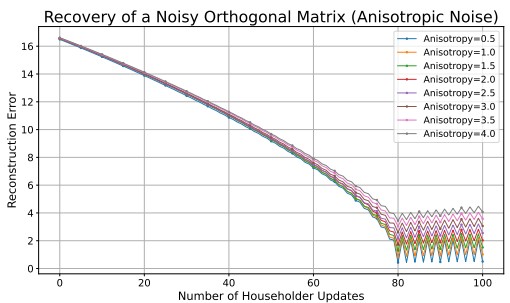 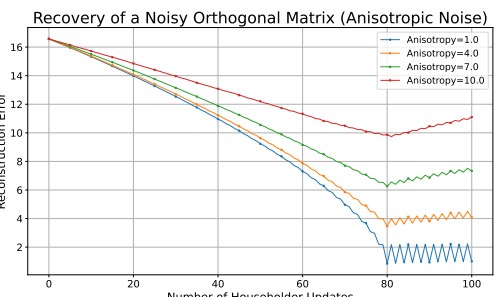

Figure 17: Reconstruction error for anisotropic noise ($m = 80$, $n = 500$, baseline SNR 20 dB). Error increases with anisotropy, with recovery patterns similar to that of Ginibre-noise.

### D.3.3 ANISOTROPIC NOISE

We can also add anisotropic noise and then use our denoising algorithm. Though we cannot provide theoretical guarantees here, as long as the scaling of the noise along each column isn't significantly larger than the chosen base SNR, we observe similar denoising patterns- Figure 16. To generate the noise, we use the following method: anisotropic noise is generated by first setting a baseline Gaussian noise level consistent with the desired signal-to-noise ratio (SNR), computed from the Frobenius norm of the clean orthogonal matrix. A vector of anisotropy factors is then introduced, assigning different variance multipliers to each column. Standard normal noise is sampled and scaled column-wise by these factors (range: $0.5 - 2$), producing direction-dependent perturbations that are finally added to the orthogonal matrix.

We provide additional comparison plots in Figure 17 to analyze how the level of anisotropy affects the reconstruction error. We observe that an increase in the anisotropy factors shows a gradual increase in reconstruction error. In the first plot in Figure 17, the factors increase at increments of $0.5$. The gentle reduction in error is visible from the clustered nature of the plot. To better analyze when the method breaks down, we use much larger anisotropy factors in the second plot.

### D.3.4 OTHER STRUCTURED NOISE MATRICES

Finally, to validate the utility of our algorithm in a more general setting, we test our method by adding various kinds of structured noise matrices to the original orthogonal matrix and applying our denoising algorithm. The noise matrices are generated as follows: for low rank noise, for a fixed rank $r$, two random matrices, $\mathbf{A}, \mathbf{B}$ of dimensions $n \times r$ are generated. We then obtain the final low rank noise as $c\mathbf{A}\mathbf{B}^\top$, where $c$ is an appropriate scaling factor to ensure that the noise matrix added corresponds to the desired SNR. For uniform noise, we generate the noise matrix by sampling each entry independently from a uniform distribution on $[-0.5, 0.5]$ and appropriately scaling the matrix to meet the desired SNR. For multiplicative noise, we create a Bernoulli mask with various masking

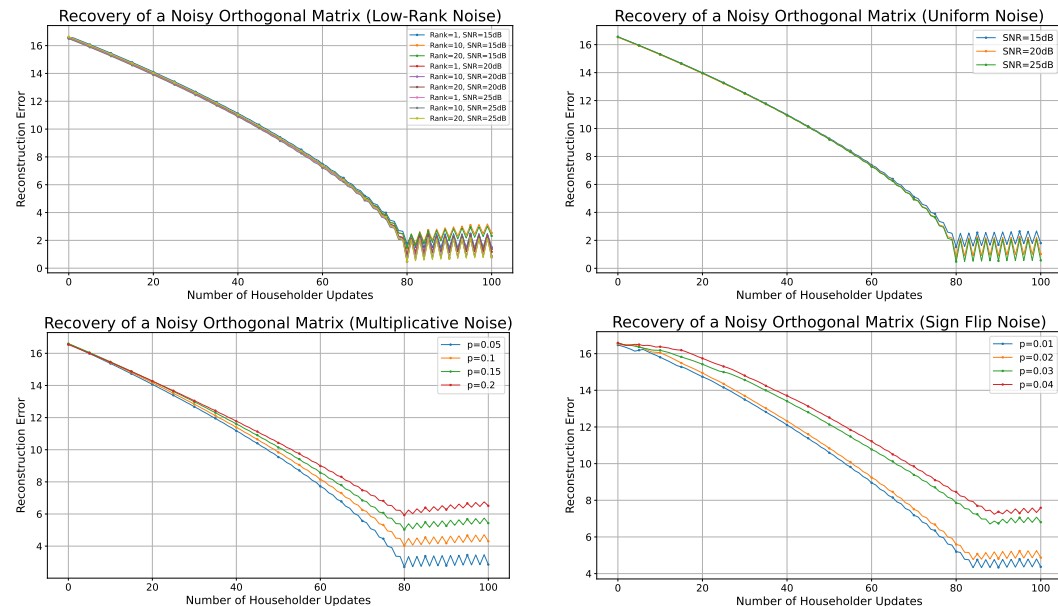

Figure 18: Reconstruction error vs. iterations for low rank noise, uniform noise, multiplicative noise, and sign flip noise ($m = 80$, $n = 500$, target SNR 20 dB). In general, we observe that the error patterns are very similar to the Ginibre noise case. The reconstruction error gets larger with the increase in distortion of the spectrum. This is to be expected since our algorithm relies on the minimum eigenvalue and its corresponding eigenvector. Note that the method is, in general, pretty robust, unless the spectrum changes by large values, such as say, greater than 20% in sign flip noise with $p = 0.03, 0.04$

probability values $p(0) = p$, similar to values one might encounter while performing drop out. We then multiply the orthogonal matrix element-wise with this mask, to effectively zero out a fraction of the entries. We observe that though the reconstruction error is not as low as it is in other case, the deviation isn't very large and the pattern of reduction in error is also similar to other cases. Finally, we also try sign-flip noise, changing the sign of $p$ fraction of the original matrix. We analyze various values of $p$ here. The results obtained are showcased in Figure 18.

**Summary.** We have shown that Algorithm 2 effectively denoises both single and multi-reflector orthogonal matrices. Theoretical guarantees, derived using eigenvector perturbation bounds and projection properties, indicate that the recovered matrix remains close (in Frobenius norm) to the ground truth under standard noise models, assuming the number of reflections is known. These results are supported empirically in Figures 4 and 5.

Our method leverages the structure of $\mathcal{H}_m$ to achieve accurate and stable recovery in the presence of noise. Compared to prior decomposition techniques, it yields consistently lower reconstruction error and more predictable behavior, making it suitable for denoising orthogonal matrices in practice.

The ability to project noisy matrices onto $\mathcal{H}_m$ with provable error control suggests a broader role for structured orthogonal parameterizations as implicit regularizers. This has potential relevance for tasks in signal processing and machine learning where robustness and efficiency are critical, such as model compression, orthogonal initialization, or constrained optimization involving orthogonal weights.

**Limitations.** Our method provides provable guarantees and strong empirical performance, but our focus has been primarily on real orthogonal matrices and their noisy non-orthogonal variants. However, complex matrices such as unitary matrices don't follow such guaranteed compression rules and remain to be explored.

Table 7: Copy Task on expRNN with Householder approximation. We compare performance after 50 iterations (partial training to a loss of 0.1697) and 2000 iterations (full convergence). For each $K$, we replace the trained recurrent matrix with an $K$-dependent approximation and measure the reconstruction loss (post-hoc compression, zero-shot). We observe graceful degradation in performance as $K$ decreases for our method. $K$ represents the number of Householders being multiplied for both our approach and the QR decomposition method, while it represents the rank of the truncated SVD for the SVD method.

| K | Loss @ 50 iters | | | Loss @ 2000 iters | | |
|---|---|---|---|---|---|---|
| | **Ours** | **SVD** | **QR** | **Ours** | **SVD** | **QR** |
| 10 | 13.3868 | 0.1533 | 9.0322 | 45.9371 | 6.9551 | 33.4273 |
| 25 | 13.8214 | 0.7868 | 9.7512 | 48.1242 | 4.5416 | 33.0858 |
| 50 | 11.8672 | 0.6859 | 5.9722 | 44.1217 | 2.9853 | 26.5186 |
| 100 | 9.8905 | 4.8487 | 5.5317 | 36.0210 | 0.2472 | 20.8665 |
| 150 | 3.8314 | 15.2670 | 4.4683 | 24.1278 | 0.1512 | 9.5570 |
| 160 | 2.8279 | 14.2986 | 6.0883 | 22.1716 | 0.1713 | 6.6875 |
| 170 | 3.6614 | 15.1026 | 13.4529 | 21.5528 | 0.1834 | 3.6667 |
| 180 | 3.2408 | 13.8267 | 15.7230 | 11.4483 | 0.1741 | 3.1566 |
| 181 | 1.7055 | 13.8862 | 5.9534 | 8.8230 | 0.3164 | 3.2020 |
| 182 | 2.1839 | 13.8459 | 11.0884 | 8.8181 | 0.3211 | 3.1045 |
| 183 | 1.4878 | 13.8475 | 14.1961 | 8.3456 | 0.3627 | 2.9236 |
| 184 | 1.4542 | 13.2771 | 5.3753 | 8.3758 | 0.3137 | 2.5228 |
| 185 | 0.9960 | 12.5786 | 1.6719 | 5.4944 | 0.2626 | 2.9947 |
| 186 | 1.2775 | 0.5761 | 1.0027 | 3.2823 | 0.1519 | 2.0486 |
| 187 | 1.0176 | 0.1259 | 0.8837 | 3.0592 | 0.1135 | 0.9540 |
| 188 | 0.4657 | 0.1252 | 0.8690 | 2.7093 | 0.0474 | 1.3293 |
| 189 | 0.4444 | 0.1446 | 8.8687 | 3.8019 | 0.0454 | 8.4687 |
| **190** | **0.1697** | **0.1697** | **1.7166** | **0.000039** | **0.000032** | **4.5705** |

## D.4 ABLATION STUDY

To evaluate the effectiveness and generality of our Householder-based compression approach, we conduct a series of ablation studies across diverse architectures and tasks. The evaluation here is zero-shot for sections D.4.1, D.4.2- we directly replace the trained matrices with their compressed versions, and we perform full training in section D.4.3. More details can be found in the respective sections.

### D.4.1 EXPRNN

We begin with the Copy Task using expRNN (which draws ideas from Lie groups for orthogonal parametrization), replacing the trained recurrent matrix with products of a varying number of Householder matrices K. We evaluate both early-stage (after 50 iterations) and converged (2000 iterations) training losses to measure how well the compressed matrices preserve learned dynamics. Note that the trained recurrent matrices here are of dimension $n \times n = 190 \times 190$. Following this, we test the denoising capabilities of our algorithm. We add noise (SNR=20) to the orthogonal recurrent matrix and then evaluate the performance of our algorithm by denoising this matrix and evaluating the performance on the Copy Task.

Next, we assess performance on sequential MNIST, where we train the model with the default settings specified in (Lezcano-Casado & Martınez-Rubio, 2019) for 2 and 25 epochs to capture both short-term and long-term training effects and replace the trained recurrent matrices with K-Householder approximations to measure the effectiveness of compression. The trained recurrent matrices here are of dimension $n \times n = 170 \times 170$.

Some observations worth noting: Unlike our method, SVD doesn't provide any computational benefits, even though its error in the trained-to-convergence case is very low. Furthermore, notice that when training isn't complete, the error can change abruptly to much larger values than the general trend, indicating the inherent instability of this method. The QR method is not as capricious, but

Table 8: Copy Task on expRNN with Householder approximation for denoising (we add noise to the trained orthogonal matrix to analyze denoising performance). We compare performance after 50 iterations (partial training to a loss of 0.1697) and 2000 iterations (full convergence). For each $K$ (defined exactly as in Table 7), we replace the trained recurrent matrix with a $K$-dependent approximation and measure the reconstruction loss (post-hoc compression, zero-shot). We observe graceful degradation in performance as $K$ decreases for our method. The loss before denoising the noisy version of the trained matrix (SNR=20) is close to 56k at 50 iterations and close to 110k at 2000 iterations.

| K | Loss @ 50 iters | | | Loss @ 2000 iters | | |
|---|---|---|---|---|---|---|
| | **Ours** | **SVD** | **QR** | **Ours** | **SVD** | **QR** |
| 10 | 13.2108 | 0.9544 | 9.4055 | 45.7067 | 0.0850 | 34.0982 |
| 25 | 14.4165 | 0.8536 | 10.5773 | 48.1055 | 0.0709 | 34.2047 |
| 50 | 11.9550 | 0.8265 | 6.1527 | 44.1801 | 0.0746 | 26.2148 |
| 100 | 10.8153 | 0.1875 | 5.0683 | 36.9808 | 0.2043 | 19.2336 |
| 150 | 7.1198 | inf | 4.7622 | 26.2717 | inf | 9.4518 |
| 160 | 3.4587 | inf | 13.2568 | 24.1295 | inf | 6.9596 |
| 170 | 4.0186 | inf | 33.5543 | 20.5581 | inf | 4.1428 |
| 180 | 2.8691 | inf | 3.4211 | 11.3013 | inf | 3.7785 |
| 181 | 3.5417 | inf | 1.8895 | 8.5601 | inf | 3.8611 |
| 182 | 2.9563 | inf | 4.9159 | 9.2709 | inf | 3.4999 |
| 183 | 2.9294 | inf | 5.5341 | 7.7196 | inf | 3.4371 |
| 184 | 1.5613 | inf | 1.6214 | 8.0029 | inf | 3.1678 |
| 185 | 1.4605 | inf | 2.0677 | 80.1877 | inf | 3.7596 |
| 186 | 1.3353 | inf | 1.2471 | 2.0617 | inf | 2.5600 |
| 187 | 1.0243 | inf | 19.5313 | 2.2971 | inf | 1.0206 |
| 188 | 0.3631 | inf | 1.8388 | 2.2060 | inf | 1.0847 |
| 189 | 0.3682 | inf | 2.0043 | 36.8791 | inf | 2.7511 |
| **190** | **0.2777** | **inf** | **1.8259** | **0.1782** | **inf** | **3.7788** |

significant deviations are still observed for multiple values of $K$. On the other hand, our method is structured and explainable, adapting to the trained parameters and exhibiting a graceful decrease in error as $K$ increases (Table 7).

Secondly, in the noisy case, SVD approximates the noisy input matrices, which, anyway, perform poorly in the network. This further explains why at low ranks, the approximation error isn't high, whereas as the rank increases, noise is approximated better, leading to the error blowing up. The QR method is far more obedient, though its error is still much higher than ours.

On the abrupt accuracy changes for nearby $K$ values for Sequential MNIST after sufficient training: In Table 9, the abrupt jumps in accuracy for certain $K$ values are a natural consequence of the underlying structure of the trained orthogonal matrix. When the true $\mathbf{V}$ lies in a product space of a large number of Householder reflectors, approximating it with a highly compressed product (small $K$) removes a significant portion of the relevant subspace information, leading to near-random accuracy (around $10\%$ for MNIST). As $K$ approaches the effective number of reflectors needed to span the subspace containing the task-relevant transformations, the approximation quality crosses a threshold where the model can suddenly recover most of the original performance, producing the observed steep jumps rather than smooth interpolation. The trends are much smoother for the copy task and the Sequential MNIST task after insufficient training. All of this solely relies on the spaces in which the trained orthogonal matrices lie. Note that a similar jump is observed for SVD as well, indicating that it relates to some underlying property of the trained matrix subspaces rather than the approximations themselves, especially for the case where sufficient training hasn't been performed.

For the Sequential MNIST task, SVD in general outperforms our method. From the observed trends it seems like the trained matrices are present in a slightly lower rank manifold (accuracy close to $90\%$ at $K = 160$) rather than a low-product-Householder space. The QR decomposition method performs significantly worse than the former two.

Table 9: Accuracy (%) on pixel-by-pixel MNIST using expRNN with approximated recurrent matrices. Accuracy shown after 2 and 25 training epochs for each $K$. Testing is zero-shot after replacing the trained matrices with their approximated versions. The variance is close to $0.3\%$ and $0.4\%$ for 2 and 25 epochs respectively.

| K | Accuracy (%) @ 2 epochs | | | Accuracy (%) @ 25 epochs | | |
|---|---|---|---|---|---|---|
| | **Ours** | **SVD** | **QR** | **Ours** | **SVD** | **QR** |
| 25 | 9.80 | 6.53 | 9.76 | 10.63 | 10.71 | 10.20 |
| 50 | 9.80 | 9.15 | 10.26 | 11.12 | 8.95 | 11.07 |
| 100 | 10.09 | 18.83 | 9.74 | 9.35 | 17.58 | 10.29 |
| 130 | 10.09 | 16.68 | 9.83 | 13.08 | 21.92 | 10.62 |
| 140 | 10.09 | 25.60 | 9.81 | 11.10 | 30.02 | 11.49 |
| 150 | 10.35 | 32.55 | 9.80 | 21.79 | 69.85 | 11.77 |
| 160 | 10.42 | 45.75 | 9.98 | 18.09 | 90.14 | 16.12 |
| 161 | 24.33 | 46.08 | 11.55 | 20.45 | 91.13 | 10.34 |
| 162 | 27.44 | 50.58 | 9.97 | 20.12 | 92.42 | 10.34 |
| 163 | 22.32 | 50.71 | 9.97 | 20.17 | 91.94 | 10.34 |
| 164 | 30.49 | 54.53 | 9.97 | 23.18 | 92.02 | 10.35 |
| 165 | 31.37 | 54.39 | 10.12 | 34.08 | 91.85 | 10.35 |
| 166 | 31.86 | 55.26 | 10.11 | 28.09 | 94.63 | 10.35 |
| 167 | 31.35 | 31.61 | 10.72 | 31.06 | 94.67 | 10.36 |
| 168 | 73.69 | 87.49 | 10.98 | 30.11 | 95.18 | 10.36 |
| 169 | 74.88 | 90.43 | 10.47 | 33.26 | 97.27 | 10.45 |
| 170 | 88.47 | 90.74 | 10.17 | 96.43 | 97.24 | 8.90 |

### D.4.2 VIT

We move beyond RNNs and apply our method to Vision Transformers on CIFAR-10. Specifically, we replace the QKV weights of a single attention block in a ViT-Tiny model with Householder-compressed versions and measure classification accuracy at different values of K. Across all settings, we observe a consistent trend of graceful degradation in performance as K decreases, validating the utility of our approach as a controllable approximation method for structured compression. Note that since the ViT attention matrices aren't orthogonal, this can be seen as a denoising approximation, and thus, after replacement, even with $K = 96$, exact recovery isn't possible, unlike the Copy Task and the Sequential MNIST task. Moreover, as the number of blocks replaced increases, the accuracy also increases. On replacing all blocks, the accuracy languishes around the 14% mark. We believe this is because the parameters themselves don't lie on the orthogonal manifold. However, we still try to perform an approximation for all blocks simultaneously, compounding the effect of the difference in parameters in each block, resulting in performance loss. Quantitatively, one might justify this precipitous drop by analyzing how far the attention matrices are from being orthogonal. We present this justification in Table 10.

Table 10: The distance (in Frobenius norm) from identity of the various attention matrices in the ViT under consideration- $\|\mathbf{M}^\top \mathbf{M} - \mathbf{I}\|_F$, where $\mathbf{M}$ is one of the three attention matrices.

| | 0 | 1 | 2 | 3 | 4 | 5 |
|---|---|---|---|---|---|---|
| Q | 26.10 | 24.86 | 21.98 | 21.93 | 26.45 | 28.85 |
| K | 39.17 | 27.90 | 26.49 | 23.21 | 27.05 | 27.62 |
| V | 8.11 | 19.40 | 19.25 | 18.08 | 18.44 | 18.71 |

Next, we test our compression on a larger dataset and a bigger model. We use the ViT small with a patch size of 32 and embedding dimension 384, depth 12, and 6 heads, and test our compression on the Tiny ImageNet. The images are resized to meet the $224 \times 224$ resolution requirements for the ViT small model. We use the pretrained ImageNet model, with a batch size of 512, optimizer AdamW (lr $= 3 \times 10^{-3}$, weight decay 0.01), RandAugment (`rand-m9-mstd0.5-inc1`), and stochastic depth rate 0.5, and the mimetic initialization. We train only for 100 epochs and get a baseline accuracy of $68.93\%$.

Table 11: Accuracy (%) on CIFAR-10 after replacing only one attention block in ViT-Tiny (patch size 2, depth 6, and embedding dimension 96 using mimetic initialization (Trockman & Kolter, 2023)) with Householder-compressed QKV weights. Results are reported for different values of $K$ (number of reflections). The remaining blocks are untouched. Performance gracefully degrades as $K$ decreases. Note that the attention matrices in the ViT are not orthogonal. Therefore, the model with replacement, even for K= 96, doesn't reach the roughly $87.22\%$ accuracy achieved by the model in which none of the blocks are replaced. The variance is close to $0.5\%$ across blocks and K values. We do not add SVD here for comparison since it is an approximation for arbitrary (non-orthogonal) matrices, so it will provide a much better approximation than our orthogonal methods.

| Method | Block | K=96 | K=84 | K=72 | K=60 | K=48 | K=36 | K=24 | K=12 | K=1 |
|---|---|---|---|---|---|---|---|---|---|---|
| Ours | 0 | 83.35 | 83.31 | 83.30 | 83.15 | 82.27 | 81.32 | 79.68 | 79.40 | 80.37 |
| | 1 | 73.22 | 74.22 | 73.78 | 72.66 | 71.86 | 71.79 | 70.62 | 69.32 | 68.37 |
| | 2 | 74.52 | 74.76 | 74.32 | 73.54 | 72.79 | 71.44 | 70.33 | 69.06 | 67.56 |
| | 3 | 78.24 | 79.28 | 78.76 | 78.45 | 77.99 | 76.90 | 75.43 | 73.92 | 73.04 |
| | 4 | 78.87 | 78.91 | 78.05 | 77.07 | 75.31 | 73.48 | 73.39 | 71.95 | 69.88 |
| | 5 | 79.34 | 79.40 | 79.13 | 77.80 | 75.62 | 72.80 | 69.47 | 65.89 | 62.46 |
| QR | 0 | 79.64 | 79.34 | 79.34 | 78.86 | 79.05 | 80.40 | 80.71 | 80.44 | 80.16 |
| | 1 | 70.12 | 70.61 | 70.61 | 69.60 | 69.61 | 70.16 | 70.20 | 70.56 | 69.49 |
| | 2 | 69.52 | 69.24 | 69.71 | 70.06 | 70.59 | 69.36 | 68.03 | 68.57 | 67.72 |
| | 3 | 76.21 | 76.51 | 75.47 | 75.04 | 75.10 | 75.47 | 75.90 | 76.10 | 76.82 |
| | 4 | 70.29 | 71.54 | 71.26 | 71.61 | 70.87 | 69.92 | 70.05 | 67.53 | 68.03 |
| | 5 | 66.60 | 66.47 | 65.42 | 64.03 | 64.00 | 63.47 | 62.23 | 62.51 | 62.85 |

Table 12: Accuracy (%) on Tiny ImageNet after replacing only one attention block in ViT-Small (patch size 32, depth 12, and embedding dimension 384 using mimetic initialization (Trockman & Kolter, 2023)) with Householder-compressed QKV weights. Results are reported for different values of $K$ (number of reflections). The remaining blocks are untouched. Performance gracefully degrades as $K$ decreases for all block indices, though the drop is significant for block index 0. When we try to replace multiple blocks, the accuracy drops (for example, on compressing block indices 1 and 2, the accuracy is close to $45\%$)- since the parameter matrices are not orthogonal. Also, as a consequence, the model with replacement, even for $K = 384$, doesn't reach the accuracy achieved by the model in which none of the blocks are replaced. The variance is close to $0.2\%$ across blocks and K values.

| Block | K=384 | K=336 | K=288 | K=240 | K=192 | K=144 | K=96 | K=48 | K=1 |
|---|---|---|---|---|---|---|---|---|---|
| 0 | 23.43 | 23.70 | 23.50 | 23.35 | 22.74 | 22.07 | 21.56 | 20.66 | 19.56 |
| 1 | 65.39 | 65.36 | 65.35 | 65.28 | 65.15 | 65.02 | 64.86 | 64.46 | 63.88 |
| 2 | 62.80 | 62.81 | 62.79 | 62.79 | 62.73 | 62.65 | 62.60 | 62.56 | 62.53 |
| 3 | 66.66 | 66.66 | 66.64 | 66.60 | 66.65 | 66.52 | 66.49 | 66.47 | 66.40 |
| 4 | 64.88 | 64.89 | 64.87 | 64.82 | 64.78 | 64.65 | 64.65 | 64.58 | 64.60 |
| 5 | 65.18 | 65.14 | 65.08 | 65.12 | 65.05 | 65.00 | 64.98 | 64.96 | 64.86 |
| 6 | 65.60 | 65.55 | 65.63 | 65.54 | 65.43 | 65.37 | 65.48 | 65.47 | 65.45 |
| 7 | 65.76 | 65.73 | 65.71 | 65.71 | 65.66 | 65.61 | 65.58 | 65.48 | 65.50 |
| 8 | 65.64 | 65.67 | 65.62 | 65.56 | 65.48 | 65.49 | 65.57 | 65.44 | 65.47 |
| 9 | 67.30 | 67.24 | 67.28 | 67.20 | 67.13 | 67.12 | 67.03 | 66.95 | 66.96 |
| 10 | 67.20 | 67.22 | 67.24 | 67.24 | 67.25 | 67.26 | 67.20 | 67.15 | 67.02 |
| 11 | 68.06 | 68.06 | 68.09 | 68.02 | 68.04 | 67.93 | 67.83 | 67.78 | 67.77 |

### D.4.3 ViT- Initialization Compression

Up to this point, in both the RNNs and ViT, the evaluation was zero-shot. Now, we test our approach during training. We train the same ViT model described above by compressing the mimetic initialization weights given by Trockman & Kolter (2023) for the query and key matrices for all heads and blocks while keeping the value and projection matrices the same as the original initialization. Training is performed for 100 epochs, as is done in the original paper. We used no class embedding, a sine positional encoding, and global average pooling. Training is performed for 100 epochs with AdamW (lr $= 3 \times 10^{-3}$, weight decay 0.01), batch size 512, RandAugment (`rand-m9-mstd0.5-inc1`), and stochastic depth rate 0.5. We do not constrain the training blocks to the orthogonal space in this experiment- Note that the accuracy without any compression is 87.22%. The training results are summarized in Table 13. If we compress the key and projection matrices as well to the space of one Householder reflector, the accuracy drops to 83.84%- on using $K = 96$, the accuracy is 86.84%.

We observe that we can train with these compressed Householder initialization while staying very close to the accuracy achieved by the mimetic initialization. In some cases, we get the same accuracy as the mimetic initialization. Furthermore, we do not need to store the initialization weight matrices, since just one Householder compression is also very close to the best value. One can store just the single $\mathcal{O}(n)$ sized Householder vectors corresponding to the compressed Householder blocks.

We hypothesize that just a single Householder compression seems to be sufficient to retain accuracy because of the structure of such matrices- they are roughly close to the identity matrix for careful choices of the Householder vector. Since the mimetic initialization was precisely modeled as an identity matrix plus noise, our compression performs very well even for low $k$.

Table 13: Accuracy (%) on CIFAR-10 after replacing the $Q$ and $K$ blocks of the mimetic initialization with Householder-compressed weights and performing unconstrained training (parameters not restricted to the orthogonal manifold) with the aforementioned modified initialization. The model is ViT-Tiny (patch size 2, depth 6, and embedding dimension 96. Results are reported for different values of $K$ (number of reflections). The remaining blocks are untouched. The variance is close to 0.05% for varying $K$. Note that training using an arbitrary Householder matrix gives an accuracy of 83.71%

| K | Accuracy (%) |
|---|---|
| 1 | 87.11 |
| 8 | 87.22 |
| 16 | 87.18 |
| 24 | 86.70 |
| 32 | 86.98 |
| 40 | 87.00 |
| 48 | 86.88 |
| 56 | 86.92 |
| 64 | 87.02 |
| 72 | 87.18 |
| 80 | 87.11 |
| 88 | 86.76 |
| 96 | 87.15 |

# E  ADDITIONAL DISCUSSION

## E.1  SCOPE AND PRACTICALITY OF APPROXIMATION GUARANTEES

The theoretical approximation guarantees depend on performing eigendecomposition at each iteration, which can be computationally expensive for very high-dimensional or ill-conditioned matrices. In practice, stable and scalable eigensolvers—such as iterative methods, randomized algorithms, or subspace tracking approaches—can be incorporated to mitigate computational costs while maintaining approximation quality.

Numerical Stability and Floating-Point Issues: A deeper treatment of numerical stability and finite-precision effects is essential for practical adoption. Sometimes, though rare, these prevent convergence. For example, for $n = 500$, $m = 400$ may not converge. This generally doesn't happen when $m \ll n$, but becomes a possibility for larger $m$. If necessary, for the version without Lanczos' algorithm, an Orthogonal Procrustes step can be added to improve stability. Although not explored in detail here, we note that the monotone decrease in error (parity-wise) and eigenspace projections provides inherent noise robustness in moderate dimensions. Future work will rigorously explore efficient approximation of eigenspaces suited for large-scale and noisy settings.

## E.2  DEMONSTRATION OF ORTHOGONAL MATRIX RECOVERY

We demonstrate how recovery works by presenting the eigenvalues in a run of Algorithm 1 for an orthogonal matrix $\mathbf{V} \in \mathcal{H}_9$ for $n = 10$. The columns contain the eigenvalues in sorted order from left to right ($\lambda_1 \leq \lambda_2 \leq \cdots \leq \lambda_{10}$).

Table 14: Demonstration of orthogonal matrix recovery: the columns contain the eigenvalues in sorted order from left to right.

| Iter | $\lambda_1$ | $\lambda_2$ | $\lambda_3$ | $\lambda_4$ | $\lambda_5$ | $\lambda_6$ | $\lambda_7$ | $\lambda_8$ | $\lambda_9$ | $\lambda_{10}$ |
|------|------|------|------|------|------|------|------|------|------|------|
| 1 | -1 | -0.8035 | -0.8035 | -0.1892 | -0.1892 | 0.0194 | 0.0194 | 0.8913 | 0.8913 | 1 |
| 2 | -0.8035 | -0.8035 | -0.1892 | -0.1892 | 0.0194 | 0.0194 | 0.8913 | 0.8913 | 1 | 1 |
| 3 | -1 | -0.1892 | -0.1892 | 0.0194 | 0.0194 | 0.8913 | 0.8913 | 1 | 1 | 1 |
| 4 | -0.1892 | -0.1892 | 0.0194 | 0.0194 | 0.8913 | 0.8913 | 1 | 1 | 1 | 1 |
| 5 | -1 | 0.0194 | 0.0194 | 0.8913 | 0.8913 | 1 | 1 | 1 | 1 | 1 |
| 6 | 0.0194 | 0.0194 | 0.8913 | 0.8913 | 1 | 1 | 1 | 1 | 1 | 1 |
| 7 | -1 | 0.8913 | 0.8913 | 1 | 1 | 1 | 1 | 1 | 1 | 1 |
| 8 | 0.8913 | 0.8913 | 1 | 1 | 1 | 1 | 1 | 1 | 1 | 1 |
| 9 | -1 | 1 | 1 | 1 | 1 | 1 | 1 | 1 | 1 | 1 |
| 10 | 1 | 1 | 1 | 1 | 1 | 1 | 1 | 1 | 1 | 1 |

### E.2.1  ON THE CHOICE OF EIGENVECTOR

We choose the eigenvector corresponding to the minimum eigenvalue as a greedy step because it produces the highest decrease (or lowest increase) in error at each iteration. The algorithm would converge even if we chose an eigenvector corresponding to any other eigenvalue. However, it would be much slower in general (since the eigenvector choice is now random) and somewhat defeat the purpose. Guaranteed convergence would still occur at the $n^{\text{th}}$ iteration.

Note that when an eigenvalue has multiplicity greater than one, Algorithm 1 may choose any eigenvector within the eigenspace; all such choices remain valid since the reflector eliminates the entire eigenspace component, and convergence remains guaranteed.

## E.3  CHOOSING $m$ DURING DENOISING

The primary intuition behind the stopping condition is that after the ground truth $m$, the decrease in error is not significant enough to warrant continuing the algorithm. The stopping condition is derived from the patterns in the eigenvalue of the chosen eigenvector in Algorithm 2. See Table 15. Notice the switch in eigenvalues' sign when we cross the ground truth $m$. This indicates we need to stop. Note that the trend in Table 15 continues for further iterations too. For example, for SNR 15dB, at iterations 97, 98, and 99, the chosen eigenvalues are 0.9477, -1.0000, and 0.9490,

Table 15: eigenvalues of the chosen eigenvectors over iterations at different SNRs for $m = 25, n = 500$

| Iter | SNR = 10 | SNR = 15 | SNR = 20 | SNR = 25 |
|------|----------|----------|----------|----------|
| 0 | -1.0703 | -1.0312 | -1.0078 | -1.0054 |
| 1 | -1.0000 | -1.0000 | -0.9996 | -0.9994 |
| 2 | -1.0000 | -1.0000 | -1.0000 | -1.0000 |
| 3 | -0.9971 | -0.9967 | -0.9945 | -0.9949 |
| 4 | -1.0000 | -1.0000 | -1.0000 | -1.0000 |
| 5 | -0.9937 | -0.9870 | -0.9845 | -0.9835 |
| 6 | -1.0000 | -1.0000 | -1.0000 | -1.0000 |
| 7 | -0.9800 | -0.9706 | -0.9806 | -0.9785 |
| 8 | -1.0000 | -1.0000 | -1.0000 | -1.0000 |
| 9 | -0.9649 | -0.9530 | -0.9628 | -0.9637 |
| 10 | -1.0000 | -1.0000 | -1.0000 | -1.0000 |
| 11 | -0.9437 | -0.9410 | -0.9530 | -0.9393 |
| 12 | -1.0000 | -1.0000 | -1.0000 | -1.0000 |
| 13 | -0.9311 | -0.9264 | -0.9093 | -0.9293 |
| 14 | -1.0000 | -1.0000 | -1.0000 | -1.0000 |
| 15 | -0.9039 | -0.9042 | -0.8936 | -0.9131 |
| 16 | -1.0000 | -1.0000 | -1.0000 | -1.0000 |
| 17 | -0.8720 | -0.8435 | -0.8759 | -0.8853 |
| 18 | -1.0000 | -1.0000 | -1.0000 | -1.0000 |
| 19 | -0.8209 | -0.7951 | -0.8568 | -0.8419 |
| 20 | -1.0000 | -1.0000 | -1.0000 | -1.0000 |
| 21 | -0.7847 | -0.7250 | -0.7361 | -0.7834 |
| 22 | -1.0000 | -1.0000 | -1.0000 | -1.0000 |
| **23** | **-0.7140** | **-0.6659** | **-0.7211** | **-0.7172** |
| **24** | **-1.0000** | **-1.0000** | **-1.0000** | **-1.0000** |
| **25** | **0.9055** | **0.9701** | **0.9908** | **0.9970** |
| 26 | -1.0000 | -1.0000 | -1.0000 | -1.0000 |
| 27 | -0.9094 | 0.9711 | 0.9909 | 0.9972 |
| 28 | -1.0000 | -1.0000 | -1.0000 | -1.0000 |
| 29 | 0.9116 | 0.9722 | -0.9912 | 0.9973 |
| 30 | -1.0000 | -1.0000 | -1.0000 | -1.0000 |
| 31 | 0.9136 | 0.9726 | -0.9914 | 0.9974 |
| 32 | -1.0000 | -1.0000 | -1.0000 | -1.0000 |
| 33 | 0.9141 | 0.9730 | -0.9916 | 0.9974 |
| 34 | -1.0000 | -1.0000 | -1.0000 | -1.0000 |
| 35 | 0.9171 | 0.9738 | -0.9918 | 0.9975 |

respectively. The corresponding values for SNR 15dB are 0.9835, -1.0000, 0.9836. Similar patterns are also observed at other SNRs, with values 0.9948, -1.0000, and 0.9949 at SNR 20dB, and 0.9983, -1.0000, and 0.9984 at SNR 25dB. However, *this pattern generally appears only when $m \ll n$.*

Consider another example where $m = 90$ and $n = 100$, as shown in Figure 19. Notice that positive eigenvalues appear at around iteration (update) 40 for most SNRs (error increases instead of decreasing at this step, indicating that the Householder vector chosen corresponds to an eigenvector with a positive eigenvalue) in the first plot and at around iteration 12 in the second plot.

We observe that in general, when the eigenvalue of the eigenvector corresponding to the Householder vector is positive and just crosses the value $0.9$, the error is close to its optimal value. This is merely a heuristic and is chosen based on various trends seen across $(m, n)$ pairs and different ground truth Householder vectors and noise types. An alternate naive approach is to run the algorithm until $k = n$ and pick the final matrix. This can, however, under some circumstances, as illustrated in the left plot (for SNR=20dB) of Figure 19, be detrimental.

Table 16: Eigenvalues of the chosen eigenvectors over iterations at different SNRs for $m = 25, n = 30$

| Iter | SNR = 10 | SNR = 15 | SNR = 20 | SNR = 25 |
|------|----------|----------|----------|----------|
| 0 | -1.1687 | -1.0364 | -1.0200 | -0.9929 |
| 1 | -0.9762 | -0.9996 | -0.9524 | -0.9520 |
| 2 | -1.0000 | -1.0000 | -1.0000 | -1.0000 |
| 3 | -0.8733 | -0.8999 | -0.8465 | -0.8662 |
| 4 | -1.0000 | -1.0000 | -1.0000 | -1.0000 |
| 5 | -0.6352 | -0.8248 | -0.7130 | -0.7468 |
| 6 | -1.0000 | -1.0000 | -1.0000 | -1.0000 |
| 7 | -0.3383 | -0.5317 | -0.4075 | -0.5864 |
| 8 | -1.0000 | -1.0000 | -1.0000 | -1.0000 |
| 9 | -0.1068 | -0.0726 | -0.2404 | -0.3862 |
| 10 | -1.0000 | -1.0000 | -1.0000 | -1.0000 |
| 11 | 0.0777 | 0.1793 | -0.0439 | 0.0558 |
| 12 | -1.0000 | -1.0000 | -1.0000 | -1.0000 |
| 13 | 0.4200 | 0.2499 | 0.2459 | 0.3087 |
| 14 | -1.0000 | -1.0000 | -1.0000 | -1.0000 |
| 15 | 0.5354 | 0.3968 | 0.4135 | 0.5041 |
| 16 | -1.0000 | -1.0000 | -1.0000 | -1.0000 |
| 17 | 0.7059 | 0.7000 | 0.6188 | 0.6274 |
| 18 | -1.0000 | -1.0000 | -1.0000 | -1.0000 |
| **19** | **0.8491** | 0.8002 | **0.7963** | **0.8518** |
| **20** | **-1.0000** | -1.0000 | **-1.0000** | **-1.0000** |
| **21** | **0.9205** | 0.8774 | **0.9407** | **0.9169** |
| 22 | -1.0000 | **-1.0000** | -1.0000 | -1.0000 |
| 23 | 0.9791 | **0.9579** | 0.9788 | 0.9584 |
| 24 | -1.0000 | -1.0000 | -1.0000 | -1.0000 |
| 25 | 0.9963 | 0.9961 | 0.9985 | 0.9997 |
| 26 | -1.0000 | -1.0000 | -1.0000 | -1.0000 |
| 27 | 0.9984 | 0.9996 | 0.9999 | 0.9998 |
| 28 | -1.0000 | -1.0000 | -1.0000 | -1.0000 |
| 29 | 0.9999 | 0.9999 | 0.9999 | 0.9999 |
| 30 | -1.0000 | -1.0000 | -1.0000 | -1.0000 |

Finally, even for very low SNR values (for example, SNR=2dB), the heuristic works well. Even though it doesn't give optimal results, the decrease in error after the point at which the algorithm terminates is not significant enough to warrant a change in the heuristic.

Our chosen heuristic is consistent with the theoretical analysis: beyond the chosen point, in general, further reflectors primarily capture noise rather than signal. While more sophisticated model-selection criteria (e.g., cross-validation, information-theoretic scores) could be applied, we find this simple rule to be robust across datasets and noise levels. Importantly, it preserves the anytime nature of our method: at each iteration, we can either continue or stop with a valid orthogonal approximation.

### E.4 ON THE ORIGINAL EIGENVALUE SPECTRUM

If many of the eigenvalues of the input orthogonal matrix are large and negative (for example, close to -1), the error will decrease at a steeper rate in the initial iterations, and the rate of decrease of error will eventually slow down. The more positive eigenvalues there are, the slower the initial convergence will be. Though eventual convergence is guaranteed, if we want an approximation in $\mathcal{H}_m$ with $m \ll n$, ideally, the input orthogonal matrix should have an eigenvalue spectrum shifted towards -1. Our method can also recover products of Givens rotation matrices in iterations less than or equal to the number of rotations, when applied at different locations.

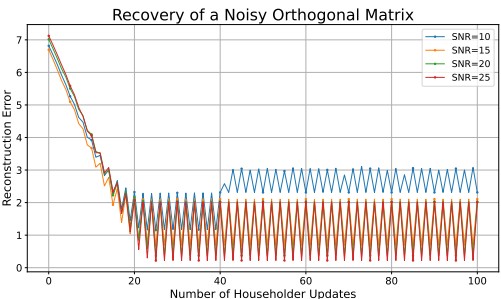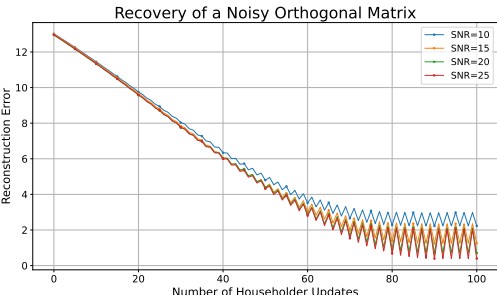

Figure 19: Reconstruction error vs. number of Householder updates at varying signal-to-noise ratio (SNR) for $m = 25, n = 30$ (left) and $m = 90, n = 100$ (right).

### E.5    EXTENSION TO COMPLEX-VALUED OR NON-ORTHOGONAL MATRICES

Our current framework and analysis focus on real orthogonal matrices. Extensions to complex unitary matrices and generalized reflectors are both natural and impactful directions. We briefly discuss these possibilities and the potential for richer building blocks that broaden expressiveness without sacrificing structure.

Note that for matrices that are too far from orthogonal matrices, we still get graceful degradation, as seen in the ViT experiments in section D.4. However, these cases become equivalent to the low SNR regimes shown in Figure 4. This is where the theoretical guarantees break down, since $\alpha > 1/2$ isn't guaranteed anymore (see section 4). Thus, the error cannot be lower than a certain threshold, in general.

**Extension to non-square matrices**    Consider matrices of the form $\mathbf{W} \in \mathbb{R}^{p \times n}$, where $p < n$ (a similar approach to what we show can be adopted for $n < p$ as well). We can simply pad zeros to the rows/columns to convert this to a square matrix. On running our algorithm on such matrices, we observe performance similar to the noisy and non-orthogonal cases, where we observe a decrease in error for a substantial number of iterations, post which the error oscillates between a couple of values. This is useful, for example, in convolutions where the input and output dimensions are generally different. An illustrative example is Orthogonal Convolutional Neural Networks (OCNN) (Wang et al., 2020).

Concrete algorithmic formulations and theoretical guarantees for such extensions are promising topics that we aim to address in future work, targeting applications in quantum computing, signal processing, and beyond.

A note on LLM usage: Used for grammar checks and fixing typos.

A note on supplementary material: All codes have been included in the supplementary material, except the ones for Table 12 and Figures 6, 7.

