# OpenReview forum: "One Reflection at a Time: Provable Compression and Denoising of Orthogonal Matrices"
_ICLR.cc/2026/Conference — Submitted to ICLR 2026_

### Official Review · Reviewer_AtBr · 2025-10-30

**Soundness:** 3
**Presentation:** 3
**Contribution:** 3
**Rating:** 6
**Confidence:** 2

**Summary:**

The paper presents a greedy, eigenspace-based algorithm for approximating or recovering arbitrary orthogonal matrices using a minimal number of Householder reflections. The approach is theoretically grounded, with provable accuracy guarantees, robust to noise, and broadly applicable. By recursively projecting the input onto eigenspaces induced by Householder reflections, it delivers verifiable approximation bounds and a consistent, interpretable decomposition. The framework is especially relevant for applications that require structured orthogonality, such as orthogonal parameterizations in RNNs and Transformers.

**Strengths:**

* The proposed greedy eigenspace–based algorithm approximates or recovers arbitrary orthogonal matrices using the minimal number of Householder reflections.

* The method denoises structured orthogonal matrices under noise, leveraging the same greedy eigenspace framework.

* The approach is versatile, with demonstrated applicability to dictionary learning, model initialization, and neural network compression.

**Weaknesses:**

I acknowledge that I am not fully familiar with prior algorithms or works related to Householder-based orthogonal matrix approximation, and therefore cannot conclusively determine whether the comparisons in this paper are sufficiently comprehensive or representative of the state of the art. My comments below are thus partial and intended for reference only:

* The theoretical assumptions appear relatively strong — the noise analysis relies on GOE/Ginibre models, while the noise structures in practical deep networks (e.g., non-Gaussian or structured noise) are usually more complex. It would be valuable to discuss whether the theoretical results in Section 4 can be extended to non-Gaussian noise settings.

* The paper does not clearly analyze the improvement or difference between Algorithm 2 and Algorithm 1. From the expression $V_{k+1} = H_k \hat{V}^\top V = H_k V_k$, it seems that the two algorithms follow essentially the same update rule, except for the stopping criterion. If this understanding is incorrect, the authors should explicitly clarify the distinction between the two algorithms.

* In Lemma 4, Theorem 3, and Theorem 4, the results are said to hold with high probability (w.h.p.), but the corresponding probability level is not specified. Providing explicit probability bounds or constants would make the theoretical claims more concrete.

* The appendix compares the proposed method with SVD, QR, and Hm-DLA in terms of reconstruction error, but does not include a comparison of computational time among the three. Including runtime results would strengthen the empirical evaluation.

**Questions:**

* Does $\hat H$ denote the denoised estimate of the perturbed matrix $V$ introduced in Lemma 4? The paper uses $V$, $\hat V$, and $\hat H$ without clear, explicit definitions, which is confusing.

* The main text presents numerical results only for Algorithm 1, yet the description suggests Algorithm 2 is also a primary contribution. I recommend moving (at least a representative subset of) Algorithm 2’s numerical comparisons from the appendix into the main paper to substantiate its impact.

* Please state the exact value(s) of the stopping threshold $\epsilon$ used in the experiments (e.g., for Figures 1 and 3), and briefly justify the choice.

---

> ### Author Response · Authors · 2025-11-17
>
> Dear Reviewer AtBr, thank you so much for the detailed feedback. We have made the following attempts to answer the points brought up.
>
> 1. We have extended the theoretical results in section 4 for non-Gaussian settings and also conducted additional experiments to validate our claims. These can be found in section C.5 (pages 29-30) and D.3.4 (pages 40-41). We would like to note that these bounds may have some scope for tightening.
> 2. Yes, you are spot on with your assessment here- the two algorithms essentially rely on the same update rule. We wanted to illustrate that the idea of greedy projections is a general technique in identifying the best subspace to project on, normally, and even when there is noise, making it a principled approach with controllable error.
> 3. The exact probability levels have now been added in section C.5 (lines 1574-1581).
> 4. We have now added tables comparing run times and speed-ups- for Hm-DLA, this can be found in section D.2.2 (page 36); for SVD, this can be found in section D.1.4 (pages 34-35). The QR method also uses Householder reflections during inference, so any gain there would be reflected in our method as well. However, it isn't as accurate as ours.
> 5. Yes- we have now added a small phrase regarding the same in the statement of the Theorems and clarified notations in sections 3.1 and 3.2 (pages 3-4).
> 6. A couple of representative denoising examples have now been moved into the main text in section 5 (lines 419-425 and 486-513).
> 7. We had been using $0.00005$; however, this was only for the sake of illustration- to show that at $m$ iterations, if $\mathbf{V} \in \mathcal{H}_m$, we would indeed recover the decomposition. We have slightly modified it to match the theory- continue to run the algorithm until the minimum eigenvalue is 1 (essentially giving $\mathbf{V}_k=\mathbf{I}$ and $\hat{\mathbf{V}}=\mathbf{V}$). In the original version, a note regarding this was included from lines 239-241, which we have removed since the condition is made explicit in the algorithm itself.
>
> We hope that we have been able to do justice to your questions and would love to discuss more!

---

### Official Review · Reviewer_2ndh · 2025-10-31

**Soundness:** 3
**Presentation:** 3
**Contribution:** 2
**Rating:** 2
**Confidence:** 4

**Summary:**

The goal of the paper is to approximate a $n \times n$ real orthogonal matrix $V$ by a product of householder reflections. A householder reflection is a matrix of the form $I - uu^T$ for a unit vector $u$. The main guarantee is an algorithm which outputs a product $V'$ of $m$ (assuming it is even) householder matrices and satisfies $\|V - V' \|_F^2$ $\le  O( n - Tr(V) - m +\sum_i \lambda_i  )$, where the sum is only over the smallest $m$ eigenvalues of $(V + V^T)/2$. The algorithm works by iteratively finding a top singular vector of $V$ minus the partial product found so far and adding that as the next $u$ in the new householder reflection.

**Strengths:**

The problem of approximating orthogonal matrices by householder reflections is a cute numerical analysis problem and empirically the algorithm seems to perform well on synthetic matrices. The proposed algorithm is also quite simple to implement.

**Weaknesses:**

It was not clear to me if the error guarantees of the main theorem (Theorem 2) are very meaningful. In the regime where we think of $m$ as small, it seems like the $n$ value in the error term could dominate, which would be not be so nice since one can easily get an error $\sqrt{n}$ approximation by approximating an orthogonal matrix trivially by the all zeros matrix. There also seems to be a slight issue in the statement of the theorem since the algorithm has an $\epsilon$ parameter but it is not present in the theorem statement.

I am also not sure if this specific type of approximation is well justified in practice. The authors argue that existing method such as SVD 'lack structure' but don't address why householder reflections are a 'good structure' to have in practice. Sure, geometrically they are nice but a low rank approximation is also quite easy to store/fast to compute on in practice. It is also not so clear to my why there is a strong need to approximate orthogonal matrices. I understand some layers in some neural network architectures are orthogonal matrices, but a vast majority of weight layers are not. Since their method only works for orthogonal matrices and uses a very niche form of approximation (product of householder reflections), this feels like a niche result where I cannot identify a community that would be very excited about it.

I am also not convinced by their experimental results which are conducted on very synthetic matrices, exactly tailored to their theorem statement. While this can be nice as a proof of concept result, it does not convince the reader that this task is 'widely applicable' as stated in the intro. Given that this problem itself is not so theoretically interesting version of matrix approximation, it is disappointing that the experiments are very lacking in their scope. In terms of baselines, I don't think just comparing to prior work using only household reflections is enough. This is because the central thesis of the paper is that we must have fast approximations to orthogonal matrices, so they why not consider other approximation methods as well? For example, SVD or other much faster low rank approximation algorithms based on sampling or sketching. These are not discussed in the paper.

Ultimately, I think this paper would be a better fit in a numerical analysis / linear algebra journal or conference.

**Questions:**

Can the authors comment on the tightness of the error bounds? How does it compare with the SVD bound? How should the reader think about what the right 'scale' of the error term? I.e., why is the error bound given in Theorem 2 the natural bound? Having any sort of lower bound on the error term (i.e. any approximating using $m$ householder reflections must incur some error) would help.

---

> ### Author Response · Authors · 2025-11-17
>
> Dear Reviewer 2ndh, thank you so much for the detailed feedback. We have made the following attempts to answer the points brought up. We've had to split the response into two comments due to the character limit.
>
> 1. There are two regimes we need to consider here: the first, where the ground truth orthogonal matrix is a product of a few Householder reflections (small $m$). In this case, the trace term actually compensates for the small value of $m$, giving $\mathcal{O}(1)$ error. For example, consider a $10000 \times 10000$ matrix that is a product of say, $m=2$ Householders. Our bound provides an error of $\mathcal{O}(1)$ (solely dependent on $\mathbf{u}_1^\top\mathbf{u}_2$), significantly lower than the error obtained on approximating with all 0, identity or other fixed matrices, which would give an error close to $100$. In fact, we prove that our method is optimal irrespective of $n$ in section B.4 with an illustration for $\mathbf{V} \in \mathcal{H}_2$ in sections B.6.1 and B.6.2 (pages 21-24). The second regime is when the ground truth $\mathbf{V}$ is a product of $K \gg m$ Householder matrices. In this case, we are guaranteed to converge- with guaranteed improvement at every step. The trace term here, however, may not be able to compensate for a large $n$. This also makes sense, since otherwise, if we could get very low error at very small $m$- values for arbitrary orthogonal matrices (even though they themselves are a product of $K \gg m$ Householder matrices), one would be able to express most orthogonal matrices by a product of a few Householders, which individually, are not powerful enough to capture the entire space of all orthogonal matrices. Finally, regarding optimality, we have added section B.4 (page 20), proving the optimality of our approach.
> 2. We have updated the stopping criteria in algorithm 1 in order to avoid any confusion with $\epsilon$. In the original version, a note regarding this was included from lines 239-241, which we have removed since the condition is made explicit in the algorithm itself. We introduced this criterion solely for the purpose of illustrating that, at $m$ iterations, we effectively achieve zero error, as expected from the theory we derived. However, the theory and $\epsilon$ are not connected. Thus, we have removed this criterion.
>  3. The key distinction from SVD is that we want to approximate within the orthogonal group. This is because many modern ML architectures (orthogonal RNNs, normalizing flows, orthogonal transformers, special capsule networks, etc.) all require orthogonality in order to preserve gradient norms and ensure invertibility. A low-rank SVD truncation, while compact, destroys orthogonality and hence cannot be used as a drop-in replacement in such layers. Householder products, by contrast, preserve exact orthogonality at every truncation level, offering a continuous trade-off between model size and fidelity while maintaining the spectral norm constraint crucial for stable training and improved inference. Orthogonal matrices and their approximations are also key in sparse signal processing, such as ICA and dictionary learning. Furthermore, recent work has revealed that normalization and residual connections, both of which are extensively used in modern ML systems, push weight vectors towards orthogonality. Householder matrices are themselves also used in knowledge graph embeddings, positional encoding, linear RNNs, fast dictionary learning, and several modern variants of transformers. We have added some recent work for reference below.
>  4. We have tried to show the utility of our method across some of the applications we mentioned in the introductory part of our manuscript. These include a comparison with modern, fast, dictionary learning algorithms like Hm-DLA in section D.2.2, post raining compression for copy and sequential MNIST tasks using expRNN (section D.4.1), post raining compression and compressed initialization matrices before training for Vision Transformers on CIFAR10 and Tiny ImageNet (D.4.2, D.4.3). Our method can be applied for compression across any orthogonal parametrization, independent of domain, with controllable trade-offs between amount of compression and accuracy. In the sections mentioned above (D.4.1), we have also compared with SVD and QR. Our method shows controllable and graceful trade-offs for the same. We also demonstrate the speedup of our method in section D.1.4. Additionally, we present a comparison with the Orthogonal Procrustes solution in section C.4.1.

---

> ### Author Response · Authors · 2025-11-17
>
> 5.  We can actually express the error at any given step exactly (equations 19, 20 in section B.6). That is, we could express the error at any step without performing any computation at all, apart from the single step required to determine the spectrum of the original matrix. Keeping the parameter count constant, for a fixed $K$, the SVD error bound is $\sqrt{n-K/2}$, which is much higher than our bound. Even if we consider just $K$ (not equating the parameters), the error is $\sqrt{n-K}$. To illustrate this, we have added section D.2.4 (pages 37-38) to demonstrate the superiority of our method over SVD. Moreover, we have also enhanced the experiments and illustrations for the Orthogonal Procrustes solution, which stems from SVD, in section D.3.1 (pages 38-39).
>
> 6. In terms of scale, effectively depending on $m$, we can get close to $O(1)$ error in very few steps when $m$ (the number of Householders multiplied to generate the ground truth matrix) is small. In general, the error reduces in a very predictable fashion, with very specific eigenvalue pattern changes at every iteration (please refer table 14, section E.2 (page 47) for an illustration). This is a natural bound since we are working in the eigenspace instead of the column space. Thus, it is expected that the error will be predominantly dependent on the eigenvalues at various iterations. Furthermore, even trace is effectively the sum of all eigenvalues, and the $m$ terms arise from the minimum eigenvalue becoming -1 at alternate iterations. Thus, we are effectively moving away from maximum error $\sqrt{n}$ at each step, based on the minimum eigenvalue at that iterate. As stated above, we have added section B.4 to prove the optimality of our approach.
>
> We hope that we have been able to do justice to your questions and would love to discuss more!
>
> [1]Mhammedi et al. Efficient orthogonal
> parametrisation of recurrent neural networks using householder reflections
>
>
> [2]Mathiasen et al. aster orthogonal parameterization with householder matrices
>
>
> [3]Tomczak et al. mproving variational auto-encoders using householder flow.
>
>
> [4]Yuan et al. Bridging the gap between low-rank and orthogonal adaptation via householder reflection adaptation
>
> [5]Huang et al. Orthogonal transformer: An efficient vision transformer backbone with token orthogonalization.
>
>
> [6]Rusu et al. Fast orthonormal sparsifying transforms based on householder reflectors
>
> [7]Narayanan et al. Blind compressive sensed signal reconstruction using householder transforms
>
>
> [8]Siems et al. DeltaProduct: Improving State-Tracking in Linear RNNs via Householder Products
>
>
> [9]Yang et al. PaTH Attention: Position Encoding via Accumulating Householder Transformations
>
>
> [10]Li et al. HousE: Knowledge Graph Embedding with Householder Parameterization
>
>
> [11]Lu et al. The Orthogonality of Weight Vectors: The Key Characteristics of Normalization and Residual Connections

---

### Official Review · Reviewer_AunQ · 2025-11-01

**Soundness:** 3
**Presentation:** 3
**Contribution:** 3
**Rating:** 6
**Confidence:** 3

**Summary:**

The paper extends the constructive decomposition of [Uhlig01] for noisy orthogonal matrices. The generalized idea is similar to that of [Uhlig01], where, by the so-called CDS theorem, any $n\times n$ orthogonal matrix $V$ can be factorized exactly into $n-m$ Householder reflections, where $m=dim(ker(V-I_{n}))$. Concretely, we have $H_{n-m}\dots H_{1}V=I$ where $H_{k}$ are Householder reflection, and therefore $V=H_1\cdots H_{n-m}$. The key innovation of the paper is to provide a different scheme for selecting the Householder vector when doing this decomposition. In [Uhlig01], the Householder vector $u$ is constructed by first randomly selecting a unit vector $x$ satisfying $Ux\neq x$, then set $u=Ux-x$, where $U=H_{k}\dots H_{1}V$ is the symmetrized partial decomposition of $V$. In this paper, the author sets $u$ to be the eigenvector of symmetrized $U$ with the minimal eigenvalue. The author shows that this choice is also good for denoising, i.e., $H_1\cdots H_{n-m}$ is a good approximation of $V'$ when $V=V'+N$ where $V'$ is orthogonal and $N$ is a real Ginibre matrix. The author also provides a projection-quality bound, which appears to be novel. Empirically, the method outperforms [Uhlig01] in approximation error for both noiseless and noisy targets, and compares favorably to Hm‑DLA for orthogonal dictionary learning.

**Strengths:**

The paper proposed a novel algorithm for denoising approximation of orthogonal matrices, and provided a comprehensive theoretical analysis of the error bound. The experiments were also well structured and comprehensive. The writing was easy to follow for the most part. The pseudocode of Alg. 1/2 is concise and easy to implement.

**Weaknesses:**

The biggest weakness to me is that the author failed to convey the importance of the task of denoising an orthogonal matrix. Besides ViT QKV, when do we have a near orthogonal matrix with the nice noise distribution as tested in the paper? If we parameterize something with an orthogonal matrix, then there is no noise and the method from [Uhlig01] suffices. If we start with some random square matrix, then there is no guarantee that the matrix can be written in the form of orthogonal + i.i.d. Gaussian as assumed in the paper.

On experiment:
- The paper emphasizes reconstruction error and accuracy retention (e.g., expRNN Copy Task, seqMNIST, ViT), but does not provide benchmarking on inference/training speedups or peak memory of their methods.

Also, the writing is sometimes too terse to be clear. For example:
- In Prior Work that introduces the method of [Uhlig01], the definition of $x$ is not explained.
- In Theorem 4, the definition of $\Delta$ and $\theta$ was not explained.

**Questions:**

- In the simulation, it seems that the bound is always below the ground truth error. I get that the bound is probabilistic, but it cannot be that the bound is violated all the time?
- The stopping rule for algo 1 is pretty clear. But for algo 2, how does it link to the theory (e.g. theorem 4)? It is not very clear to me.

---

> ### Author Response · Authors · 2025-11-17
>
> Dear Reviewer AunQ, thank you so much for the detailed feedback. We have made the following attempts to answer the points brought up.
>
> 1. We have enhanced section 4 (lines 304-319) to include examples from literature and applications requiring denoising orthogonal matrices. These range from applications in computer vision, 3D-graphics and geometry, to point cloud mapping and standard signal processing tasks like ICA [1-6]. These also find applications in modern ML architectures and techniques especially involving either orthogonal layers/weight matrices or even end-to-end orthogonal transformers. To address the point regarding i.i.d. Gaussian noise, we have added more experiments to our original anisotropic noise experiment, as well as conducted newer experiments on other structured noise matrices - low rank, uniform, Bernoulli-masking, as well as sign flip.
> 2. We have added a table analyzing inference and memory (static storage and peak memory) in section D.1.4 (pages 34-35) for general inference, instead  of emphasizing a particular specialized setup.
> 3. We have added the definition of $\mathbf{x}$ for [Uhlig01]- thank you for the careful perusal!
> 4. We have made an addition for this in the proof (line 1511). $\Delta$ is been briefly touched upon in the Theorem statement, and elaborated on in equations 40 and 43.
> 5. There are three plots in the figure 1 for a fixed m- one of them is the ground truth error- i.e., the error between the original Householder matrices multiplied to construct the orthogonal matrix and the final orthogonal matrix. The second one is the difference between the product of Householder matrices we obtain by applying our algorithm and the orthogonal matrix, and the third one is the bound. In the first plot of figure 1, the bound is farther away (more reconstruction error) than the ground truth product, which aligns with what you mentioned. We would also like to add that we have actually derived the exact error expression (Equations 19 and 20, page 21). Thus, we can actually obtain the 'bound' that matches the exact error we get when we apply our algorithm. Finally, the cause of this observed pattern has all to do with the non-uniqueness of orthogonal matrix decompositions. We have shown that our method is provably optimal (section B.4, page 20). Thus, our method will always do at least as well as the ground truth error. It so happens that in most experiments we run, the decomposition we obtain is closer to the final orthogonal matrix- i.e., the matrices used during generation weren't the optimal choices.
> 6. For algorithm 2, if we know $m$, i.e., the number of Householder matrices that have been used to generate the ground truth orthogonal matrix, we can simply stop at $m$ iterations- plug this $m$ into all the proofs and the results hold. However, the $\epsilon$ stopping parameter was introduced to address cases where $m$ is unknown. $\epsilon$ was chosen based on a heuristic, as observed in nearly all of our experiments and error reduction patterns. The details of this procedure are presented in section E.3.
>
> We hope that we have been able to do justice to your questions and would love to discuss more!
>
>
> [1]Bhamre et al. Orthogonal matrix retrieval in cryo-electron microscopy.
>
> [2]Wang et al. Exact and stable recovery of rotations for robust synchronization.
>
> [3]Daigavane et al. Matching the optimal denoiser in point cloud diffusion with (improved) rotational alignment.
>
> [4]Lawrence et al. A purely algebraic justification of the Kabsch–Umeyama algorithm.
>
> [5]Huang et al. Deterministic independent component analysis.
>
> [6]Arora et al. Provable ICA with unknown Gaussian noise, with implications for Gaussian mixtures and autoencoders.

---

### Official Review · Reviewer_JqEL · 2025-11-01

**Soundness:** 3
**Presentation:** 3
**Contribution:** 3
**Rating:** 8
**Confidence:** 4

**Summary:**

This paper introduces a greedy eigenspace-based algorithm for decomposing orthogonal matrices into simpler building blocks, called Householder reflections. At each step, the algorithm identifies the most important direction in the matrix by finding a specific pattern in its symmetric part, then uses this to create the next reflection component. The method ensures the approximation stays orthogonal at every level K, provides mathematical guarantees on approximation quality (Theorem 2), and automatically identifies when exact reconstruction is achieved (Theorem 1). For cleaning noisy matrices, when random noise decreases appropriately with matrix size, the method achieves proven error reduction with high probability (Theorems 3-4). Testing shows the approach works well in practice: synthetic examples confirm the theoretical predictions, comparisons demonstrate over 80% error reduction versus previous methods by Uhlig and Hm-DLA, and applications to recurrent neural networks and vision transformers show effective compression with controllable accuracy trade-offs, reducing storage needs from quadratic to linear scaling.

**Strengths:**

- **Originality:** The work provides the first greedy projection algorithm onto $\mathcal{H}_K$ with per-iteration error bounds and orthogonality preservation at each truncation. Theorem 1's minimality certificate connects eigenspace dimension with algorithmic termination. The denoising theory (Theorems 3-4) provides a rigorous high-dimensional analysis of the Householder-based factorization under random-matrix noise. The eigenspace-based approach offers fresh geometric insight compared to column-space methods like QR.

- **Quality:** The technical details are rigorous, with complete proofs in Appendices B-C and careful treatment of odd/even iteration distinctions (Lemmas 2-3). Experimental design is comprehensive, spanning multiple matrix distributions (Gaussian, Bernoulli, sparse, chi-squared), explicit bound-vs-empirical comparisons (Figures 1-2), and diverse architecture ablations (expRNN, ViT).

- **Clarity:** The paper progresses logically from motivation through theory to applications. Algorithm 1 is straightforward, requiring only eigenvalue decomposition per iteration. Mathematical exposition balances rigor with accessibility, with complete proofs.

- **Significance:** The work addresses a critical gap at the intersection of numerical linear algebra and modern ML. By providing principled orthogonal matrix compression with provable error control and \(O(Kn)\) storage/computation, it enables practical deployment in resource-constrained settings.

**Weaknesses:**

- **Restrictive isotropic noise assumption limits denoising applicability:** The denoising theory (Theorems 3-4) assumes noise matrices from Ginibre or GOE ensembles with i.i.d. or symmetric i.i.d. entries, which is unrealistic for many practical scenarios. Real-world noise often exhibits structure, row correlations, or low-rank perturbations, which violates the isotropic assumption. Section D.4.4 introduces anisotropic noise (column scaling factors 0.5-2) and shows the method "works" in Figure 11, but explicitly states "we cannot provide theoretical guarantees here." This leaves critical questions unanswered: At what level of anisotropy does the method fail? Although the proposed technique has its merits, elaborating on the noisy reconstruction would improve the paper.

- **Fragmented complexity analysis:** Computational costs are scattered across Sections 3.1, A.3, D.1, E.1 without unified guidance. Key questions remain: (i) How many Lanczos iterations are needed versus spectral gap $\lambda_2 - \lambda_1$? (ii) What are wall-clock times for decomposition versus inference savings in neural experiments? Table 3 shows losses but no timing. A single table comparing FLOPs, memory, time, and accuracy for $K$ across all settings would clarify when the method is cost-effective.

- **Limited baseline comparisons:** Comparisons against Uhlig (Figure 3) and Hm-DLA (Figure 7) establish superiority, but gradient-based optimization directly minimizing $\vert\vert V - H_1 \cdots H_K\vert\vert_F^2$ over $Kn$ Householder parameters are absent. Adding this would clarify whether greedy approximation achieves competitive quality with the "global" optimum or sacrifices accuracy for speed.

- **Presentation:** The equations are not numbered in the manuscript, which makes it hard to refer to them. Is the LHS of both equations on Page 7 (Section 4,2), line 329 and line 336, the same?

**Questions:**

**1. Extending denoising theory to structured noise:** Can the authors extend Theorems 3-4 to handle anisotropic or structured noise? An empirical characterization of the failure mode by plotting denoising error versus the anisotropy ratio (range of column scaling factors) to identify when performance degrades would improve the paper.

**2. Spectral gap and Lanczos convergence:** For experimental matrices, report: (a) gap distributions, (b) actual Lanczos iterations to reach $10^{-8}$ tolerance at each $k$, (c) whether iteration counts grow or the gap widens via deflation. Do trained orthogonal matrices exhibit structured spectra, making Lanczos particularly fast/slow? Does $O(n^2)$ hold in practice?

**3. Orthogonality in NN compression:** Tables 6-7 demonstrate catastrophic failure when compressing all ViT blocks (14% accuracy), yet the authors provide no quantitative analysis of why. Can the authors report the $\vert\vert V^\top V - I\vert\vert_F$ for all attention matrices before compression?

---

> ### Author Response · Authors · 2025-11-17
>
> Dear Reviewer JqEL, thank you so much for the detailed feedback. We have made the following attempts to address the points raised.
> 1. **Restrictive isotropic noise assumption limits denoising applicability**: We have extended Theorems 3 and 4- section C.5: (pages 29-30) now contain bounds for some other noise matrices. We would like to note, however, that these bounds may have some scope for tightening. To support the results empirically, we conducted additional experiments, as described on pages 40-41. We analyze fine-grained anisotropy factors at increments of 0.5, from 0.5 to 4, and at increments of 3, from 1 to 10, which better characterizes when our method fails, primarily depending on whether the original spectrum changes significantly. We observe a gradual increase in reconstruction error with increasing anisotropy, which supports our theoretical claims.
> 2. **Fragmented complexity analysis**: We have now brought all computational cost analyses together to section 4.1, pages 33-35. In general, we observe that 10 Lanczos iterations suffice for convergence (where convergence implies that the minimum eigenvalue returned by eigsh is $\epsilon$-close to that returned by eigh). A single table for time and another one for memory speed-up has been added. We adopt the most generic case here, comparing the time required by our sequential Householder products as compared to dense multiplication. We would like to add that our implementation, as of now, uses NUMBA, meaning that there is still some scope for optimization, and the speed-up offered by our method is potentially higher.
> 3. **Limited baseline comparisons**: A comparison with gradient-based optimization has been added for both the clean (pages 36-37) and noisy (page 39) environments. We observe that our method does indeed identify the decomposition (and thus the Kn Householder parameters) accurately, and on most occasions, before the gradient techniques, owing to the non-convex loss landscape.
> 4. **Presentation**: This has been fixed now. Although the observed perturbed matrix and the ground truth orthogonal matrices are the same, the two equations themselves are slightly different in terms of the individual $\mathbf{H}'$s we attempt to recover. Equation 6 (line 354) uses the Householder matrices that would've been obtained had we had a noiseless setup and had attempted to recover the original orthogonal matrix. Though this might seem unnecessary at first, this step is critical in identifying theoretical bounds, since we define error in terms of the proximity of our recovered Householder vectors to those in Equation 6, not the ones in Equation 5. This is what we mean by 'trick' in line 349.
> 5. **Extending denoising theory to structured noise**: We have attempted to answer this in point 1 above.
> 6. **Spectral gap and Lanczos convergence**: The gap distributions have been added (page 33). It takes roughly 10 iterations for Lanczos' algorithm to converge for most cases. It can also converge at 5-6 iterations for some matrices, though we did observe that it would fail to converge on the odd occasion for these values. For all experiments in the original manuscript, we had used 20 iterations (included in the original manuscript)- and we have verified those with 10 iterations in the updated version. Iteration counts remain roughly the same across various spectra. Certainly, the trained orthogonal weight matrices exhibit a gap distribution with slightly higher gaps on average; however, since we observe convergence at a very small number of iterations regardless, we do not observe much change. Therefore, $\mathcal{O}(n^2)$ does hold in practice.
> 7. **Orthogonality in NN compression**: We have added these values in section D.4.2 (page 44). The failure can be best justified by the fact that even though our algorithm can recover matrices that are reasonably far from orthogonal, this isn't exact, and a product of these matrices leads to a cascading effect, leading to the poor results when all blocks are replaced.
>
>
> We hope that we have been able to do justice to your questions and would love to discuss more!

---

### Author Response · Authors · 2025-11-17
**Response Summary**

First and foremost, we would like to thank all contributors to the review process for their time and effort in reviewing our manuscript, and for the thorough feedback provided. We are pleased that the reviewers recognize the novelty, theoretical guarantees, experimental validation, and practical potential of our eigenspace-based approach – in particular, its per-step error guarantees, denoising robustness, and efficient compression.

We also recognize the importance of the concerns and questions brought up by the reviewers and have addressed them in the updated version of the manuscript (with more details shared in the individual comments). At a high level, the following improvements have been made:

1. We have added a theorem on the optimality of our approach.
2. Theorems 3 and 4 have been extended to more general noise distributions, with corresponding experiments for various distributions (including failure characterization under anisotropic noise), further establishing the utility of our method.
3. Gap distributions and convergence analysis have been added for Lanczos' algorithm.
4. All arguments and results regarding computational complexity have been brought together under one umbrella. Furthermore, we have added the timing, static memory, and peak memory results. We have also added the wall clock time comparisons with Hm-DLA.
5. A comparison with gradient-based optimization has been included for both clean and noisy settings.
6. A "Comparison with SVD" section has been added. We have also added a few more experiments and illustrations pertaining to the Orthogonal Procrustes comparison to further outline the benefits of our method compared to SVD-based approaches.


As an additional note, due to the size of the new tables and plots, not all results could be included in this response thread; however, all referenced updates are clearly presented in the revised manuscript.

---

### Meta-Review · Area_Chair_Ak4w · 2025-12-10

**Summary:**

This paper proposes a greedy eigenspace approach for approximating an input orthogonal matrix using Householder reflections, i.e. matrices of the form $I-uu^\top$, where $u$ is a unit vector. The paper gives per-iteration error bounds and includes guarantees for exact reconstruction and robustness to some forms of noise, with experiments on synthetic datasets demonstrating potential.

While reviewers agreed that the paper is technically solid, initial concerns are that the denoising guarantees assume idealized isotropic noise and the problem is somewhat specialized. This was perhaps more evident as the initial experiments focused on synthetic datasets and Householder-based baselines without broader comparisons or unified runtime analysis. Though the rebuttal partially addressed other forms of noise and added more experiments, the comparisons to SVD are not quite clear.

**Reviewer Concerns:**

I believe questions about notational ambiguity or omission were resolved, as were questions about other forms of noise (though only to a small extent, as the authors note). Additional experiments were also conducted, which addressed some concerns about other comparisons to other statistics.

**Reviewer Scores:**

It is not evidently clear to me that any of the reviewer scores would have changed, either increase or decrease, following the initial rebuttal. As multiple experiments were conducted and additional theory was included, I do not believe the scores would have decreased. On the other hand, while many of the questions were addressed, it is not obvious that a number of important questions, such as the practical motivation for the work and the quality of the error bounds, were fully resolved. Finally, it is worth mentioning that review quality and reviewer experience were considered in the interpretation of these scores.

---

### Decision · Program_Chairs · 2026-01-26

Reject